# Projected Coupled Diffusion for Test-Time Constrained Joint Generation

**Hao Luan**[1*]  **Yi Xian Goh**[2*†]  **See-Kiong Ng**[1,3]  **Chun Kai Ling**[1]

[1]School of Computing, National University of Singapore
[2]Faculty of Computer Science and Information Technology, Universiti Malaya
[3]Institute of Data Science, National University of Singapore

haoluan@comp.nus.edu.sg, u2102756@siswa.um.edu.my,
{seekiong, chunkail}@nus.edu.sg

## Abstract

Modifications to test-time sampling have emerged as an important extension to diffusion algorithms, with the goal of biasing the generative process to achieve a given objective without having to retrain the entire diffusion model. However, generating jointly correlated samples from multiple pre-trained diffusion models while simultaneously enforcing task-specific constraints without costly retraining has remained challenging. To this end, we propose *Projected Coupled Diffusion* (PCD), a novel test-time framework for constrained joint generation. PCD introduces a coupled guidance term into the generative dynamics to encourage coordination between diffusion models and incorporates a projection step at each diffusion step to enforce hard constraints. Empirically, we demonstrate the effectiveness of PCD in application scenarios of image-pair generation, object manipulation, and multi-robot motion planning. Our results show improved coupling effects and guaranteed constraint satisfaction without incurring excessive computational costs. Our code is available at https://github.com/EdmundLuan/pcd.

## 1 Introduction

Diffusion models have achieved remarkable success in generative modeling, with a plethora of applications ranging from image (Rombach et al., 2022), video (Ho et al., 2022), language (Li et al., 2022), graph generation (Niu et al., 2020; Madeira et al., 2024; Luan et al., 2025), as well as robotics (Janner et al., 2022; Chi et al., 2023; Carvalho et al., 2023). One of the crucial factors underlying these achievements is the use of *test-time* conditional sampling techniques such as classifier guidance (Dhariwal & Nichol, 2021; Song et al., 2021b), inpainting (Lugmayr et al., 2022; Liu et al., 2023), reward alignment (Uehara et al., 2025; Kim et al., 2025), and projection (Christopher et al., 2024; Sharma et al., 2024).

While these methods are primarily designed for sampling from univariate distributions, numerous real-world tasks require sampling highly correlated variables from joint distributions, *e.g.*, image pairs (Zeng et al., 2024), multimedia (Ruan et al., 2023; Tang et al., 2023; Hayakawa et al., 2025), traffic prediction (Jiang et al., 2023; Wang et al., 2024b), and multi-robot motion planning (Shaoul et al., 2025; Liang et al., 2025). Directly training a diffusion model to capture one single joint distribution is costly and inefficient. First, high-quality annotated datasets of joint behaviors are scarce, expensive, and often proprietary. One example is real-world traffic trajectory data, which is essential for prediction and planning in autonomous driving (Li et al., 2023). Second, training joint distributions becomes increasingly computationally demanding as the number of variables grows (Gu et al., 2024), and relearning the entire joint distribution becomes necessary when marginals are changed. For instance, coordinating robot teams may require *retraining the entire model* even if only one robot's behavior differs from its marginal of the pretrained joint for a new task.

Inspired by compositional modeling (Du et al., 2020; Liu et al., 2022; Du & Kaelbling, 2024; Wang et al., 2024a; Cao et al., 2025), we opt for a more practical approach of modeling multiple marginal

---

[*]These authors contributed equally.
[†]This work was done at National University of Singapore.

distributions independently — each cheaper and simpler to train — and to couple them during *test-time* in a sensible way to obtain the required joint distribution. Unfortunately, such test-time coupling alone does not efficiently *guarantee* adherence to task-specific *hard* constraints such as safety protocols and physical limits. To address such limitation, we propose extending standard Langevin dynamics by combining projection methods with coupled dynamics. Our method cleanly integrates *multiple* pretrained diffusion models through a coupling cost while explicitly incorporating a projection step to ensure strict adherence to task-specific constraints throughout joint sampling.

**Contributions.** We propose *Projected Coupled Diffusion* (PCD), a novel test-time framework unifying both coupled generation leveraging multiple pre-trained diffusion models and projection-based generation to enforce hard constraints only specified at inference. PCD suitably generalizes some conditional sampling techniques including classifier guidance. We show empirically that PCD with both a coupled cost and projection is superior in *jointly* generating highly *correlated* samples with *hard constraints* compared to alternatives with the absence of either or both components.

## 2 RELATED WORK

**Diffusion Models and Guidance** Diffusion models conceptualize generation as a progressive denoising process, *i.e.*, Denoising Diffusion Probabilistic Models (DDPM) (Ho et al., 2020), or equivalently, as a gradient-descent-like procedure that leverages the score of the data distribution within a Langevin dynamics framework (Welling & Teh, 2011; Song & Ermon, 2019; Song et al., 2021b). Improving DDPM, Song et al. (2021a) introduced DDIM to accelerate sampling, and Karras et al. (2022) systematically clarified key design choices for practitioners. Guidance mechanisms form an important class of conditioning techniques for diffusion sampling. Dhariwal & Nichol (2021) first introduced *classifier guidance* (CG) to steer pretrained diffusion models at inference time without retraining, while Ho & Salimans (2022) proposed *classifier-free guidance* by integrating conditioning signals directly during training. Building upon CG, subsequent work has extended guidance beyond classifiers to include analytic functions (Guo et al., 2024; Lee et al., 2025) and property predictors (Meng & Fan, 2024; Feng et al., 2024) in tasks beyond image generation.

**Constrained Diffusion** To address requirements of constraints in real-world tasks, researchers resort to constraint-guided diffusion generation (Yang et al., 2023; Kondo et al., 2024; Feng et al., 2024). However, such paradigm falls short of *enforcing* constraint satisfaction. This incentives the introduction of projection at each step of diffusion (Bubeck et al., 2015; Christopher et al., 2024; Liang et al., 2025). A primal-dual LMC method by Chamon et al. (2024) handles both inequality and equality constraints, and Zampini et al. (2025) proposed a Lagrangian relaxation of projection in the latent space. Lou & Ermon (2023) proposed Reflected diffusion for hypercube-constrained data while Metropolis sampling (Fishman et al., 2023b), log-barrier and reflected dynamics (Fishman et al., 2023a) were also introduced to Riemannian diffusion models.

**Diffusion for Joint Generation** Previous studies in joint generation using multiple diffusion models primarily targeted multimodal generation. Bar-Tal et al. (2023) and Lee et al. (2023) demonstrated panoramic image synthesis by synchronizing several image diffusion models. Xing et al. (2024) and Hayakawa et al. (2025) demonstrated synchronized audio–video generation, and Tang et al. (2023) introduced a framework for generating and conditioning content across combinations of a set of modalities. Joint high- and low-level robot planning is also studied in (Hao et al., 2025).

## 3 PRELIMINARIES

**Notation.** Denote by $\mathbb{Z}^+$ the set of all positive integers, $\|\cdot\|$ the Euclidean norm and $\|\cdot\|_F$ the Frobenius norm. Let $X \in \mathbb{R}^{D_x}$ and $Y \in \mathbb{R}^{D_y}$ be random variables where $D_x, D_y \in \mathbb{Z}^+$ are their dimensionality, respectively. We denote $p_X(x)$ as the probability density function of random variable $X$, likewise for $Y$, and may omit the subscript indicating the random variable when it is clear from the context for notational brevity. Let $\mathcal{N}(\mu, \Sigma)$ be a normal distribution with mean $\mu$ and covariance $\Sigma$. Denote $\Pi_{\mathcal{K}_X}(x) : \mathbb{R}^{D_x} \to \mathbb{R}^{D_x}$ as a projection onto a *nonempty* set $\mathcal{K}_X \subseteq \mathbb{R}^{D_x}$: $\Pi_{\mathcal{K}_X}(x) \triangleq \arg\min_{z \in \mathcal{K}_X} \|z - x\|$ [1].

---

[1] We break ties arbitrarily if $\arg\min_{z \in \mathcal{K}_X} \|z - x\|$ is not unique. Uniqueness always holds for convex $\mathcal{K}_X$.

**Diffusion and Score-based Generative Models.** We examine diffusion models' inference from the perspective of Langevin dynamics. Let $E_X(x) : \mathbb{R}^{D_x} \to \mathbb{R}$ be a continuously differentiable energy function with Lipschitz-continuous gradients and $Z = \int_{\mathbb{R}^{D_x}} \exp\left(-E_X(x)\right) \mathrm{d}x < \infty$. This energy function defines a probability density $p_X(x) = 1/Z \cdot \exp\left(-E_X(x)\right)$. To sample from $p_X(x)$, one may leverage Langevin Monte Carlo (LMC) (Roberts & Tweedie, 1996; Welling & Teh, 2011). Given an initial sample from a prior distribution $X_0 \sim p'_X(x)$ and a fixed time step size $\delta \in \mathbb{R}^+$, the LMC iterates are as follows:

$$X_{t+1} = X_t + \delta \nabla_x \log p_X(X_t) + \epsilon_t \tag{1}$$

where $\epsilon_t \in \mathcal{N}(0, 2\delta \boldsymbol{I})$ and $\nabla_x \log p_X(x)$ is called the *(Stein) score function* of $p_X(x)$. When $\delta \to 0$ and $T \to \infty$, the distribution of $X_T$ converges to $p_X(x)$ under some regularity conditions (Welling & Teh, 2011; Song & Ermon, 2019). In practice, the analytic form of $E_X(x)$ is not accessible. Instead, the score $\nabla_x \log p_X(x)$ or equivalently the gradient $-\nabla E_X(x)$ is approximated by a neural network parameterized by $\theta$ and trained via denoising score matching (Song & Ermon, 2019): $s_X^\theta(x, t) \approx \nabla_x \log p_{X,t}(x)$.

**Classifier guidance.** Classifier guidance (CG) is a "soft" way to steer the diffusion sampling process toward desired distributions. Given a desired attribute $y_0$ as (a constant) condition, the objective of CG is to sample from a target conditional distribution $p_{X|Y}(x \mid y = y_0)$. CG achieves this by perturbing the original learned score with a likelihood term to obtain the posterior score:

$$\nabla_x \log p_{X|Y}(x \mid y = y_0) = \nabla_x \log p_X(x) + \nabla_x \log p_{Y|X}(y = y_0 \mid x),$$

where $\nabla_x \log p_X(x)$ is approximated by the trained score model $s_X^\theta$, and the likelihood $p_{Y|X}(y \mid x)$ can be modeled by a classifier, predictor, or a differentiable function.

**Projected diffusion.** To *enforce* the hard constraint $X \in \mathcal{K}_X$, projected LMC performs a projection operation at every diffusion time step:

$$X_{t+1} = \prod_{\mathcal{K}_X} \left(X_t + \delta \nabla_x \log p_X(x) + \epsilon_t\right) \tag{2}$$

with $\epsilon_t \sim \mathcal{N}(0, 2\delta I)$. The convergence properties of Eq. (2) is analyzed by Bubeck et al. (2015) when the constraint set $\mathcal{K}_X$ is convex and the distribution $p_X(x)$ is log-concave.

# 4 PROJECTED COUPLED DIFFUSION

We study the problem of generating *correlated* samples $(X, Y)$ subject to the *test-time constraints* of $X \in \mathcal{K}_X$ and $Y \in \mathcal{K}_Y$, with two pre-trained scores or diffusion models $s_X^\theta(x, t)$ and $s_Y^\phi(y, t)$, parameterized by $\theta$ and $\phi$, respectively, *without retraining* either of them.

**Coupled Dynamics through Costs.** To facilitate correlation between the generated $X$ and $Y$, we propose using a cost function to couple their diffusion dynamics, *i.e.*, Eq. (1) for $X$ and likewise for $Y$. Let the cost function $c(x, y) : \mathbb{R}^{D_x} \times \mathbb{R}^{D_y} \to \mathbb{R}$ be continuously differentiable.[2] Then the coupled joint dynamics of $(X, Y)$ are

$$X_{t+1} = X_t - \gamma \delta \nabla_x c(X_t, Y_t) + \delta s_X^\theta(X_t, t) + \epsilon_{X,t} \tag{3a}$$

$$Y_{t+1} = Y_t - \gamma \delta \nabla_y c(X_t, Y_t) + \delta s_Y^\phi(Y_t, t) + \epsilon_{Y,t} \tag{3b}$$

where $\gamma \in \mathbb{R}^+$ is a coupling strength parameter, $\epsilon_{X,t} \in \mathcal{N}(0, 2\delta \boldsymbol{I}_{D_x}), \epsilon_{Y,t} \in \mathcal{N}(0, 2\delta \boldsymbol{I}_{D_y})$ are i.i.d. Gaussian noise drawn at each diffusion time step. The cost function $c(x, y)$ can either be an analytic function or a neural network, *e.g.*, a trained classifier or a regression model, and we demonstrate the use of both in our experiments.

As an extension, for each cost function instance $c(x, y)$, we can derive its posterior sampling (PS) variant $c_{\mathrm{PS}}(x, y, t)$ with the DPS method (Chung et al., 2023). Concretely, by Tweedie's formula (Efron, 2011) we may obtain a point estimate for each variable's denoised version through the trained scores, and then compute the cost with those estimates:

$$c_{\mathrm{PS}}(x, y, t) = c\left(X_t + \sigma_{X,t}^2 s_X^\theta(X_t, t), Y_t + \sigma_{Y,t}^2 s_Y^\phi(Y_t, t)\right),$$

where $\sigma_{X,t}^2$ and $\sigma_{Y,t}^2$ are the noise levels at time step $t$ associated with the score models.

---

[2] In practice, this can be relaxed to cost functions that possess subgradients.

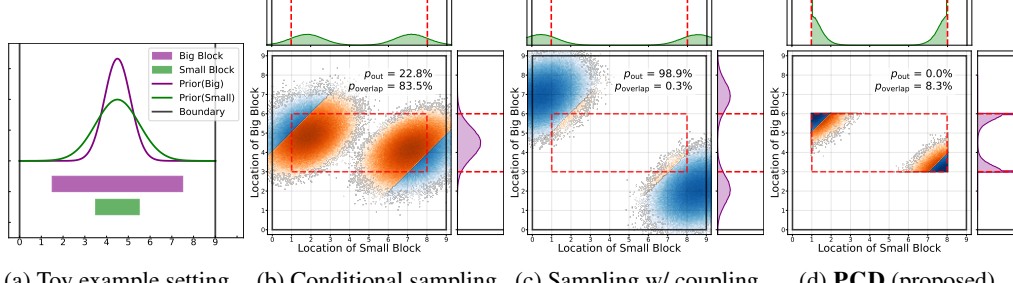

(a) Toy example setting    (b) Conditional sampling    (c) Sampling w/ coupling    (d) **PCD** (proposed)

Figure 1: Toy example of fitting two blocks with different sizes into a narrow corridor. (a) Setting: the bigger block (purple) and small block (green) are with lengths 6 and 2. Both score functions are those of two Gaussians centered at the middle of the red corridor of length 9. (b) Naive conditional approach: the big block occupies the center and the small block is highly possible to overlap if to stay within the corridor (orange for probability mass of overlapping positions, blue for that of non-overlapping positions). (c) The coupled dynamics can place both blocks at different sides with much lower overlapping probabilities, but still can go out of the corridor. (d) PCD produces solutions with low overlapping probabilities and *guarantees* both blocks stay within the corridor.

*Remark* 1. The PS variant does not exactly match the description of a cost function in Eq. (3) due to the extra dependence on diffusion time step $t$. Yet empirically we find them performing well in our proposed framework.

**Projected Coupled Diffusion.** On top of promoting correlations between the generated samples, we also aim to *enforce* the constraints given only *at test-time*. As such, we propose to join the coupled dynamics and projection, resulting in the *Projected Coupled Diffusion* (PCD)[3]:

$$X_{t+1} = \prod_{\mathcal{K}_X} \left( X_t - \gamma\delta\nabla_x c(X_t, Y_t) + \delta s_X^\theta(X_t, t) + \epsilon_{X,t} \right) \tag{4a}$$

$$Y_{t+1} = \prod_{\mathcal{K}_Y} \left( Y_t - \gamma\delta\nabla_y c(X_t, Y_t) + \delta s_Y^\phi(Y_t, t) + \epsilon_{Y,t} \right) \tag{4b}$$

where $\delta \in \mathbb{R}^+$ is the LMC step size parameter, $\gamma \in \mathbb{R}^+$ is the coupling strength parameter, $\epsilon_{X,t} \sim \mathcal{N}(0, 2\delta I_{D_x})$, $\epsilon_{Y,t} \sim \mathcal{N}(0, 2\delta I_{D_y})$ are i.i.d. noise drawn per step and $X_0 \sim \mathcal{N}(0, I_{D_x})$, $Y_0 \sim \mathcal{N}(0, I_{D_y})$. Generation algorithms of our method adopting LMC or DDPM are in Appendix **A**.

We illustrate the importance of *simultaneous* coupling and projection through a toy example (Figure 1a). Two 1D blocks of different lengths must fit within a corridor and avoid overlapping as far as possible. Each of the block's center (denoted by $X$ and $Y$) is generated with a score model that has learned a Gaussian distribution centered at the midpoint of the corridor. A naive approach is to first generate $X$ with only its learned score, and then generate $Y$ *conditioned on* $X$ via classifier guidance. However, this can lead to samples with poor *mutual* correlations, *e.g.*, the first generated $X$ occupies the center of the corridor regardless of $Y$, leaving not enough room to fit both (Figure 1b)[4]. In contrast, coupled dynamics incorporates mutual influence into the generation process of both variables via the cost function, resulting in a much lower overlap probability, yet could violate the hard corridor constraint (Figure 1c). Our proposed method, PCD, combines coupled dynamics and projection (Figure 1d), avoiding overlaps while enforcing the corridor constraint.

### RELATIONSHIP TO OTHER METHODS

**Classifier Guidance.** Interestingly, our framework can encompass the prevailing technique of (CG) as a special case. If we (i) set the cost function as $c(x, y) \propto -\log p_{Y|X}(y \mid x)$, assuming

---

[3]While the assumptions in (Bubeck et al., 2015) do not directly apply due to the non-convex (non-concave) nature of the neural networks used for score approximation, we observe that PCD performs well in practice with reasonable parameters.

[4]If the small block is placed first, it is highly likely to become even *infeasible* to position the big block without exceeding the corridor or overlapping.

a continuously differentiable density $p_{Y|X}$ exists and the (approximated) gradient $\nabla_x p_{Y|X}(y \mid x)$ is accessible, (ii) let $\mathcal{K}_Y$ be a singleton only containing the constant condition $\mathcal{K}_Y = \{y_0\}$, and (iii) $\mathcal{K}_X = \mathbb{R}^{D_x}$, then PCD reduces to

$$X_{t+1} = X_t + \delta \nabla_x \log\left[(p_{Y|X}(y_0 \mid X_t))^\gamma p_X(X_t)\right] + \epsilon_t, \quad \epsilon_t \sim \mathcal{N}(0, 2\delta \boldsymbol{I}) \tag{5}$$

where $\gamma$ becomes a temperature for the likelihood and trivially $Y_t = y_0$; when $\gamma = 1$, the gradient term fully recovers the score of the posterior distribution $p_{X|Y}(x \mid y = y_0)$. In that regard, CG can be seen as PCD with one variable fixed and projection removed in the other. We numerically demonstrate this point on the toy example in Figure 1; see Appendix **C.1** for details.

**Projected Diffusion.** We can consider projected diffusion as PCD without coupling. In PDM (Christopher et al., 2024), which only concerns a single variable $X$, the LMC dynamics of $X$ is projected onto a nonempty but not necessarily convex set $\mathcal{C} \subset \mathbb{R}^{D_x}$. This work fits into PCD in the sense of (i) $\mathcal{K}_X = \mathcal{C}$, $\mathcal{K}_Y = \{y_0\}$, and (ii) the cost function $c(x, y) \equiv 0$, preserving the projection and decoupling the dynamics (turning $Y_t$ into a dummy variable). A numerical verification of PD as a special case of PCD on the toy example in Figure 1 is in Appendix **C.1**.

**Compositional Diffusion.** Another notable line of research is in compositional diffusion (Liu et al., 2022; Wang et al., 2024a; Xu et al., 2024; Cao et al., 2025). Similar to ours, this class of methods also aim to "combine" multiple distributions modeled by energy-based models (EBMs), *e.g.*, diffusion or score models. Unlike ours, however, they still focus on a univariate distribution (*c.f.* a joint distribution in ours). These works attempt to sample from a target product distribution $p_X^{\text{prod}}(x) \propto p_X(x) \prod_{i=1}^N \left(p_X^i(x)/p_X(x)\right) \propto \exp(-E_X(x)) \cdot \exp\left(N \cdot E_X(x) - \sum_i^N E_{\theta^i}^i(x)\right)$ where $E_{\theta^i}^i(x)$ are the corresponding energy functions for $i = 1, \ldots, N$. In practice, the gradients of the $N$ energy functions $\nabla E_{\theta^i}^i(x)$ are learned (independently) by $N$ diffusion models $s_X^{\theta^i}$. PCD can conceptually cover those methods by (i) projecting $X_t$ onto $\mathbb{R}^{D_x}$, (ii) projecting $Y_t$ onto a singleton $\{y_0\}$ and rendering it a dummy variable, and (iii) setting the cost function as: $c(x, y) = N \cdot E_X(x) - \sum_i^N E_{\theta^i}^i(x)$, whereas only its partial gradient $\nabla_x c(x, y)$ is being used.

**Joint Diffusion.** Hayakawa et al. (2025) propose to decompose the "joint scores" $\nabla_x \log p(x, y)$ and $\nabla_y \log p(x, y)$ with Bayes rule: $\nabla_{X_t} \log p(X_t, Y_t) = \nabla_{X_t} \log p_X(X_t) + \nabla_{X_t} p_{Y|X}(Y_t|X_t)$ and likewise for $\nabla_{Y_t} \log p(X_t, Y_t)$, and train one single discriminator $\mathcal{D}_\theta(x, y)$ to approximate both conditional scores: $\nabla_{X_t} \log p(Y_t|X_t) \approx \nabla_{X_t} \log \frac{\mathcal{D}_\theta(X_t, Y_t)}{1 - \mathcal{D}_\theta(X_t, Y_t)}$, likewise for $\nabla_{Y_t} \log p(X_t|Y_t)$. PCD can cover this by setting the constraint sets to $\mathcal{K}_X = \mathbb{R}^{D_x}$, $\mathcal{K}_Y = \mathbb{R}^{D_y}$, and the cost to $c(x, y) = -\log \frac{\mathcal{D}_\theta(x, y)}{1 - \mathcal{D}_\theta(x, y)}$.

## 5 EXPERIMENTS

We seek to address the following research question:

> *How effective is our proposed PCD method in terms of jointly generating correlated samples with test-time constraints compared to generation only with projection, coupling costs, or neither?*

We also explore via ablations the tradeoffs between coupling and trained distribution adherence.

### 5.1 CONSTRAINED MULTI-ROBOT NAVIGATION

We demonstrate PCD in constrained multi-robot motion planning tasks (Shaoul et al., 2025) and its extension to more than two variables[5]. Given a start and goal location for each robot, our objective is to use pretrained diffusion models to generate 2D path trajectories that: (i) avoid collisions with static obstacles *and* any other robots, (ii) respect hard velocity limits specified *at test time*, and (iii) exhibit specific motion patterns dictated by the environment. See Appendix **B.2** for task details.

---

[5]Instead of aiming for state-of-the-art performance, these experiments demonstrate that a simple method like PCD can achieve strong results without intricate domain-specific extensions as MMD(Shaoul et al., 2025).

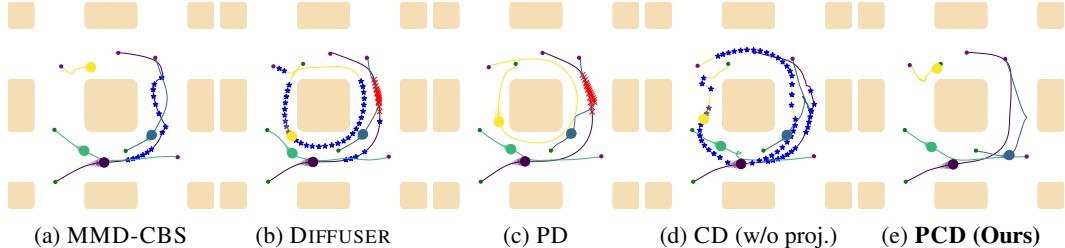

Figure 2: Robot trajectories in Highways with $N = 4$ robots generated by the compared methods. Red crosses mark collisions; blue stars mark velocity constraint violations. The desired motion pattern is to circle the central obstacle *counterclockwise*. (a) MMD-CBS excels in collision avoidance but cannot guarantee velocity constraint satisfaction. (b) Vanilla DIFFUSER *fails* to generate both collision-free and velocity constraint-compliant trajectories. (c) Projection only enforces velocity constraint but ignores collision avoidance. (d) A coupling cost facilitates inter-robot collision avoidance but cannot guarantee velocity constraint. (e) Our method can effectively generate collision-free trajectories *and enforce* the velocity constraint. More qualitative results are in Appendix **C.3**.

**Projection.** We use projection to *enforce* max velocity constraints on each robot. Denote the velocity limit as $v_{\max}$, the (physical) time step size as $\Delta t$, and trajectory horizon as $H$. The projection can be written as an optimization of which the feasible set is our constraint set $\mathcal{K}_X$:

$$\min_{X \in \mathbb{R}^{H \times 2}} \|X - \widehat{X}\|_F^2 \tag{6a}$$

$$\text{s.t.} \quad \|x_0 - X_1\| \leq v_{\max} \Delta t, \tag{6b}$$

$$\|X_h - X_{h-1}\| \leq v_{\max} \Delta t, \quad h = 2, \ldots, H, \tag{6c}$$

where $\widehat{X}$ is the diffusion-predicted trajectory for one robot in matrix form, $X_h \in \mathbb{R}^2$ is the position vectors at (physical) discrete time step $h$ and $x_0 \in \mathbb{R}^2$ is a known starting position. This *convex* optimization problem can be efficiently solved in parallel using the Alternating Direction Method of Multipliers (Boyd et al., 2011; Parikh et al., 2014). Detailed derivations and algorithm are in Appendix **B.2.2**.

**Coupling Cost.** We aim to avoid both collisions among robots and collisions with known static obstacles via coupling. Prior work (Carvalho et al., 2023; Shaoul et al., 2025) achieves static obstacle avoidance by CG, which we have shown is a special case of PCD coupling. Thus, our coupling cost function is a linear combination of a robot-collision and obstacle-collision costs:

$$c(X, Y) = \lambda_{\text{robo}} c_{\text{robo}}(X, Y) + \lambda_{\text{obst}} c_{\text{obst}}(X, Y),$$

where $X, Y \in \mathbb{R}^{H \times 2}$ are trajectories of 2 robots' positions. We experiment with two robot collision cost functions, (i) a log-barrier (LB) cost

$$c_{\text{LB}}(X, Y) = -\sum_{h=1}^{H} \log \left( \|X_h - Y_h\| + \alpha \right) \tag{7}$$

where $\alpha > 0$ is a parameter, and (ii) a "squared hinge distance" (SHD) cost:

$$c_{\text{SHD}}(X, Y) = \sum_{h=1}^{H} \left( \mathbf{1} \left[ \|X_h - Y_h\| \leq r \right] \cdot (r - \|X_h - Y_h\|) \right)^2 \tag{8}$$

where $\mathbf{1}[\cdot]$ is the indicator function and $r > 0$ is the active range parameter. For $N > 2$ robots, $X^1, \ldots, X^N$, we extend both costs to $c_\diamond(X^1, \ldots, X^N) = \sum_{1 \leq i < j \leq N} c_\diamond(X^i, X^j)$ with $\diamond \in \{\text{LB}, \text{SHD}\}$. We follow Carvalho et al. (2023) in designing the obstacle-avoidance cost. See details in Appendix **B.2**.

**Experiment Setup.** We test with 2 and 4 robots on four different tasks from Shaoul et al. (2025). For each task, we use the pretrained models from Shaoul et al. (2025) and choose three $v_{\max}$s.

**Baselines.** We compare our method with a vanilla diffusion model DIFFUSER (Janner et al., 2022) and MMD-CBS (Shaoul et al., 2025). We evaluate each method on 100 trials, each with an initial

| METHOD \Metric | Task **Empty**, 4 Robots, Max Vel. **0.703** | | | | Task **Highways**, 4 Robots, Max Vel. **0.878** | | | |
|---|---|---|---|---|---|---|---|---|
| | SU(%)↑ | RS↑ | *minCS(%)↑ | *minDA↑ | SU(%)↑ | RS↑ | *minCS(%)↑ | *minDA↑ |
| DIFFUSER | 65.0 | 0.616 | 62.3 | **0.990** | 53.0 | 0.208 | 66.7 | 0.979 |
| MMD-CBS | **100** | **1.00** | 11.0 | **0.990** | **100** | **1.00** | 63.0 | 0.960 |
| DIFFUSER + proj. | 65.0 | 0.615 | **100** | **0.990** | 54.0 | 0.214 | **100** | 0.978 |
| CD-LB (w/o proj.) | **100** | 0.993 | 0.0312 | 0.814 | **100** | 0.999 | 0.00 | **0.986** |
| CD-SHD (w/o proj.) | **100** | **1.00** | 35.3 | **0.990** | **100** | **1.00** | 34.6 | 0.976 |
| PCD-LB | 96.0 | 0.916 | 100 | 0.489 | **100** | 0.950 | 100 | 0.957 |
| PCD-SHD | **100** | 0.993 | **100** | 0.960 | **100** | 0.996 | **100** | 0.963 |

Table 1: Performance comparison with $N = 4$ robots on two out of four tasks. Left: `Empty`, $v_{max} = 0.703$; Right: `Highways`, $v_{max} = 0.878$. *For CS and DA, which should have been $N$-tuples, we report here the *minimum of them* due to space limit. See additional results in Appendix **C.3**.

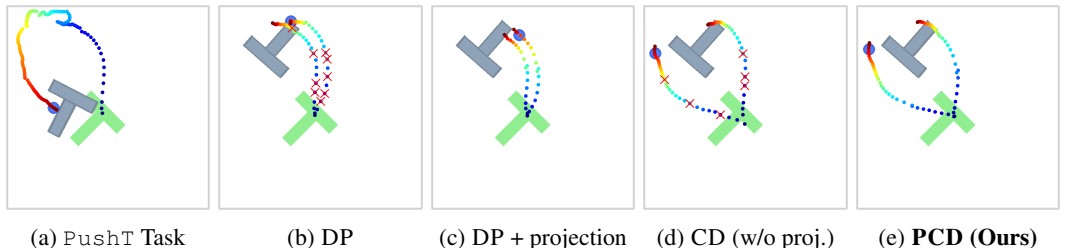

| (a) `PushT` Task | (b) DP | (c) DP + projection | (d) CD (w/o proj.) | (e) **PCD (Ours)** |

Figure 3: (a) The `PushT` task: One single robot (blue circle) pushes a T block (gray) to a target pose (green) given different initial positions of the block and robot. Robot trajectories are plotted with a colormap (warmer colors indicate later time steps), with red crosses marking velocity violations. Only the first few dozen steps are shown for visual clarity. (b) Vanilla DP *fails* to generate trajectory pairs, each in a distinct mode, and adhering to velocity limits. (c) Projection enforces velocity limits but cannot "split" trajectories apart. (d) A coupling cost encourages non-intersecting trajectories but does not strictly enforce velocity constraints. (e) Our method generates non-intersecting trajectories *and* ensures strict adherence to velocity constraints.

configuration (start and goal locations for each robot) sampled uniformly at random by rejection sampling. Except for MMD-CBS, we generate 128 i.i.d. samples; for MMD-CBS[6], we also set its diffusion sampling batch size as 128. We run 25 diffusion inference steps for all methods.

**Evaluation Metrics.** We evaluate performance of the methods in terms of task completion or adherence to original data distribution, constraint satisfaction, and inter-robot collision avoidance. We adopt *success rate* (SU) and *data adherence* (DA) from (Shaoul et al., 2025) to evaluate task completion. SU is the average, over all initial configurations, of an indicator for whether at least one trajectory in the batch completes the task without collision. *Constraint satisfaction* (CS) is an indicator of whether a trajectory satisfies the velocity constraint at all time steps. *Inter-robot safety* (RS) is an indicator of whether a trajectory tuple is inter-robot collision-free. All metrics except SU are reported as empirical means over a batch of i.i.d. samples.

**Results.** Figure 2 shows sample trajectories from compared methods. Table 1 summarizes quantitative results for two environments with 4 robots under one velocity constraint. All constraint-agnostic methods (without projection) achieve low CS rates. MMD-CBS unsurprisingly achieves perfect SU score by selecting and stitching a single optimal trajectory tuple. PCD- and CD- methods approach this upper bound, while vanilla DIFFUSER and its projected variant lag behind. PCD-SHD shows slightly reduced data adherence, while the LB cost more aggressively trades data adherence for collision avoidance due to its steep gradients. Similar patterns are observed in the other tasks. Overall, PCD effectively promotes inter-robot safety through coupling while enforcing hard test-time velocity constraints, with a tradeoff between coupling strength and data adherence depending on the cost function. See additional results with ablations, and runtime in Appendix **C.3** and **C.2**.

---

[6]See a detailed discussion of difference between PCD and MMD-CBS in *Remark 4*, Appendix **C.3**.

## 5.2 Constrained and Diverse Robot Manipulation

We evaluate our method on the `PushT` task (Florence et al., 2022; Chi et al., 2023). As shown in Figure 3a, a diffusion model is trained to generate trajectories for a robot to push the gray `T`-shaped block from different starting positions till it matches the green target pose. Our objective is to utilize such pretrained models to generate a *pair* of distinct trajectories strictly satisfying a *maximum velocity constraint* imposed *at test time* and do not intersect as far as possible[7].

**Projection & Coupling Cost.** As in multi-robot experiment, we enforce velocity limits via projection, using the formulation in Eq. (6). We experimented with two cost functions for encouraging trajectories from a pair to stay away from each other. The first cost builds upon the Determinantal Point Process (DPP) guidance (Feng et al., 2025) designed for promoting trajectory diversity:

$$c_{\text{DPP}}(X, Y) = \log \left( \cos \angle(\tilde{X}, \tilde{Y}) + \varepsilon \right) \tag{9}$$

where $\tilde{X}, \tilde{Y} \in \mathbb{R}^{2H}$ are the flattened vectors of the trajectories, and $\varepsilon > 1$ is a constant. The other cost is the log-barrier cost Eq. (7). For both costs, we also devise their posterior sampling variants.

**Setup & Evaluation Metrics.** We adopt DIFFUSION POLICY (DP) by Chi et al. (2023) as our base algorithm, using pretrained weights from Feng et al. (2024)[8]. Compared methods are vanilla DP, DP with only projection, DP with only coupling and PCD. We evaluate each method on 50 uniformly random initial conditions. With each method, we generate 100 *pairs* of full trajectories, under three different max velocity limits. We use 32 diffusion steps at inference, with other settings recommended by Chi et al. (2023). **Metrics:** Four quantitative metrics are evaluated: Dynamic Time Warping (DTW) (Berndt & Clifford, 1994; Müller, 2007), discrete Fréchet distance (DFD) (Alt & Godau, 1995), velocity constraint satisfaction rate (CS), and task completion score (TC) (Florence et al., 2022; Chi et al., 2023). DTW and DFD quantify dissimilarity between two trajectories. CS evaluates fraction of trajectories satisfying the velocity constraint. TC measures how well the block-pushing task is accomplished, where 1.0 is the best and 0 the worst. We report all metrics by their empirical means over all initial staring locations and trajectory pairs. Details are in Appendix **B.3**.

**Results.** We report results in Figure 3b–3e and Table 2. From Table 2, we see all projection-involved methods achieve perfect velocity constraint satisfaction, outperforming both the baseline and coupling-only (CD) approaches. CD and our PCD methods consistently produce higher DTW and DFD than baseline with or without projection, suggesting that coupling effectively discourages intersecting trajectories. In terms of task completion, all methods except the baseline show degraded performance, likely due to the velocity limit enforced by projection. These results show that our framework can enforce test-time velocity constraints and spatially separate generated trajectories without significantly sacrificing data adherence. Without projection, both PS variants of the coupling costs exhibit higher DTW and DFD than their respective non-PS version, and better preserves the task completion with projection. Full results and additional ablation study are in Appendix **C.4**.

## 5.3 Constrained Coupled Image Pair Generation

We demonstrate a *toy example* of paired face generation using two latent diffusion models (LDMs) (Rombach et al., 2022) (Figure 4). Each generated pair must (i) satisfy *gender and facial attribute constraints*, and (ii) exhibit a clear *contrast between age groups*. We enforce (i) via projection and promote (ii) through a classifier-driven coupling cost.

**Projection.** For each LDM, we generate two structurally similar exemplar images of one individual at different ages (Figure 4a) using ChatGPT (OpenAI, 2023), encode them via the model's VAE, and form convex hulls as feasible latent regions. At each diffusion step, we use mirror descent (Nemirovsky & Yudin, 1983; Beck & Teboulle, 2003) to project intermediate latents onto these hulls, enforcing strict *gender and facial attribute constraints* (see Appendix **B.4** for details).

**Coupling Cost.** To induce an *age-group contrast*, we use a latent classifier that classifies age groups Y (young, $< 50$) and O (old, $\geq 50$). Our coupling cost is defined as $c_{\text{XOR}}(x, y) = -\sum_{a \in \{Y, O\}} \text{XOR}(x, y)$, where $\text{XOR}(x, y) = p(a|x)\big(1 - p(a|y)\big) + p(a|y)\big(1 - p(a|x)\big)$ and $p(a|\cdot)$ are the probabilities of a sample belonging to age-class $a \in \{Y, O\}$, obtained from the classifier.

---

[7]Note that the two trajectories in a pair *are not* pushing the block together at the same time.

[8]The model from (Feng et al., 2024) was trained on an augmented dataset with broader coverage than that provided by Chi et al. (2023), yielding more diverse and feasible trajectories.

| METHOD | DTW↑ | DFD↑ | CS(%)↑ | TC↑ |
|---|---|---|---|---|
| DP | 3.2 | .47 | (65, 65) | (**.93**, **.93**) |
| DP + proj. | 3.0 | .43 | (**100**, **100**) | (.90, .89) |
| CD-DPP | 3.7 | .54 | (63, 62) | (.92, .92) |
| CD-DPP-PS | 4.5 | .65 | (57, 57) | (.91, .92) |
| CD-LB | 4.1 | .60 | (59, 59) | (.91, .91) |
| CD-LB-PS | 4.5 | .65 | (59, 59) | (.92, .93) |
| PCD-DPP | 4.6 | .64 | (**100**, **100**) | (.83, .83) |
| PCD-DPP-PS | 4.4 | .62 | (**100**, **100**) | (.89, .89) |
| PCD-LB | **5.1** | **.71** | (**100**, **100**) | (.78, .79) |
| PCD-LB-PS | 4.4 | .62 | (**100**, **100**) | (.89, .88) |

Table 2: Results of PushT task by all compared methods with velocity limit $v_{max} = 8.4$. CD denotes DP+coupling only; -PS denotes posterior sampling. See full results in appendix Table 14.

| Method | XOR%↑ | M/F%↑ | SE-C*↑ | SE-L*↓ | IS-L*↑ |
|---|---|---|---|---|---|
| SD | 20 | 71/14 | .40/.41 | .77/.76 | **.75**/**.75** |
| CNet† | 8 | **100**/56 | .68/.75 | .48/.50 | .45/.50 |
| SD+P | 44 | **100**/**100** | **.88**/.91 | .15/.16 | .06/.09 |
| Sync† | 48 | 95/17 | .62/.61 | .67/.68 | .70/.69 |
| SD+C | 48 | 51/47 | .45/.46 | .76/.74 | **.75**/**.75** |
| SD+C† | 64 | 47/37 | .55/.60 | .70/.68 | .69/.69 |
| PCD | **96** | **100**/**100** | **.88**/**.92** | **.11**/**.14** | .11/.13 |

Table 3: Paired face-generation results. Baselines: CNet = ControlNet-XS; Sync = SyncDiffusion. Metrics: SE-C* = SE-CLIP; SE-L* = SE-LPIPS; IS-L* = IS-LPIPS. Boldface indicates the best score(s) for each metric. Full results including SE-FID are deferred to Table 15 in appendix.

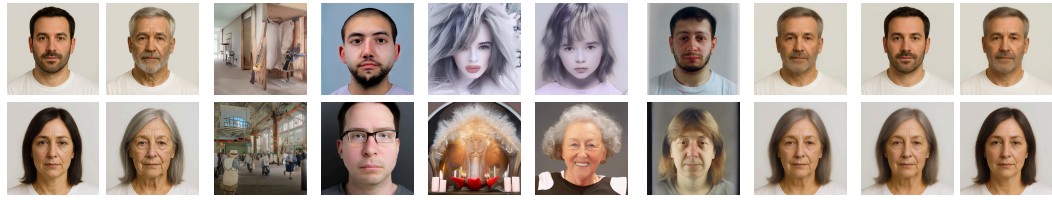

(a) Exemplar Images  (b) SD  (c) Sync†  (d) SD+C  (e) SD+C†  (f) CNet†  (g) SD+P  (h) **PCD (Ours)**

Figure 4: Paired face generation with two Stable Diffusion models (SD) (row-wise). Methods: Sync = SyncDiffusion, CNet = ControlNet-XS, +C = coupling; +P = projection; † = with text prompt. (a) Exemplar images of each model, their latents form the convex hulls. (b) Vanilla SD often fails to produce faces. (c, d) SyncDiffusion and Coupling steers samples toward the target *age-group contrast* but with attribute drift. (e) Adding text prompts to Coupling yields faces, yet violates attribute constraints. (f, g) ControlNet-XS encourages gender and facial attribute alignment; Projection enforces it; but both fail to promote the *age-group contrast*. (h) Our method yields pairs that satisfy the gender and facial-attribute constraints and simultaneous promote sheer *age-group contrast*.

**Setup & Metrics.** We employ STABLE DIFFUSION V2.1-BASE (SD-2.1) (Stability AI, 2022) as our base model and disable classifier-free guidance during sampling to achieve unconditional generation unless specified otherwise. For coupling, we train a *latent classifier* for the model using the FFHQ-Aging Dataset (Or-El et al., 2020). **Baselines**: We compare our method with vanilla SD-2.1, SD-2.1 with only projection, SD-2.1 with only coupling, and SD-2.1 with coupling *and* a generic text prompt. We also compare against SyncDiffusion (Lee et al., 2023) and ControlNet-XS (Zavadski et al., 2024)[9] for reference[10]. We use 100 diffusion steps and generate 25 *pairs* of samples using each method. See setup details in Appendix **B.4**. **Metrics:** To evaluate projection, we report five metrics: (i) *Gender constraint satisfaction rate* (M/F); (ii) *Sample-Exemplar CLIP similarity* (SE-CLIP) (Radford et al., 2021); (iii) *Sample-Exemplar LPIPS* (SE-LPIPS) (Zhang et al., 2018); (iv) *Sample-Exemplar FID* (SE-FID) (Heusel et al., 2017); (v) *Intra-Sample LPIPS* (IS-LPIPS). SE-CLIP, SE-LPIPS and SE-FID serve as proxies for adherence to exemplar-specified facial attribute constraints (noting that satisfaction is guaranteed by design of our projection operator), while IS-LPIPS quantifies diversity across generated samples. To evaluate coupling, we measure the *age-group contrast satisfaction rate* (XOR) with another age-group *image classifier* trained on the FFHQ-Aging Dataset. We average XOR and M/F over generated pairs, SE-CLIP and SE-LPIPS

---

[9]Note that ControlNet-XS requires additional training and we use pretrained weights released by Zavadski et al. (2024). See details in Appendix **B.4**.

[10]SyncDiffusion and ControlNet-XS are included *solely for reference*. We are *not* claiming state-of-the-art performance. Instead, we are demonstrating potential uses of PCD in image-related generation tasks.

over sample–exemplar pairs, IS-LPIPS over intra-model sample pairs. SE-FID is computed as a single scalar between the exemplar set and generated samples set of each model.

**Results.** We report results in Figures 4b–4h and Table 3. Projection-based methods (SD+P, PCD) attain 100% gender satisfaction (M/F) and the strongest alignment to exemplars (higher SE-CLIP and lower SE-LPIPS) compared to vanilla SD, coupling-only variants and other baselines. Projection reduces diversity (low IS-LPIPS); adding coupling partially recovers diversity. Coupling-based methods improve age-group XOR satisfaction, with PCD attaining the highest rate (96%). SyncDiffusion matches SD+C on XOR but shows slightly reduced sample variation (lower IS-LPIPS). ControlNet-XS exhibits a male-bias, producing more male samples and thus yielding 100% male gender satisfaction (M). See additional results with larger exemplar sets and ablations in Appendix **C.6**, and runtime profile in Appendix **C.2**.

## 6    LIMITATIONS AND DISCUSSIONS

First, the per-step projection operations in PCD introduce computational overhead compared to vanilla or standard guided diffusion methods. In practice, however, this overhead can be mitigated through several simple strategies: (i) reduced projection frequency: projections need not be performed at every diffusion step but can instead be applied every $K > 1$ diffusion steps (including the final step to enforce constraints); (ii) approximated projections: limit the number of optimization iterations within each projection operation to achieve *near-convergence*, and only run projections to full convergence every $K$ diffusion steps (including the final step); (iii) warm-starting: initialize the ADMM primal and dual variables from the previous diffusion step. These strategies can effectively reduce runtime while preserving the core benefits of PCD.

Second, PCD assumes differentiability or at least the existence of subgradients for the cost functions as many other approaches (Dhariwal & Nichol, 2021; Nichol et al., 2022; Chung et al., 2022; 2023; Carvalho et al., 2023; Römer et al., 2025). For various practical applications, differentiable approximations are available for reasonable yet non-differentiable and discontinuous costs such as temporal logic specifications (Feng et al., 2024; Meng & Fan, 2024). Moreover, alternatives that do not require gradients such as Monte Carlo based importance sampling could be leveraged for approximation (Dou & Song, 2024; Phillips et al., 2024; Jung et al., 2025; Li et al., 2025).

Third, PCD assumes that the feasible region of the test-time constraints has an overlap with the support of the distribution learned by the pretrained model. This issue becomes critical if the feasible solutions differ substantially from the training data. For example, restricting end-effector velocity to only 1% of its original average in the `PushT` task would cause PCD to generate trajectories failing the manipulation task at most times.

Finally, PCD introduces a coupling strength hyperparameter, and practitioners may need to tune it for desired outcomes. Degraded data fidelity is observed if the coupling strength is set too high, indicating a trade-off between data fidelity and desired correlations between variables. Recent efforts (Jung et al., 2025) in promoting compatibility between test-time objectives and pretrained diffusion's data fidelity point a promising direction to addressing this issue.

## 7    CONCLUSION

We introduced Projected Coupled Diffusion (PCD), a test-time framework for joint generation with multiple diffusion models under hard constraints. Our method combines coupled dynamics and projection operation, generalizing existing techniques like classifier guidance and projection-based diffusion inference without requiring model retraining. Experiments on image-pair generation, object manipulation, and multi-robot motion planning show that PCD effectively facilitates mutual correlation and provides guaranteed test-time hard constraint satisfaction. Future work includes exploring more sophisticated and non-differentiable coupled costs and non-convex constraints.

### REPRODUCIBILITY STATEMENT

Implementation code is available at `https://github.com/EdmundLuan/pcd`.

LARGE LANGUAGE MODEL USAGE DECLARATION

As stated in Section 5.3, the images used as exemplars for the paired-faces experiments were generated by ChatGPT (OpenAI, 2023). We also used ChatGPT (OpenAI, 2023) to perform grammar check and minor language polishing for the first three paragraphs of Section 1 with further human edition.

ACKNOWLEDGMENTS

This research/project is supported by the National Research Foundation, Singapore under its National Large Language Models Funding Initiative, (AISG Award No: AISG-NMLP-2024-002), and the National University of Singapore, under the Start-Up Grant Scheme. Any opinions, findings and conclusions or recommendations expressed in this material are those of the author(s) and do not reflect the views of National Research Foundation, Singapore. The authors would like to thank Ziqiao Meng, Jiaying Wu, and Sze Jue Yang for helpful discussions and feedback on early drafts.

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

# A    METHOD DETAILS

Our proposed PCD framework can perform inference with LMC and Denoising Diffusion Probablistic Models (DDPM) (Ho et al., 2020). Moreover, it is also easy to apply Diffusion Posterior Sampling (DPS) (Chung et al., 2023) within our framework. We present here three algorithms under PCD framework: PCD-LMC, PCD-DDPM, and PCD-DPS, detailed in Algorithm 1, Algorithm 2, and Algorithm 3, respectively.

---

**Algorithm 1** Projected Coupled Diffusion with LMC

---

**Require:** Score models $s_X^\theta$, $s_Y^\phi$; projectors $\Pi_{\mathcal{K}_X}$, $\Pi_{\mathcal{K}_Y}$; coupling strength $\gamma$; LMC step size $\delta$; max iteration $T$.
1: $X_0 \sim \mathcal{N}(0, \boldsymbol{I}_{D_x}), Y_0 \sim \mathcal{N}(0, \boldsymbol{I}_{D_y})$       ▷ *Initialize from std. Gaussian*
2: **for** $t = 1$ to $T - 1$ **do**
3:     ▷ *Coupled LMC dynamics*      ◁
4:     $X_{t+1} \leftarrow X_{t+1} - \delta s_X^\theta(X, t) - \gamma\delta\nabla_X c(X_t, Y_t)$
5:     $Y_{t+1} \leftarrow Y_{t+1} - \delta s_Y^\phi(Y, t) - \gamma\delta\nabla_Y c(X_t, Y_t)$
6:     $\epsilon_X \sim \mathcal{N}(0, \boldsymbol{I}_{D_x}), \epsilon_Y \sim \mathcal{N}(0, \boldsymbol{I}_{D_y})$      ▷ *i.i.d. noise*
7:     $X_{t+1} \leftarrow X_{t+1} + \sqrt{2\delta}\epsilon_X$
8:     $Y_{t+1} \leftarrow X_{t+1} + \sqrt{2\delta}\epsilon_Y$
9:     ▷ *Projection step*      ◁
10:    $X_{t+1} \leftarrow \Pi_{\mathcal{K}_X}(X_{t+1})$
11:    $Y_{t+1} \leftarrow \Pi_{\mathcal{K}_Y}(Y_{t+1})$
12: **return** $(X_T, Y_T)$      ▷ *Return joint samples*

---

**Algorithm 2** Projected Coupled Diffusion with DDPM

---

**Require:** Score models $s_X^\theta$, $s_Y^\phi$; projectors $\Pi_{\mathcal{K}_X}$, $\Pi_{\mathcal{K}_Y}$; coupling strength $\gamma$; DDPM noise schedule $\{\alpha_t\}_{t=1}^T$; DDPM inference step $T$.
1: $X_T \sim \mathcal{N}(0, \boldsymbol{I}_{D_x}), Y_T \sim \mathcal{N}(0, \boldsymbol{I}_{D_y})$      ▷ *Initialize from std. Gaussian*
2: **for** $t = T$ to $1$ **do**
3:     ▷ *Normal diffusion*      ◁
4:     $\epsilon_X \sim \mathcal{N}(0, \boldsymbol{I}_{D_x}), \epsilon_Y \sim \mathcal{N}(0, \boldsymbol{I}_{D_y})$
5:     $s_X \leftarrow s_X^\theta(X_t, t), s_Y \leftarrow s_Y^\phi(Y_t, t)$
6:     $X_{t-1} \leftarrow \frac{1}{\sqrt{\alpha_t}}(X_t + (1 - \alpha_t)s_X) + \sqrt{1 - \alpha_t}\epsilon_X$
7:     $Y_{t-1} \leftarrow \frac{1}{\sqrt{\alpha_t}}(Y_t + (1 - \alpha_t)s_Y) + \sqrt{1 - \alpha_t}\epsilon_Y$
8:     ▷ *Coupling step*      ◁
9:     $X_{t-1} \leftarrow X_{t-1} - \gamma\nabla_{X_t} c(X_t, Y_t)$
10:    $Y_{t-1} \leftarrow Y_{t-1} - \gamma\nabla_{Y_t} c(X_t, Y_t)$
11:    ▷ *Projection step*      ◁
12:    $X_{t-1} \leftarrow \Pi_{\mathcal{K}_X}(X_{t-1})$
13:    $Y_{t-1} \leftarrow \Pi_{\mathcal{K}_Y}(Y_{t-1})$
14: **return** $(X_0, Y_0)$      ▷ *Return joint samples*

---

---

**Algorithm 3** Projected Coupled Diffusion with DPS

---

**Require:** Score models $s_X^\theta$, $s_Y^\phi$; projectors $\Pi_{\mathcal{K}_X}$, $\Pi_{\mathcal{K}_Y}$; coupling strength $\gamma$; DDPM noise schedule $\{\alpha_t\}_{t=1}^T$; DDPM inference step $T$.

1: **Pre-compute** $\bar{\alpha}_t = \prod_{\tau=1}^t \alpha_\tau$ for $t = 1, \ldots, T$
2: $X_T, Y_T \sim \mathcal{N}(0, \boldsymbol{I})$     ▷ *Initialize from std. Gaussian*
3: **for** $t = T$ to $1$ **do**
4:     ▷ *Normal diffusion*     ◁
5:     $\epsilon_X \sim \mathcal{N}(0, \boldsymbol{I}_{D_x}), \epsilon_Y \sim \mathcal{N}(0, \boldsymbol{I}_{D_y})$
6:     $s_X \leftarrow s_\theta(X_t, t), s_Y \leftarrow s_\phi(Y_t, t)$
7:     $X_{t-1} \leftarrow \frac{1}{\sqrt{\alpha_t}}(X_t + (1 - \alpha_t)s_X) + \sqrt{1 - \alpha_t}\epsilon_X$
8:     $Y_{t-1} \leftarrow \frac{1}{\sqrt{\alpha_t}}(Y_t + (1 - \alpha_t)s_Y) + \sqrt{1 - \alpha_t}\epsilon_Y$
9:     ▷ *Coupling with posterior sampling*     ◁
10:     $\hat{X}_0 \leftarrow \frac{1}{\sqrt{\bar{\alpha}_t}}(X_t + (1 - \bar{\alpha}_t)s_X)$     ▷ *Tweedie's formula*
11:     $\hat{Y}_0 \leftarrow \frac{1}{\sqrt{\bar{\alpha}_t}}(Y_t + (1 - \bar{\alpha}_t)s_Y)$
12:     $X_{t-1} \leftarrow X_{t-1} - \gamma \nabla_{X_t} c(\hat{X}_0, \hat{Y}_0)$
13:     $Y_{t-1} \leftarrow Y_{t-1} - \gamma \nabla_{Y_t} c(\hat{X}_0, \hat{Y}_0)$
14:     ▷ *Projection step*     ◁
15:     $X_{t-1} \leftarrow \Pi_{\mathcal{K}_X}(X_{t-1})$
16:     $Y_{t-1} \leftarrow \Pi_{\mathcal{K}_Y}(Y_{t-1})$
17: **return** $(X_0, Y_0)$     ▷ *Return joint samples*

---

# B  IMPLEMENTATION DETAILS

## B.1  GENERAL

**Computational Hardware.**  All experiments were run on a workstation with 1 AMD Ryzen Threadripper PRO 5995WX 64-Core CPU, 504 GB RAM, and 2 NVIDIA RTX A6000 GPUs each with 48GB VRAM. For each experiment run, only 1 GPU was utilized.

**Software and Code Bases.**  All experiments were run using PyTorch (Paszke et al., 2019). Image experiments were also run with Diffusers (von Platen et al., 2022). The PushT experiment is adapted from LTLDoG (Feng et al., 2024) and DIFFUSION POLICY (Chi et al., 2023). The multi-robot experiment builds upon MMD (Shaoul et al., 2025).

## B.2  MULTI-ROBOT EXPERIMENT

### B.2.1  SIMULATION TASKS AND ENVIRONMENTS

Four simulated environments used in our experiments are from the MMD benchmark Shaoul et al. (2025), Empty, Highways, Conveyor and DropRegion, as shown in Figure 5. In each environment, given a start and a goal position $(s, g) \in \mathbb{R}^2 \times \mathbb{R}^2$ of a single robot, a diffusion model is trained to generate 2D trajectories $X \in \mathbb{R}^{H \times 2}$, where $H$ is the trajectory length. Each environment comes with a trajectory distribution through a trajectory dataset collected under a specific motion pattern:

- In Empty, the motion pattern is straight-line movements from start to goal.

- In Highways, the pattern is circling around a central block in *counterclockwise* direction when moving from start to goal, resembling a traffic roundabout.

- In Conveyor, the pattern is only moving in one single direction (either left or right) when entering the two corridors in the middle of the map.

- In DropRegion, the pattern is to stay around one of the four "dropping points" in the map for a certain amount of time steps when moving from start to goal, resembling a delivery job.

The task in both environments is to generate collision-free trajectories for $N$ robots, given an initial configuration consisting of start and goal positions for all robots:

$$\mathcal{P} \triangleq \{(s_1, g_1), \ldots, (s_N, g_N)\} \in (\mathcal{W}_{\text{free}} \times \mathcal{W}_{\text{free}})^N,$$

where $s_i$, $g_i$ are the start and goal positions for robot $i$, and $\mathcal{W}_{\text{free}} \subset \mathbb{R}^2$ denotes the obstacle-free environment space (free workspace).

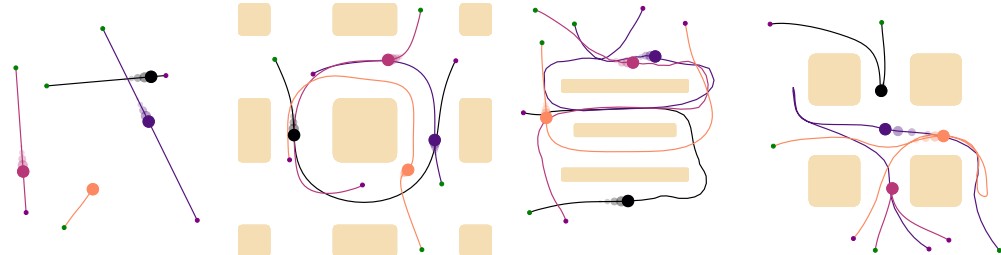

Figure 5: Multi-robot motion planning tasks with 4 robots and simulation environments (from left to right) Empty, Highways, Conveyor, and DropRegion, each with their own motion patterns. Each of the trajectories' colors denotes an individual robot; green and purple small dots represents start and goal positions.

### B.2.2 ADMM-BASED PROJECTION DESIGN

We repeat Eq. (6) here for readers' convenience:

$$\min_{X \in \mathbb{R}^{H \times 2}} \|X - \widehat{X}\|_F^2 \tag{6a}$$

$$\text{s.t.} \quad \|x_0 - X_1\| \le v_{\max} \Delta t, \tag{6b}$$

$$\|X_h - X_{h-1}\| \le v_{\max} \Delta t, \quad h = 2, \dots, H, \tag{6c}$$

where $\widehat{X}$ is the diffusion-predicted trajectory for one robot in matrix form, $X_h \in \mathbb{R}^2$ is the position vector at (physical) discrete time step $h$ and $x_0 \in \mathbb{R}^2$ is a known starting position.

A direct approach for solving the optimization in Eq. (6) is to leverage off-the-shelf solvers to optimize *each trajectory*. However, this incurs significant computation overheads upon large batch of trajectories. Alternatively, we can reformulate Eq. (6) and efficiently solve a batch of such problem instances *in parallel* using Alternating Direction Method of Multipliers (ADMM) (Boyd & Vandenberghe, 2004).

To apply ADMM, we need to introduce auxiliary variables $Z_h \in \mathbb{R}^2$ representing the per-step positional displacements:

$$Z_1 = X_1 - x_0, \tag{10a}$$

$$Z_h = X_h - X_{h-1} \quad \text{for } h = 2, \dots, H. \tag{10b}$$

The constraints Eq. (6b) (6c) then become:

$$\|Z_h\| \le v_{\max} \Delta t, \quad h = 1, \dots, H.$$

Let $Z = [Z_1, \cdots, Z_H]^\top \in \mathbb{R}^{H \times 2}$ and define the constraint set for $Z$:

$$\mathcal{K}_Z = \left\{ Z \in \mathbb{R}^{H \times 2} \mid \|Z_h\| \le v_{\max} \Delta t, h = 1, \dots, H \right\}, \tag{11}$$

together with the indicator function of $\mathcal{K}_Z$:

$$\mathbb{I}_{\mathcal{K}_Z}(Z) = \begin{cases} 0 & \text{if } Z \in \mathcal{K}_Z, \\ \infty & \text{otherwise.} \end{cases} \tag{12}$$

Let $A \in \mathbb{R}^{H \times H}$ be a coefficient matrix

$$A = \begin{bmatrix} 1 & 0 & 0 & \cdots & 0 \\ -1 & 1 & 0 & \cdots & 0 \\ 0 & -1 & 1 & \cdots & 0 \\ \vdots & \vdots & \ddots & \ddots & \vdots \\ 0 & 0 & \cdots & -1 & 1 \end{bmatrix} \tag{13}$$

and define an offset matrix $b \in \mathbb{R}^{H \times 2}$

$$b = \left[ x_0 \; \underbrace{\mathbf{0}_2 \; \cdots \; \mathbf{0}_2}_{H-1} \right]^\top \tag{14}$$

where $\mathbf{0}_2 \in \mathbb{R}^2$ is a zero column vector.

With $X$ and $Z$, optimization problem Eq. (6) can be reformulated in an ADMM fashion as:

$$\min_{X \in \mathbb{R}^{H \times 2}, Z \in \mathbb{R}^{H \times 2}} \|X - \hat{X}\|_F^2 + \mathbb{I}_{\mathcal{K}_Z}(Z) \tag{15a}$$

$$\text{s.t.} \quad AX - Z = b. \tag{15b}$$

The augmented Lagrangian of Eq. (15) is:

$$\mathcal{L}_\xi(X, Z, \Lambda) = \|X - \hat{X}\|_F^2 + \mathbb{I}_{\mathcal{K}_Z}(Z) + \text{Tr}\left(\Lambda^\top (AX - Z - b)\right) + \frac{\xi}{2}\|AX - Z - b\|_F^2 \tag{16}$$

where $\Lambda \in \mathbb{R}^{H \times 2}$ is the dual variable, $\xi > 0$ is the augmented Lagrangian penalty parameter, $\text{Tr}(\cdot)$ denotes the trace of a matrix, and $\|\cdot\|_F$ denotes the Frobenius norm.

The update rules of $X$, $Z$ and the dual $\Lambda$ is derived as follows by ADMM:

- $X$-update:
$$X^{k+1} = \arg\min_X \mathcal{L}_\xi(X, Z^k, \Lambda^k).$$

It has a closed-form solution given by taking the gradient w.r.t. $X$ and setting it to zero:
$$2(X - \hat{X}) + A^\top \Lambda^k + \xi A^\top (AX - Z^k - b) = 0,$$

which yields
$$X^{k+1} = (2\boldsymbol{I}_H + \xi A^\top A)^{-1}(2\hat{X} + \xi A^\top Z^k + \xi A^\top b - A^\top \Lambda^k). \tag{17}$$

- $Z$-update:
$$Z^{k+1} = \arg\min_Z \mathcal{L}_\xi(X^{k+1}, Z, \Lambda^k),$$

which is
$$Z^{k+1} = \arg\min_{Z \in \mathcal{K}_Z} \left( \mathrm{Tr}\left(-Z^\top \Lambda^k\right) + \frac{\xi}{2}\|Z - (AX^{k+1} - b)\|_F^2 \right).$$

The solution to the above is
$$Z^{k+1} = \Pi_{\mathcal{K}_Z}\left( AX^{k+1} - b + \frac{1}{\xi}\Lambda^k \right), \tag{18}$$

where the projection operation is applied row-wise
$$Z_h^{k+1} = \begin{cases} (v_{\max}\Delta t)\frac{w_h}{\|w_h\|} & \text{if } \|w_h\| > v_{\max}\Delta t, \\ w_h & \text{otherwise,} \end{cases}$$

with $w_h$ being the $h$-th row of $\left( AX^{k+1} - b + \frac{1}{\xi}\Lambda^k \right)$.

- $\Lambda$-update:
$$\Lambda^{k+1} = \Lambda^k + \xi(AX^{k+1} - Z^{k+1} - b). \tag{19}$$

*Remark* 2. Note that the matrix $\left(2\boldsymbol{I}_H + \xi A^\top A\right)$ in Eq. (17) is symmetric positive definite and constant across iterations, which allows for caching its inverse. A more efficient approach is to perform LU or Cholesky decomposition on $(2\boldsymbol{I}_H + \xi A^\top A)$ once and solve the linear system
$$(2\boldsymbol{I}_H + \xi A^\top A)X^{k+1} = 2\hat{X} + \xi A^\top Z^k + \xi A^\top b - A^\top \Lambda^k$$

at each iteration using the cached LU or Cholesky factors.

The above derivations lead to Algorithm 4.

### B.2.3 OBSTACLE COST

For static obstacle avoidance, we follow Carvalho et al. (2023) using a cost based on signed distance to a static obstacle. Specifically, let $\varphi(x) : \mathbb{R}^2 \to \mathbb{R}$ be a *differentiable* signed distance from a robot to its *closest* obstacle, and then the obstacle cost term reads
$$c_{\text{obst}}(X^1, \ldots, X^N) = \sum_{h=1}^H \sum_{i=1}^N \mathbf{1}\left[\varphi(X_h^i) \leq r'\right] \cdot \left(r' - \varphi(X_h^i)\right) \tag{20}$$

where $r' > 0$ is also a parameter.

### B.2.4 HYPERPARAMETERS

Let all robots share the same radius $R$. We set $\lambda_{\text{robo}} = 1.0$ and $\lambda_{\text{obst}} = 0.1/\gamma$. For SHD cost, we set $\rho = 6R$ and typically $\gamma \in [0.6, 3.0]$. For LB cost, we set $\alpha = 1.9R$ and typically $\gamma \in [0.01, 0.06]$. Regarding projection, typically we set the penalty $\xi = 10$, max iteration $K_{\max} = 1000$, and tolerance $\varepsilon = 3 \times 10^{-6}$.

## B.3 DIVERSE ROBOT MANIPULATION EXPERIMENT

### B.3.1 DIFFUSION POLICY

We adopt DIFFUSION POLICY (DP) (Chi et al., 2023) as our base algorithm, using pretrained weights from (Feng et al., 2024)[11]. DP is a conditional diffusion model operating in a receding-

---

[11]The model from (Feng et al., 2024) was trained on an augmented dataset with broader coverage than that of (Chi et al., 2023), yielding more diverse and feasible trajectories.

---

**Algorithm 4** Batched ADMM Projection for Velocity-Constrained Trajectories

---

**Require:** Predicted trajectory batch $\hat{\mathbf{X}}$, starting position batch $\mathbf{x}_0$, $v_{\max} > 0$, $\Delta t > 0$, penalty $\xi > 0$, max iteration $K_{\max}$, tolerance $\varepsilon$.

1: **Pre-compute** $A$ and batched $\mathbf{b}$ matrices
2: **Pre-compute** $M \leftarrow 2\boldsymbol{I}_H + \xi A^\top A$ $\qquad\qquad\qquad\qquad\qquad$ ▷ *same for all batches*
3: **Caching** inverse or factors of $M$ $\qquad\qquad\qquad\qquad\qquad$ ▷ *same for all batches*
4: **Initialize** $\mathbf{Z}^0 \leftarrow \mathbf{0}$, $\boldsymbol{\Lambda}^0 \leftarrow \mathbf{0}$ $\qquad\qquad\qquad$ ▷ *zero tensors of shape* $B \times H \times 2$
5: **for** $k \leftarrow 0$ to $K_{\max} - 1$ **do**
6: $\quad$ ▷ *X-update, Eq. (17)* $\qquad\qquad\qquad\qquad\qquad\qquad\qquad\qquad\qquad\qquad$ ◁
7: $\quad \mathbf{V} = 2\hat{\mathbf{X}} + \xi A^\top \left( \mathbf{Z}^k + \mathbf{b} - \frac{1}{\xi}\boldsymbol{\Lambda}^k \right)$ $\qquad\qquad$ ▷ *A broadcasts across batches*
8: $\quad \mathbf{X}^{k+1} \leftarrow \texttt{SolveLinearSystBatch}\big(M\mathbf{X}^{k+1} = \mathbf{V}\big)$ $\quad$ ▷ *M broadcasts across batches*
9: $\quad$ ▷ *Z-update, Eq. (18)* $\qquad\qquad\qquad\qquad\qquad\qquad\qquad\qquad\qquad\qquad$ ◁
10: $\quad \mathbf{W} \leftarrow A\mathbf{X}^{k+1} - \mathbf{b} + \frac{1}{\xi}\boldsymbol{\Lambda}^k$
11: $\quad$ **for all** $(\beta, h) \in \{1, \ldots, B\} \times \{1, \ldots, H\}$ **in parallel do** $\qquad$ ▷ *Vectorized operation*
12: $\quad\quad \mathbf{Z}_{\beta,h}^{k+1} \leftarrow \min\{v_{\max}\Delta t, \|\mathbf{W}_{\beta,h}\|\} \frac{\mathbf{W}_{\beta,h}}{\|\mathbf{W}_{\beta,h}\| + \iota}$ $\qquad$ ▷ *Small* $\iota > 0$ *to avoid singularity*
13: $\quad$ ▷ *Dual-update, Eq. (19)* $\qquad\qquad\qquad\qquad\qquad\qquad\qquad\qquad\qquad$ ◁
14: $\quad \boldsymbol{\Lambda}^{k+1} \leftarrow \boldsymbol{\Lambda}^k + \xi\left(A\mathbf{X}^{k+1} - \mathbf{Z}^{k+1} - \mathbf{b}\right)$
15: $\quad$ ▷ *Optional: Convergence check* $\qquad\qquad\qquad\qquad\qquad\qquad\qquad\qquad$ ◁
16: $\quad$ **if** check convergence **then**
17: $\quad\quad \mathbf{R} \leftarrow A\mathbf{X}^{k+1} - \mathbf{Z}^{k+1} - \mathbf{b}$ $\qquad\qquad\qquad\qquad$ ▷ *Primal residuals*
18: $\quad\quad \mathbf{S} \leftarrow \xi A^\top \left(\mathbf{Z}^{k+1} - \mathbf{Z}^k\right)$ $\qquad\qquad\qquad\qquad\qquad$ ▷ *Dual residuals*
19: $\quad\quad r_{\max} \leftarrow \max_{\beta=1,\ldots,B} \{\|\mathbf{R}_\beta\|_F, \|\mathbf{S}_\beta\|_F\}$
20: $\quad\quad$ **if** $r_{\max} \leq \varepsilon$ **then**
21: $\quad\quad\quad$ **break**
22: **return** $\mathbf{X}^{k+1}$

---

horizon manner. Conditioned on an observation $O \in \mathbb{R}^{H_o \times 5}$ (a trajectory of $H_o$ steps where each step is a 5D state vector capturing the planar position of the robot end effector and the $\mathsf{T}$ block's center and orientation), DP generates an action segment $X \in \mathbb{R}^{H \times 2}$ representing future end-effector positions. Only the first $H_a \leq H$ steps are executed (in simulation for our case), after which a new observation $O'$ is obtained and the process repeats, until a total of $H_{\max}$ execution steps is reached.

### B.3.2 DETAILED SETUP

We compare our method with the vanilla DIFFUSION POLICY (DP) as baseline, DP with only projection, and DP with only coupling. We evaluate each method on 50 uniformly random initial observations. With each method, we generate 100 *pairs* of full trajectories $A \in \mathbb{R}^{H_{\max} \times 2}$ (concatenated by the *executed portion* of each action segments) conditioned on every initial observation. For projection, we choose three different max velocity limits, corresponding to the 80%, 90% and 95% quantiles of the velocities[12] generated by the baseline across the initial observations. We use 32 diffusion steps at inference, and take 1 gradient descent step for coupling. We adopt the setting of prediction horizon $H = 16$, action horizon $H_a = 8$, and observation horizon $H_o = 2$ as recommended in (Chi et al., 2023). The maximal action steps $H_{\max}$ is set to 360.

*Remark* 3. We only take the executed part of the generated action segments and concatenate them along time dimension to form a *full* trajectory $A \in \mathbb{R}^{H_{\max} \times 2}$ for evaluation.

### B.3.3 PROJECTION DETAILS

We use the same formulation and implementation of projection as in the multi-robot experiments. The parameters used are penalty $\xi = 6.0$, max iteration $K_{\max} = 250$, and tolerance $\varepsilon = 2 \times 10^{-4}$.

---

[12]Calculated by forward difference of positions of the robot.

### B.3.4 COUPLING COSTS

For all methods at all velocity limits we use the same cost-dependent $\gamma$ value. Concretely, for DPP and DPP-PS costs we set $\gamma = 0.2$; for LB and LB-PS costs we set $\gamma = 0.02$. These parameters are chosen based on a coarse parameter scan and selecting one among the Pareto front of the TC-DTW and TC-DFD relations.

### B.3.5 DETAILS IN EVALUATION METRICS

We use four quantitative metrics for evaluation: Dynamic Time Warping (DTW) (Berndt & Clifford, 1994; Müller, 2007), discrete Fréchet distance (DFD) (Alt & Godau, 1995), velocity constraint satisfaction rate (CS), and task completion score (TC) (Florence et al., 2022; Chi et al., 2023). DTW and DFD quantifies dissimilarity between two trajectories and have been widely used in robotics (Bucker et al., 2023; Memmel et al., 2025) and dynamical systems learning (Rana et al., 2020; Zhang et al., 2022). For each pair of *full* trajectories, we report the DTW and DFD between the two corresponding segments, and average them over number of segments within each full trajectory. The velocity constraint satisfaction rate for each full trajectory is defined as the fraction of action segments respecting the constraint within the full trajectory. The task completion score measures how well the manipulation task is accomplished by a *full* trajectory given an initial observation, where 1.0 is the best, and 0 the worst. We report all metrics by their empirical mean over all initial observations and full trajectory pairs.

### B.4 CONSTRAINED COUPLED IMAGE PAIR GENERATION EXPERIMENT

### B.4.1 TEXT PROMPTS FOR EXEMPLAR GENERATION

All exemplar images in this toy example were generated using ChatGPT (OpenAI, 2023). We first employ a generic prompt:

> *"Generate a simple, clean portrait of a male/female in his/her early 30s, wearing a plain T-shirt, against a white studio background with soft lighting. The portrait should be a centered headshot, similar to a passport photo."*

to generate the seed exemplar. We then provide the seed exemplar alongside a follow-up prompt instructing ChatGPT to regenerate the exact same image, but depicting the subject at an older age, or with different clothing.

### B.4.2 TEXT PROMPT FOR STABLE DIFFUSION AND BASELINES

As shown in both Figure 4e and Table 3, we also run additional comparisons against SD-2.1 with coupling, SyncDiffusion and ControlNet-XS, all with the use of a generic text prompt. For all text-prompted runs, we use

> *"High-resolution passport photo of a person, facing forward with a neutral expression. Wearing a plain white t-shirt, with a clean white background and even, soft lighting. The composition is centered and symmetrical, with the head at the center of the frame."*

We set classifier-free guidance scale $s=25$ for all SD-2.1 variants, and use the authors' recommended defaults for external baselines, that is: $s=7.5$ for SyncDiffusion, and $s=9.5$ for ControlNet-XS. We also set all negative prompts to *null* strings.

### B.4.3 DETAILED SETUP FOR SYNCDIFFUSION

We choose SyncDiffusion (Lee et al., 2023), which was originally designed for panorama generation with an LPIPS-based style loss to promote global synchronization and coherence, as a reference method to compare performance on promoting age-group contrast. Since SyncDiffusion allows for an arbitrary style loss, we replace the LPIPS loss with our coupling loss $c_{\text{XOR}}(x, y)$, parameterized by an image classifier trained on FFHQ-Aging (Or-El et al., 2020). We set the latent stride

to $s_{\text{latent}}{=}64$ to match SD-2.1's latent spatial dimensions, which disables any form of latent overlapping during sampling and essentially reduces the setup to paired image generation. We employ 100 DDIM (Song et al., 2021a) steps and use a synchronization weight $w_{\text{sync}}{=}1.3$ with exponential decay $d_{\text{sync}}{=}0.99$. We perform one synchronization step for all 100 diffusion steps. All other hyperparameters follow the defaults by Lee et al. (2023) unless stated otherwise.

### B.4.4 DETAILED SETUP FOR CONTROLNET-XS

ControlNet-XS (Zavadski et al., 2024) is used for reference to compare performance in facial attribute preservation. We use the official ControlNet-XS implementation. We adopt the *SD-2.1 Canny Edge 14M* model weights provided by the authors, rather than the *SD-2.1 Depth Map 14M* variant, as Canny edges preserve higher-frequency facial details relevant to our attributes. The Canny thresholds are set to $t_{\text{high}}{=}250$ and $t_{\text{low}}{=}100$ as per the recommended values by Zavadski et al. (2024). We use a ControlNet control weight of $w_{\text{ctrl}}{=}0.95$ and set DDIM sampling steps to 100.

In this setup, we note that ControlNet-XS serves as a soft-constraint baseline: it encourages attribute alignment via conditioning but does not enforce feasibility, in contrast to our projection-based hard-constraint method.

**Condition aggregation across exemplars.** ControlNet-XS accepts a *single* conditioning image per generation. To obtain a single representative condition from an exemplar set, we compute the arithmetic mean of the exemplars in SD-2.1's latent space and decode it back to image space for conditioning. Because the exemplars are structurally and spatially aligned (see discussion in Appendix **B.4.7**), this linear averaging preserves shared spatial attributes. Geometrically, the latent average lies near the centroid of the convex hull spanned by the exemplars, providing a neutral, exemplar-consistent condition. Conditioning images derived from $M \in \{2, 6\}$ exemplars per model are visualized in Figures 6a–6b.

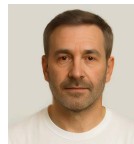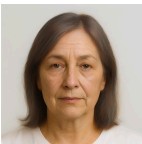    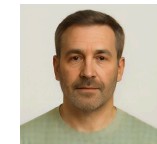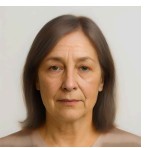

(a) Conditioning images for $M{=}2$ exemplars        (b) Conditioning images for $M{=}6$ exemplars

Figure 6: Conditioning images used for ControlNet-XS. (a) Conditioning images formed by taking a linear average in latent space for $M{=}2$ exemplar images from Figure 4a. (b) Likewise, but for $M{=}6$ exemplar images from Figure 25.

### B.4.5 PROJECTION VIA MIRROR DESCENT

Given two exemplar sets of sizes $M_x$ and $M_y$, we encode them via VAE encoders and obtain the latents $X^{(e)} = [\, X_1^{(e)} \cdots X_{M_x}^{(e)} \,] \in \mathbb{R}^{d \times M_x}$ and $Y^{(e)} = [\, Y_1^{(e)} \cdots Y_{M_y}^{(e)} \,] \in \mathbb{R}^{d \times M_y}$, where $d$ is the flattened latent dimension. Define $\mathcal{K}_X = \{\, X^{(e)}\lambda \mid \lambda \in \Delta_{M_x} \}$ the constraint set for $X$, where $\Delta_{M_x}$ is an $M_x$-simplex, and define $\mathcal{K}_Y$ likewise. At each diffusion step $t$, we project the current latent $X_t$ onto $\mathcal{K}_X$ by solving the simplex-constrained problem:

$$\lambda_{X,t}^{\star} = \arg\min_{\lambda \in \Delta_{M_x}} \|X^{(e)}\lambda - X_t\|_2^2 \tag{21}$$

via Mirror Descent (MD) using the negative-entropy mirror map, which yields exponentiated-gradient updates that remain on $\Delta_{M_x}$ by construction (Nemirovsky & Yudin, 1983; Beck & Teboulle, 2003). For the MD updates, define

$$G_X := X^{(e)\top} X^{(e)} \in \mathbb{R}^{M_x \times M_x}, \tag{22a}$$

$$b_{X,t} := X^{(e)\top} X_t \in \mathbb{R}^{M_x}. \tag{22b}$$

and let $f_{X,t}(\lambda) = \|X^{(e)}\lambda - X_t\|_2^2$. Its gradient is

$$\nabla f_{X,t}(\lambda) = 2\,(G_X \lambda - b_{X,t}). \tag{23}$$

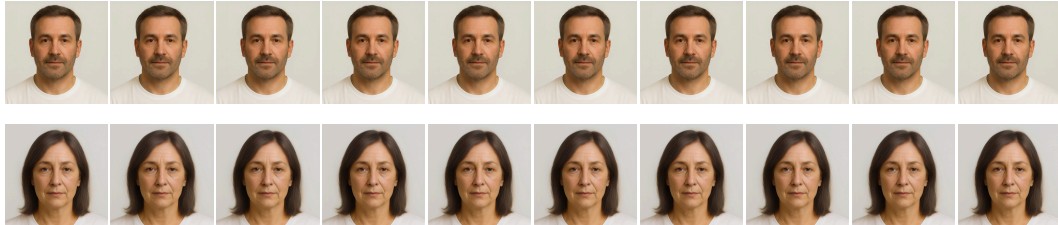

Figure 7: Generated samples obtained by projecting intermediate DDPM latents (Eq. (29)) onto the exemplar convex hulls at every step. Top: projection using the exemplar pair from the *top* row of Figure 4a; bottom: using the *bottom*-row pair. Within each row, the samples collapse to a narrow mode, illustrating significantly reduced diversity induced by per-step projection onto a fixed exemplar set.

Starting from $\lambda_{X,t}^{(0)} = \frac{1}{M_x}\mathbf{1}$, each MD step performs

$$\log \lambda_{X,t}^{(k+1)} = \log \lambda_{X,t}^{(k)} - \eta \, \nabla f_{X,t}(\lambda_{X,t}^{(k)}), \tag{24a}$$

$$\lambda_{X,t}^{(k+1)} = \mathrm{softmax}\big(\log \lambda_{X,t}^{(k+1)}\big), \tag{24b}$$

with learning rate $\eta > 0$, which is equivalent to

$$\lambda_{X,t}^{(k+1)} \propto \lambda_{X,t}^{(k)} \odot \exp\left\{-\eta \, \nabla f_{X,t}\left(\lambda_{X,t}^{(k)}\right)\right\} \tag{25}$$

followed by normalization. After $K_{\max}$ steps, we obtain $\lambda_{X,t}^{\star}$ and compute the final projected latent $\hat{X}_t = X^{(e)}\lambda_{X,t}^{\star}$. $\hat{Y}_t$ is computed likewise. We run MD for $K_{\max} = 10,000$ steps and set its learning rate $\eta = 10^{-5}$ for convergence.

**Mode Collapse.** We follow the DDPM formulation of Ho et al. (2020). Let $\{\beta_t\}_{t=1}^{T} \subset (0,1)$ be a predefined noise-variance schedule for the forward process $q(z_t \mid z_{t-1}) = \mathcal{N}(\sqrt{1-\beta_t}\, z_{t-1},\, \beta_t I)$. Define

$$\alpha_t := 1 - \beta_t, \tag{26}$$

$$\bar{\alpha}_t := \prod_{s=1}^{t} \alpha_s \quad \text{(cumulative product)}, \tag{27}$$

$$\tilde{\beta}_t := \frac{1 - \bar{\alpha}_{t-1}}{1 - \bar{\alpha}_t}\, \beta_t \quad \text{(posterior variance)}. \tag{28}$$

At inference, the standard DDPM sampling update with a learned noise predictor $\varepsilon_\theta(x_t, t)$ is

$$z_{t-1} = \frac{1}{\sqrt{\alpha_t}}\left(z_t - \frac{\beta_t}{\sqrt{1-\bar{\alpha}_t}}\, \varepsilon_\theta(z_t,t)\right) + \sqrt{\tilde{\beta}_t}\, \epsilon, \qquad \epsilon \sim \mathcal{N}(0,I) \text{ if } t > 1, \text{ else } 0. \tag{29}$$

Recall $\mathcal{K}_X = \{\, X^{(e)}\lambda : \lambda \in \Delta_{M_x} \,\}$ (and $\mathcal{K}_Y$ analogously for $Y$). When sampling with Eq. (29), we observe that projecting the latents at every step — i.e., $\hat{X}_t = \Pi_{\mathcal{K}_X}(X_t)$ and $\hat{Y}_t = \Pi_{\mathcal{K}_Y}(Y_t)$ — leads to pronounced mode collapse in the generated samples as shown in Figure 7.

As shown in (Song et al., 2021b; Luo, 2022), the (Stein) score predicted by the model at timestep $t$ can be derived from the model's noise prediction $\varepsilon_\theta(z_t, t)$ using Tweedie's formula:

$$s_\theta(z_t, t) = \nabla_{z_t} \log p_t(z_t) \approx -\frac{1}{\sqrt{1-\bar{\alpha}_t}}\, \varepsilon_\theta(z_t, t). \tag{30}$$

In Figure 8, we analyze the score field along the 1-D subspace spanned by two exemplar latents from the second row of Figure 4a. Specifically, we project the score $\nabla_{z_t} \log p_t(z_t)$ onto the line segment (the convex hull of two points) joining the two exemplars — the direction preserved by the projection

operator when there are exactly two exemplars — and visualize its signed magnitude. Because latent interpolatability in our setting relies on structurally and spatially aligned exemplars (see discussion on Latent Space Interpolatability in later sections), projection induces a narrow feasible corridor. With projection enabled, this projected score points almost exclusively toward the same endpoint across timesteps, yielding a nearly deterministic path toward the final projected latent $\hat{z}_0$. This concentration substantially reduces sample diversity for the given exemplar set.

While this concentrates samples within a narrower neighborhood of the exemplars, it also delivers strong controllability and structural fidelity — properties that are particularly valuable for tasks such as image editing, personalization, and attribute-preserving transformations. We view leveraging this precision-fidelity regime as a promising direction for such application-oriented extensions.

Following Nichol & Dhariwal (2021), which highlights the impact of reverse-process variance on sampling, we additionally adopt a DDIM-style stochasticity control and scale the noise term by a factor of $k$ during sampling to increase output diversity. The updated DDPM sampling update is hence

$$z_{t-1} = \frac{1}{\sqrt{\alpha_t}} \left( z_t - \frac{\beta_t}{\sqrt{1 - \bar{\alpha}_t}} \varepsilon_\theta(z_t, t) \right) + \sqrt{\tilde{\beta}_t}\, k\, \epsilon, \qquad \epsilon \sim \mathcal{N}(0, I) \text{ if } t > 1, \text{ else } 0. \quad (31)$$

where $k \in \mathbb{R}_{\geq 1}$ scales the stochastic term ($k = 1$ recovers standard DDPM; $k > 1$ increases the noise standard deviation by $k$, variance by $k^2$). To set $k$, we compute at each diffusion step the average ratio $r_t = \left\| \nabla_{z_t} \log p_t(z_t) \right\| / \left\| \Pi(\nabla_{z_t} \log p_t(z_t)) \right\|$ between the model score magnitude and the magnitude of its projected component (see Figure 9). We aggregate $r_t$ over timesteps and exemplar sets and choose $k$ near this summary; empirically, $k = 20$ provides a good diversity-stability trade-off for experiment runs involving projection. See Appendix C for ablation on $k$.

### B.4.6 COUPLING GUIDANCE STRENGTH

We observe that projection dampens the gradients supplied by the coupling loss, which calls for the need of $\gamma$ to be scaled well beyond the values typical for classifier guidance (Dhariwal & Nichol, 2021) in order to take effect. We find that empirically, recognizable young–old contrast appears only when $\gamma \geq 200$. This is consistent with our earlier finding that the score component aligned with the line segment connecting the exemplar latents is roughly 13–20× weaker than the full score (as illustrated in Figure 9) - the projection operator only preserves the component of gradients provided by the coupling loss that is parallel to the line segment connecting the two exemplar sets. Consequently, a simple estimate of the effective guidance strength gives

$$\gamma_{\text{eff}} \approx \gamma / r_t \quad (32)$$

which means compensating for a $r_t \approx 20$ reduction requires $\gamma$ to increase by roughly 50×. We therefore set $\gamma = 450$ (yielding $\gamma_{\text{eff}} \approx 9.0$) for all experiment runs involving both coupling and projection if not otherwise specified. See Appendix C for an ablation on $\gamma$.

### B.4.7 LATENT SPACE INTERPOLATABILITY

VAEs are trained to encourage a smooth, approximately Euclidean latent space by regularizing posteriors toward a standard Gaussian prior, making nearby latent codes decode to similar images and thereby support interpolation (Kingma & Welling, 2014; Rezende et al., 2014). However, we find empirically that *linear* interpolation is reliable only when exemplars are both structurally and spatially aligned. Concretely, if spatial layouts differ, straight-line paths (and thus convex-hull projections) tend to leave the data manifold and decode implausibly. See examples of interpolation between latents of samples from FFHQ-Aging Dataset (Or-El et al., 2020) in Figure 10. This observation accords with geometric analyses showing that semantically consistent transitions follow *curved* geodesics under the decoder-induced Riemannian metric, rather than straight lines in Euclidean latent coordinates (Arvanitidis et al., 2018). Hence, we also experimented with spherical linear interpolations (SLERP) (Shoemake, 1985) to maintain constant-norm paths under an isotropic Gaussian prior. In practice, SLERP still required close structural alignment of exemplars, and it only interpolates between two endpoints, whereas our convex-hull projection must accommodate multiple exemplars. Moreover, adopting SLERP consistently would call for a manifold-aware projection, which is nontrivial and beyond the scope of this work. Consequently, we use simple linear interpolation but restrict convex sets to closely related exemplars to preserve visual coherence.

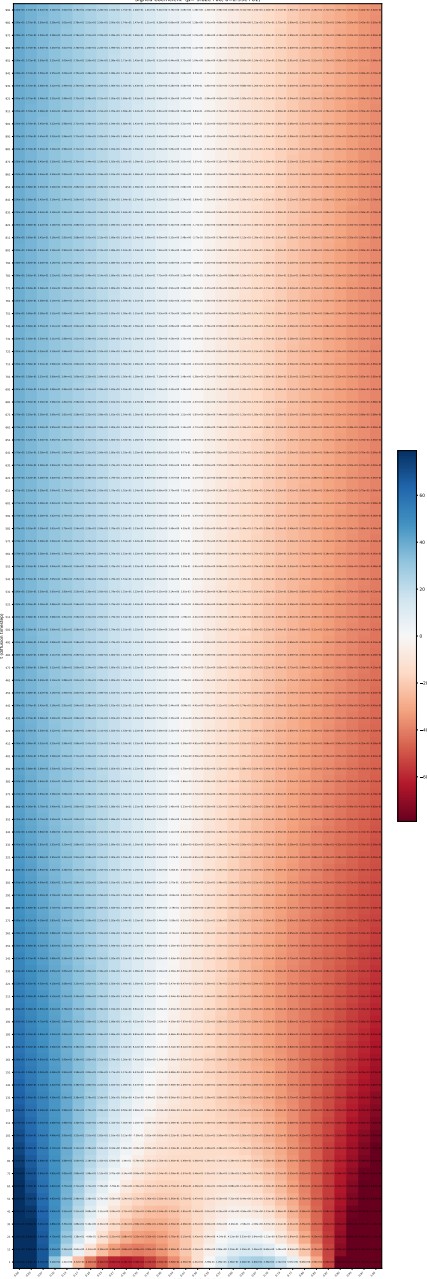

Figure 8: Signed component of the model score $\nabla_{z_t} \log p_t(z_t)$ projected onto the line segment connecting two exemplar latents (the convex hull of two exemplars). The x-axis represents interpolation between the two exemplar latents (from left to right), and the y-axis denotes the diffusion timestep $t$. *Color indicates the direction and strength of the projected score: blue values push towards the right exemplar, while red values push towards the left.* This visualization, based on two exemplars from row 2 of Figure 4a, reveals that the projected score components consistently point towards one side (left), creating a narrow "white" transition band that funnels every sample to the left exemplar — hence a nearly deterministic path and little diversity. Such behavior is observed consistently in all exemplar sets experimented and likely stems from the exemplars being relatively similar, which induces a narrow feasible region for the diffused latents. Empirically, the exemplars cannot differ too much in spatial structure. When exemplars differ significantly, interpolations between them often fail to represent coherent or meaningful images (see discussion on Latent Space Interpolatability in later sections).

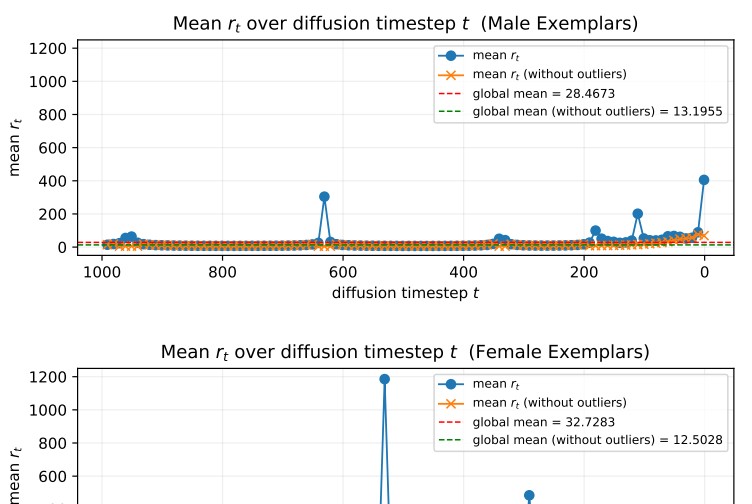

Figure 9: Mean ratio $r_t = \left\|\nabla_{z_t} \log p_t(z_t)\right\| / \left\|\Pi\!\left(\nabla_{z_t} \log p_t(z_t)\right)\right\|$ over diffusion timesteps $t$, where $\Pi(\cdot)$ projects onto the convex hull formed by exemplar latents. Top: computed with male exemplar pair (top row of Fig. 4a); bottom: female exemplar pair (bottom row). Large outlier $r_t$ values indicate the predicted score is nearly orthogonal to the chord connecting the two exemplar latents. Outliers are defined as $r_t \geq \mu + 1.5\sigma$, where $\mu$ is the global mean and $\sigma$ the global standard deviation across $t$.

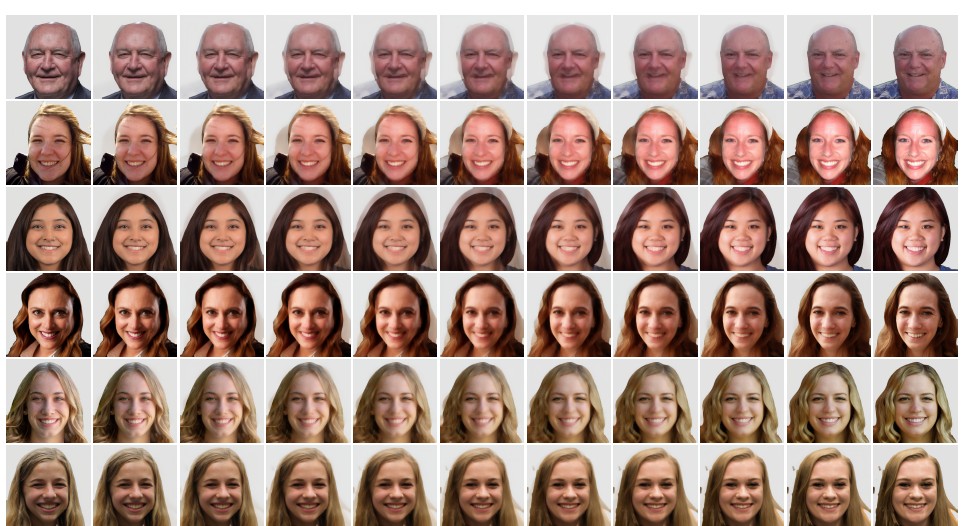

Figure 10: Linear latent interpolations between pairs of FFHQ-Aging Dataset images (Or-El et al., 2020). Rows 1–3 use exemplars that differ in pose or spatial layout; midway latents leave the data manifold and decode to implausible faces. Rows 4–6 use structurally aligned exemplars; the entire interpolation produces coherent and plausible images. The contrast illustrates that latent interpolations are reliable only for closely aligned exemplars.

| Method Pair \Metric | JS | TV | L2 |
|---|---|---|---|
| (PCD, CG) | $3.91 \times 10^{-5}$ | $5.40 \times 10^{-3}$ | $1.38 \times 10^{-3}$ |
| (PCD, PD) | $5.23 \times 10^{-5}$ | $7.97 \times 10^{-3}$ | $1.44 \times 10^{-3}$ |
| (Baseline, Baseline) | $4.48 \times 10^{-5}$ | $5.58 \times 10^{-3}$ | $1.32 \times 10^{-3}$ |

Table 4: Distributional metrics results for numerical verification of PCD special cases.

## C ADDITIONAL RESULTS

### C.1 NUMERICAL VERIFICATION OF PCD SPECIAL CASES

We numerically verify that PCD recovers into two known methods, classifier guidance (CG) and projected diffusion (PD), under specific circumstances as stated in Section 4.

**Setup** We experiment on the 1D toy example demonstrated in Section 4, Figure 1. We run PCD in two degenerated versions that correspond to CG and PD, respectively, as well as separately implemented CG and PD, and then compare their *empirical* sample distributions over the $X$ variable. Specifically, we discretize the range of $X$ into 200 bins and compare the resulting histograms with a sample size of $10^6$. For the degenerated PCD corresponding to CG, we fix the $Y$ variable in PCD to the center of the corridor to match the conditioning used in CG, and adopt the same coupling or guidance strength for PCD and CG. For the degenerated PCD corresponding to PD, we set the coupling strength in PCD as $\gamma = 0$. Additionally, we include a "baseline" where we draw two groups of samples from the same standard normal distribution, each with a different random seed, in order to provide a numerical reference in the reported metrics to demonstrate the effects of stochasticity with the chosen sample size.

**Metrics** To quantify similarity between distributions, we report three common metrics: Jensen-Shannon divergence (JS), total variation distance (TV), and $L_2$ distance (L2).

**Results** The resulting distributional discrepancies are quantified and reported in Table 4. These divergences show that the discrepancies in sample distributions between the degenerated PCD and the corresponding special cases (CG or PD) lie within the *same order of magnitude* as the baseline case, wherein two groups of samples are indeed drawn from the same distribution. Visualization of the empirical distributions is in Figure 11. These results numerically verify that PCD recovers the behaviors of CG and PD in degenerated scenarios, and thus can be regarded as a generalization.

### C.2 RUNTIME AND MEMORY

PCD is approximately $4 \sim 7\times$ slower than vanilla diffusion mainly due to the per-step projection operation. Memory overhead compared to vanilla diffusion is negligible.

**Image Pair Generation** Table 5 shows the per-diffusion step runtime of each operation in seconds, computed across 100 diffusion steps, for the *Constrained Coupled Image Pair Generation* experiment, using $M \in \{2, 6\}$ exemplars from Figure 4a and Figure 25 respectively. In terms of noticeable time differences, all runs involving projection in the $M$=6 exemplars setup is only $\approx 0.15$ seconds slower than the $M$=2 exemplars setup. This small increase in runtime is due to our projection operator being implemented as the Mirror Descent algorithm (Nemirovsky & Yudin, 1983; Beck & Teboulle, 2003) which scales effortlessly on GPUs; see Section B.4 for details. Across both setups, we can observe that: (i) diffusion model forward passes and other miscellaneous overheads have nearly constant runtime across runs; (ii) coupling adds $\sim 0.5$ second per diffusion step; (iii) projection adds $\sim 3$ seconds, with almost negligible dependence on $M$. Relative to vanilla SD, per-step runtime increases by $\sim 1.5\times$ with coupling alone, $\sim 4\times$ with projection alone, and $\sim 4.5\times$ with both.

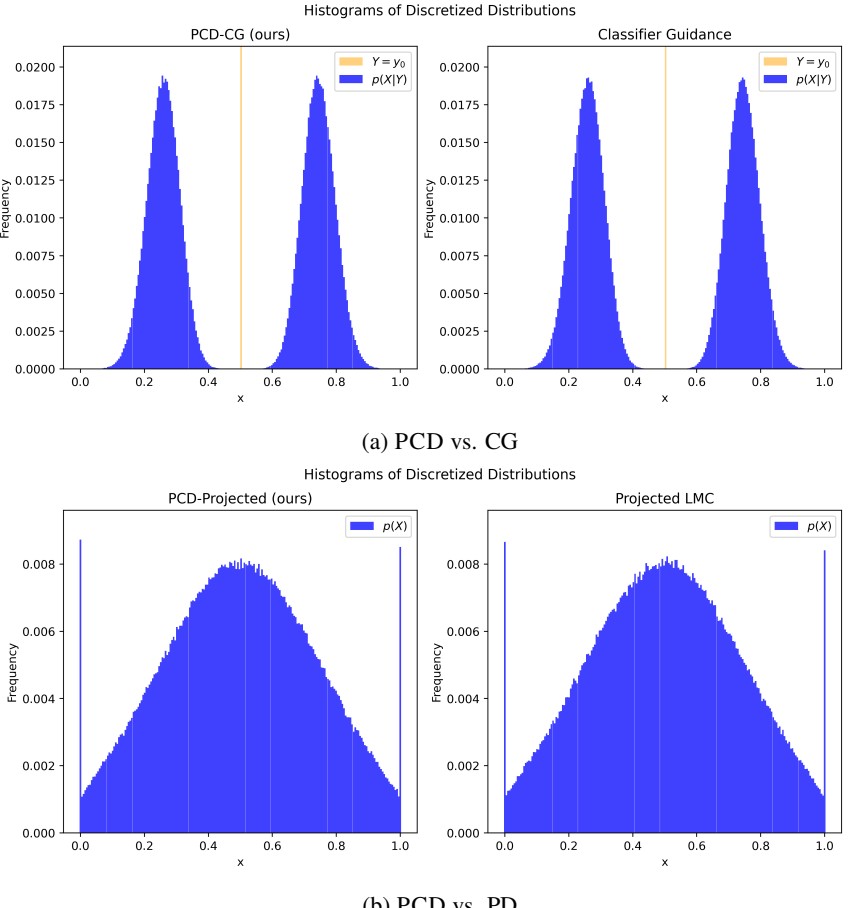

(a) PCD vs. CG

(b) PCD vs. PD

Figure 11: Empirical distributions of samples yielded by degenerated PCD compared to classifier guidance (CG) and projected diffusion (PD).

| METHOD | Model | Coupling | Projection | Misc | Total |
|--------|-------|----------|------------|------|-------|
| SD | $0.929_{\pm 0.005}$ | - | - | $0.003_{\pm 0.003}$ | $0.932_{\pm 0.007}$ |
| SD+C | $0.929_{\pm 0.005}$ | $0.505_{\pm 0.009}$ | - | $0.002_{\pm 0.002}$ | $1.436_{\pm 0.011}$ |
| SD+P | $0.919_{\pm 0.002}$ | - | $2.969_{\pm 0.027}$ | $0.002_{\pm 0.003}$ | $3.891_{\pm 0.028}$ |
| PCD | $0.919_{\pm 0.003}$ | $0.498_{\pm 0.010}$ | $2.984_{\pm 0.030}$ | $0.002_{\pm 0.002}$ | $4.402_{\pm 0.035}$ |

(a) $M=2$ exemplars.

| METHOD | Model | Coupling | Projection | Misc | Total |
|--------|-------|----------|------------|------|-------|
| SD | $0.932_{\pm 0.005}$ | - | - | $0.002_{\pm 0.003}$ | $0.934_{\pm 0.007}$ |
| SD+C | $0.929_{\pm 0.005}$ | $0.507_{\pm 0.010}$ | - | $0.002_{\pm 0.002}$ | $1.438_{\pm 0.012}$ |
| SD+P | $0.922_{\pm 0.002}$ | - | $3.042_{\pm 0.030}$ | $0.003_{\pm 0.002}$ | $3.967_{\pm 0.030}$ |
| PCD | $0.921_{\pm 0.003}$ | $0.503_{\pm 0.010}$ | $3.013_{\pm 0.011}$ | $0.002_{\pm 0.002}$ | $4.440_{\pm 0.022}$ |

(b) $M=6$ exemplars.

Table 5: Per-diffusion step runtime (in seconds; mean $\pm$ std over 100 diffusion steps) for each operation of the *Constrained Coupled Image Pair Generation* experiment. *Total* is the per-step sum. Components: *Model* = SD-2.1 UNet inference; *Coupling* = coupling step; *Projection* = projection step; *Misc* = other supporting ops (e.g., `Scheduler.step()`). Results are from a single run generating 25 samples with $M \in \{2, 6\}$ exemplars from Figure 4a and Figure 25 respectively. A dash indicates the component is not applied.

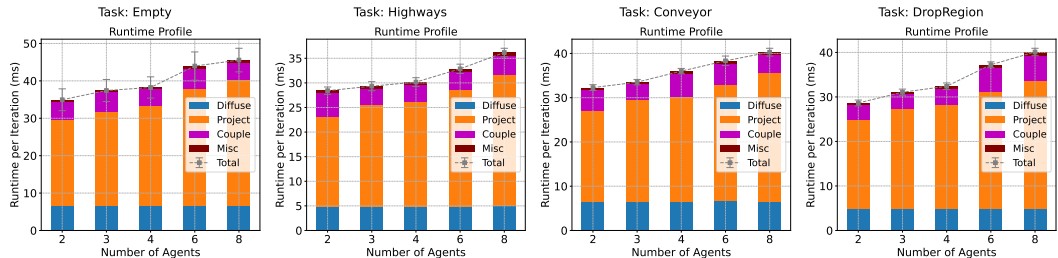

Figure 12: Runtime profile per iteration (in milliseconds) for each task scaling with the number of robots in the multi-robot experiment. The inference batch size for *each robot* is set to 32 and the total diffusion time step is set to 25 for profiling. Results are averaged over 50 random initial configurations for each task and the standard deviations are plotted with error bars.

**Multi-Robot Navigation** Figure 12 exhibits the runtime profiles of PCD running across different tasks and scaling with number of robots. Similar to image generation, the runtime fraction of the diffusion model's forward pass and associated overheads remains nearly constant despite the number of robots due to GPU parallelization, provided the total workload does not exceed the GPU's computational throughput. Runtime fraction of coupling approximately equals that of diffusion model forward pass but slightly varies across numbers of robots and tasks. The ADMM-based projection operation takes up roughly $67\%$ to $75\%$ of the total runtime and scales up approximately linearly with the numbers of robots (which conforms to theoretical complexity). Note that due to the batched convergence check described in Algorithm 4, the reported projection runtime is effectively the *worst case within each batch* per iteration. Implementing more efficient projectors such as adopting *adaptive* augmented Lagrangian penalty (Boyd et al., 2011) may help accelerate convergence.

## C.3 MULTI-ROBOT NAVIGATION

### C.3.1 ADDITIONAL QUALITATIVE RESULTS

Figures 13, 14, 15, 16 exhibit more trajectory samples generated by the compared methods on all four tasks with $N = 2$ and $N = 4$ robots. These results demonstrate the effectiveness of PCD in generating highly correlated trajectories and simultaneously enforcing hard constraints compared to methods with the absence of either coupling or projection, or both.

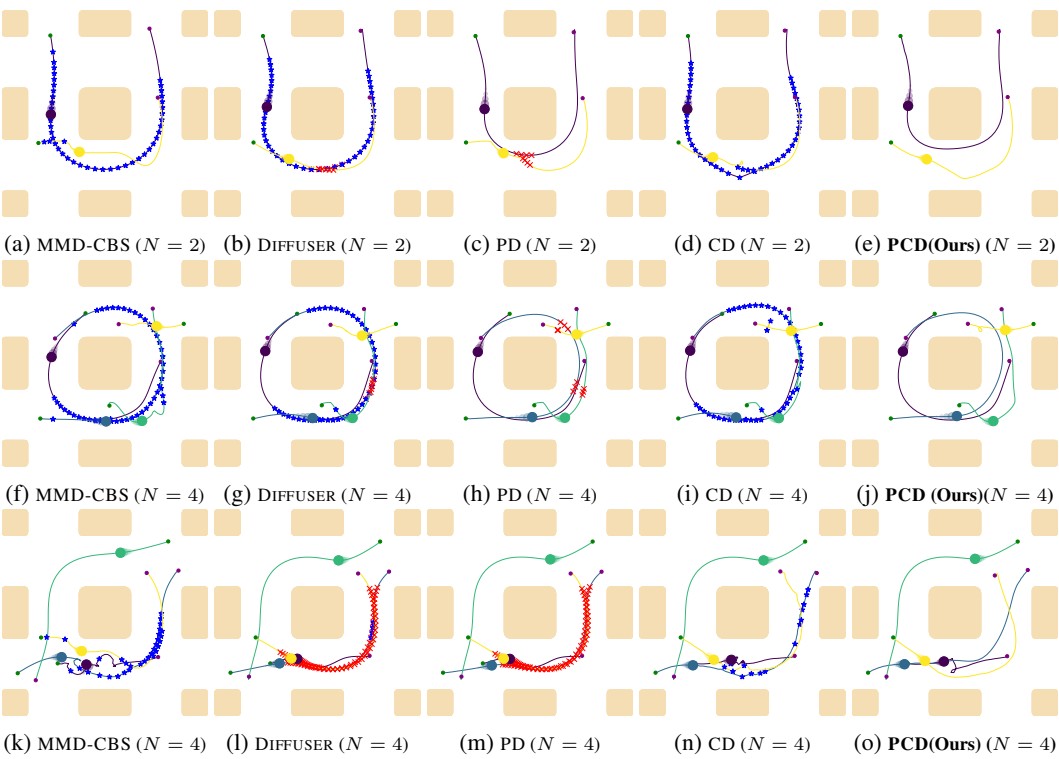

(a) MMD-CBS ($N = 2$)   (b) DIFFUSER ($N = 2$)   (c) PD ($N = 2$)   (d) CD ($N = 2$)   (e) **PCD(Ours)** ($N = 2$)

(f) MMD-CBS ($N = 4$)   (g) DIFFUSER ($N = 4$)   (h) PD ($N = 4$)   (i) CD ($N = 4$)   (j) **PCD (Ours)** ($N = 4$)

(k) MMD-CBS ($N = 4$)   (l) DIFFUSER ($N = 4$)   (m) PD ($N = 4$)   (n) CD ($N = 4$)   (o) **PCD(Ours)** ($N = 4$)

Figure 13: Robot trajectories in environment `Highways` generated by the compared methods with $N = 2$ and $N = 4$ robots running. Red crosses mark collisions and blue stars mark velocity constraint violations. Each row corresponds to one initial configuration (start and goal positions for each robot).

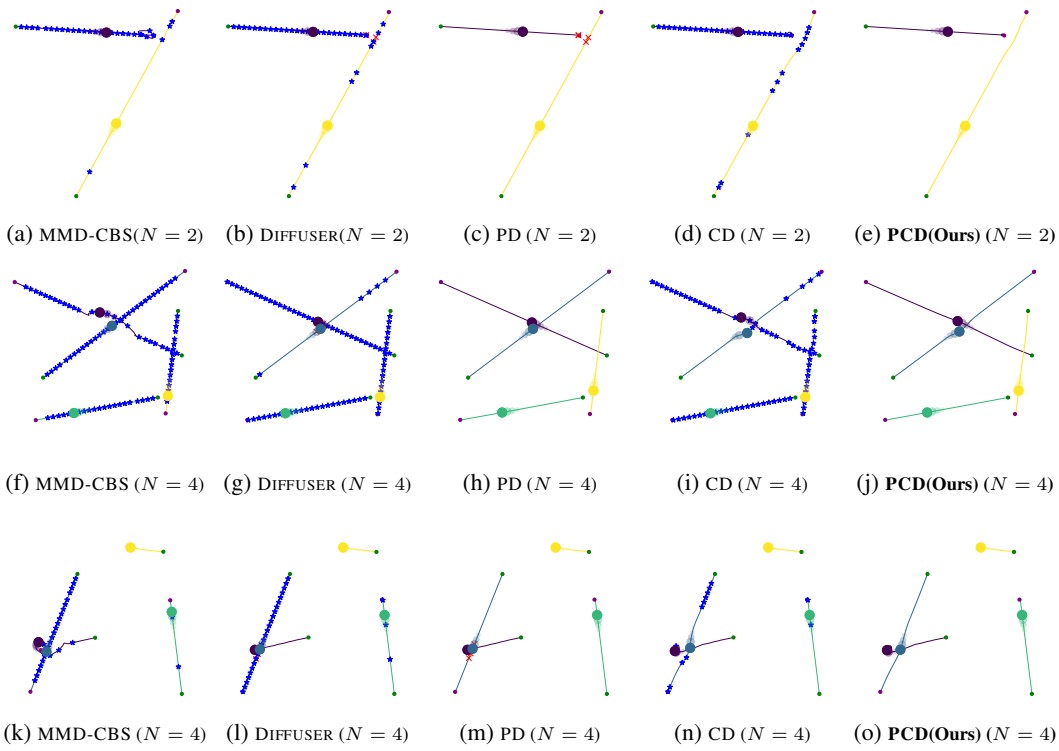

(a) MMD-CBS($N = 2$)    (b) DIFFUSER($N = 2$)    (c) PD ($N = 2$)    (d) CD ($N = 2$)    (e) **PCD(Ours)** ($N = 2$)

(f) MMD-CBS ($N = 4$)    (g) DIFFUSER ($N = 4$)    (h) PD ($N = 4$)    (i) CD ($N = 4$)    (j) **PCD(Ours)** ($N = 4$)

(k) MMD-CBS ($N = 4$)    (l) DIFFUSER ($N = 4$)    (m) PD ($N = 4$)    (n) CD ($N = 4$)    (o) **PCD(Ours)** ($N = 4$)

Figure 14: Robot trajectories in environment `Empty` generated by the compared methods with $N = 2$ and $N = 4$ robots running. Red crosses mark collisions and blue stars mark velocity constraint violations. Each row corresponds to one initial configuration.

### C.3.2 ADDITIONAL QUANTITATIVE RESULTS

Tables 6, 7, 8 and 9 summarize quantitative evaluation on all of the four environments with $N = 2$ robots, each subject to three different maximum velocity constraints. Results on all environments with $N = 4$ robots are in Table 10, 11, 12,and 13. In terms of constraint satisfaction, constraint-agnostic methods (vanilla DIFFUSER, MMD-CBS, and all coupling-only CD- variants) achieve similar rates: as low as 8–22% in `Empty` and between around 28% and 62% in `Highways`. In contrast, every projection-based variant (DIFFUSER with projection and our PCD-LB/SHD) enforces the constraint in all cases, confirming the effectiveness of projection. Inter-agent safety scores show the similar trend. Because MMD-CBS repeatedly samples and then stitches together *one* optimal trajectory (see *Remark 4*), it unsurprisingly attains perfect score. Among the one-shot methods, PCD- and CD- methods can almost match this upper bound, while vanilla DIFFUSER and and its projected variant performs much worse. Slightly degraded data adherence performance is again observed in our method: the LB cost function gets affected more due to its steeper gradients by design. Similar trends can be observed in the other two environments. Overall, the results show that PCD effectively promotes inter-robot collision avoidance through appropriate coupling costs, while enforcing hard test-time velocity constraints. A tradeoff exists between coupling strength and data adherence, depending on the coupling cost.

*Remark* 4 (Comparison to MMD-CBS). A direct comparison between MMD-CBS (Shaoul et al., 2025) and our method is *not* straightforward due to fundamental differences in their sampling procedures. MMD-CBS is a search-based approach that generates a batch of trajectories using diffusion models, *selects the best one* based on a cost function, *adopts only a partial segment* of the selected trajectory to resolve collisions, and repeats this process iteratively. As a result, it produces only *one trajectory* per robot per forward pass, making it inefficient for generating multiple i.i.d. samples. In contrast, DIFFUSER and our method produce a full batch of i.i.d. trajectories in a single pass, which preserves the benefit of massive parallel sampling from generative models. Thus, directly comparing

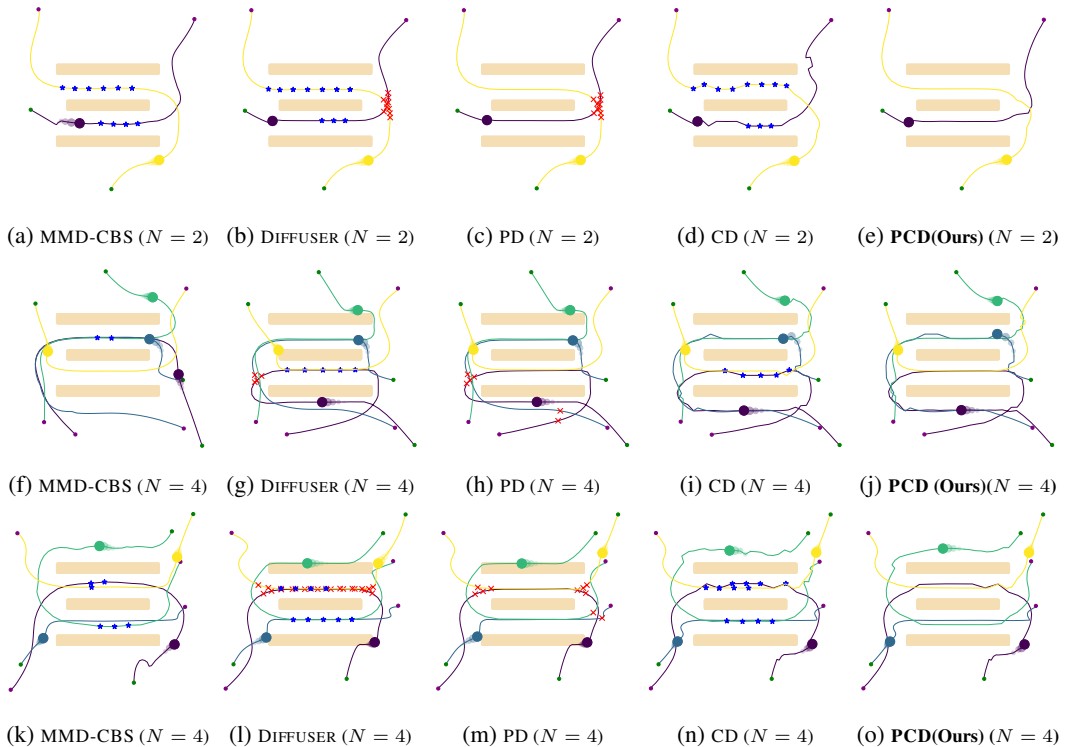

Figure 15: Robot trajectories in environment `Conveyor` generated by the compared methods with $N = 2$ and $N = 4$ robots running. Red crosses mark collisions and blue stars mark velocity constraint violations. Each row corresponds to one initial configuration.

a 128-sample batch from DIFFUSER or ours against a single *best* output from MMD-CBS, or *vice versa*, would *not* be very meaningful.

### C.3.3 ABLATION STUDY

We perform ablation study on the coupling strength $\gamma$ for both tasks `Empty` and `Highways` with the velocity limits reported in Table 1, but with both 2 and 4 robots. Results are presented in Figure 17 and Figure 18. In task `Empty`, as the coupling strength $\gamma$ increases, SU and RS increases and saturated near 1.0, with an exception in $N = 4$ where RS drops a bit when $\gamma$ is too high; CS remains at 100% by projection, and DA in general gradually drops. Results of `Highways` demonstrates the similar trends, with the difference where SU drops significantly for the LB cost when $\gamma$ is large. This is because SU also takes into accounts collisions with *static obstacles*. When $\gamma$ is high, the gradient of the robot-collision cost overwhelms that of the obstacle-avoidance term[13], resulting in the robots bumping into obstacles. This is supported by Figure 19 in which we report the decreasing obstacle safe rate, suggesting more trajectories are hitting static obstacles as $\gamma$ increases. A potential remedy is to leverage projection to also enforce static obstacle avoidance as Christopher et al. (2024), but that would introduce non-convexity into the projection process and might cause convergence issues.

### C.4 CONSTRAINED DIVERSE ROBOTIC MANIPULATION

### C.4.1 QUALITATIVE RESULTS

We present in Figure 20 more comparative results on the trajectories by all compared methods. Sheer contrast between our PCD method and others highlights the efficacy of our framework in jointly generating correlated samples while enforcing hard constraints.

---

[13]The coefficient for the gradient of the obstacle-avoidance cost is *fixed* in our experiments. See Appendix B.2 for details.

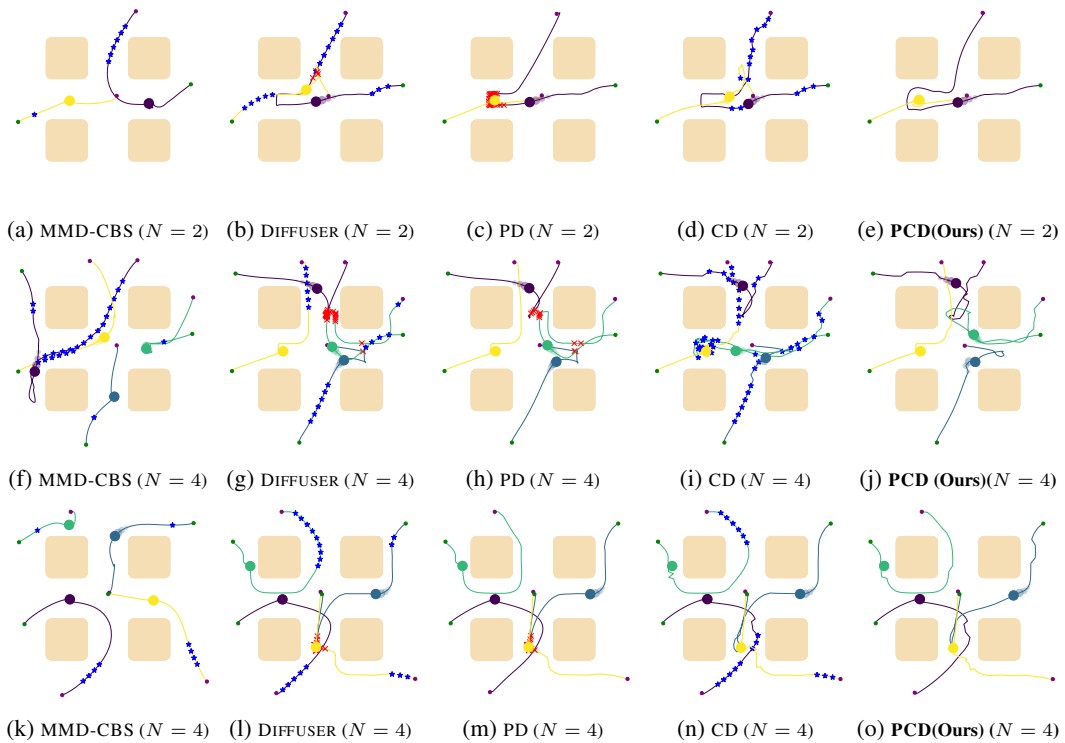

Figure 16: Robot trajectories in environment `DropRegion` generated by the compared methods with $N = 2$ and $N = 4$ robots running. Red crosses mark collisions and blue stars mark velocity constraint violations. Each row corresponds to one initial configuration.

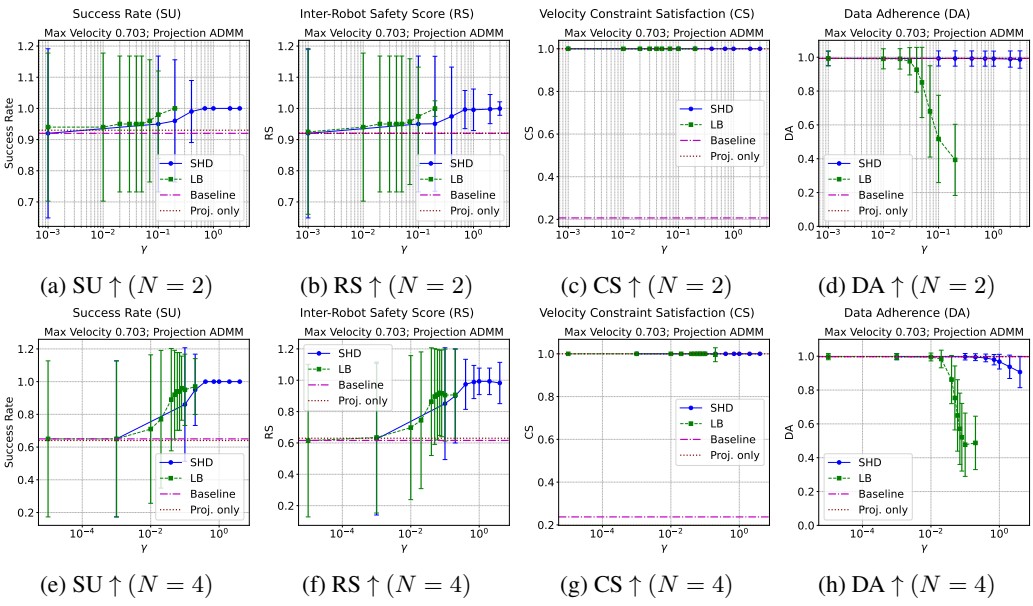

Figure 17: Coupling Strength Ablation of **PCD** on task **Empty** with velocity limit $v_{\max} = 0.703$. (a,b,c,d) $N = 2$ robots; (e,f,g,h) $N = 4$ robots.

| **Task Empty**, **2** Robots | | | | |
|---|---|---|---|---|
| METHOD \ Metric | SU(%)↑ | RS↑ | CS(%)↑ | DA↑ |
| **Max Vel. = 0.703** | | | | |
| vanilla DIFFUSER | $92_{\pm 27}$ | $0.92_{\pm 0.27}$ | $(59_{\pm 49}, 65_{\pm 48})$ | $(0.99_{\pm 0.085}, 1.0_{\pm 0.0})$ |
| MMD-CBS | $100_{\pm 0}$ | $1.0_{\pm 0.0}$ | $(8.0_{\pm 27}, 12_{\pm 32})$ | $(0.99_{\pm 0.073}, 1.0_{\pm 0.0})$ |
| DIFFUSER + projection | $92_{\pm 27}$ | $0.92_{\pm 0.27}$ | $(100_{\pm 0}, 100_{\pm 0})$ | $(0.99_{\pm 0.085}, 1.0_{\pm 0.0})$ |
| CD-LB (w/o proj.) | $100_{\pm 0}$ | $1.0_{\pm 0.065}$ | $(7.3_{\pm 26}, 5.5_{\pm 23})$ | $(0.92_{\pm 0.2}, 0.93_{\pm 0.18})$ |
| CD-SHD (w/o proj.) | $100_{\pm 0}$ | $1.0_{\pm 0.0}$ | $(45_{\pm 50}, 52_{\pm 50})$ | $(0.99_{\pm 0.09}, 1.0_{\pm 0.0062})$ |
| PCD-LB | $95_{\pm 22}$ | $0.95_{\pm 0.22}$ | $(100_{\pm 0}, 100_{\pm 0})$ | $(0.98_{\pm 0.12}, 1.0_{\pm 0.0087})$ |
| PCD-SHD | $100_{\pm 0}$ | $1.0_{\pm 0.061}$ | $(100_{\pm 0}, 100_{\pm 0})$ | $(0.99_{\pm 0.089}, 1.0_{\pm 0.0092})$ |
| **Max Vel. = 0.692** | | | | |
| vanilla DIFFUSER | $92_{\pm 27}$ | $0.92_{\pm 0.27}$ | $(40_{\pm 49}, 44_{\pm 50})$ | $(0.99_{\pm 0.085}, 1.0_{\pm 0.0})$ |
| MMD-CBS | $100_{\pm 0}$ | $1.0_{\pm 0.0}$ | $(8.0_{\pm 27}, 12_{\pm 32})$ | $(0.99_{\pm 0.073}, 1.0_{\pm 0.0})$ |
| DIFFUSER + projection | $93_{\pm 26}$ | $0.92_{\pm 0.27}$ | $(100_{\pm 0}, 100_{\pm 0})$ | $(0.99_{\pm 0.085}, 1.0_{\pm 0.0})$ |
| CD-LB (w/o proj.) | $100_{\pm 0}$ | $1.0_{\pm 0.065}$ | $(6.9_{\pm 25}, 4.6_{\pm 21})$ | $(0.92_{\pm 0.2}, 0.93_{\pm 0.18})$ |
| CD-SHD (w/o proj.) | $100_{\pm 0}$ | $1.0_{\pm 0.0}$ | $(31_{\pm 46}, 37_{\pm 48})$ | $(0.99_{\pm 0.09}, 1.0_{\pm 0.0062})$ |
| PCD-LB | $95_{\pm 22}$ | $0.95_{\pm 0.22}$ | $(100_{\pm 0}, 100_{\pm 0})$ | $(0.98_{\pm 0.12}, 1.0_{\pm 0.0088})$ |
| PCD-SHD | $100_{\pm 0}$ | $1_{\pm 0.065}$ | $(100_{\pm 0}, 100_{\pm 0})$ | $(0.98_{\pm 0.088}, 1.0_{\pm 0.012})$ |
| **Max Vel. = 0.675** | | | | |
| vanilla DIFFUSER | $92_{\pm 27}$ | $0.92_{\pm 0.27}$ | $(19_{\pm 40}, 22_{\pm 41})$ | $(0.99_{\pm 0.085}, 1.0_{\pm 0.0})$ |
| MMD-CBS | $100_{\pm 0}$ | $1.0_{\pm 0.0}$ | $(8.0_{\pm 27}, 12_{\pm 32})$ | $(0.99_{\pm 0.073}, 1.0_{\pm 0.0})$ |
| DIFFUSER + projection | $93_{\pm 26}$ | $0.92_{\pm 0.27}$ | $(100_{\pm 0}, 100_{\pm 0})$ | $(0.99_{\pm 0.085}, 1.0_{\pm 0})$ |
| CD-LB (w/o proj.) | $100_{\pm 0}$ | $1_{\pm 0.065}$ | $(6.45_{\pm 25}, 3.6_{\pm 19})$ | $(0.92_{\pm 0.2}, 0.93_{\pm 0.18})$ |
| CD-SHD (w/o proj.) | $100_{\pm 0}$ | $1.0_{\pm 0.0}$ | $(18.4_{\pm 39}, 21.3_{\pm 41})$ | $(0.99_{\pm 0.09}, 1.0_{\pm 0.0062})$ |
| PCD-LB | $95_{\pm 22}$ | $0.95_{\pm 0.22}$ | $(100_{\pm 0}, 100_{\pm 0})$ | $(0.92_{\pm 0.2}, 0.93_{\pm 0.17})$ |
| PCD-SHD | $100_{\pm 0}$ | $1.0_{\pm 0.069}$ | $(100_{\pm 0}, 100_{\pm 0})$ | $(0.99_{\pm 0.088}, 1.0_{\pm 0.012})$ |

Table 6: Task Empty, 2 robots, 100 random tests, sample size 128 except MMD-CBS.

### C.4.2 QUANTITATIVE RESULTS AT DIFFERENT VELOCITY LIMITS

Table 14 summarizes quantitative results of all compared methods with all three velocity limits. We also report standard deviations in the tables.

### C.4.3 ABLATION STUDY ON COUPLING STRENGTH

We perform ablation study investigating how the coupling strength $\gamma$ affects the performance of our PCD method. Figure 21 shows the trends of evaluation metrics' (DTW, DFD, CS, TC) change as the coupling strength parameter $\gamma$ increase, with all three velocity limits we experimented. Above all, the general trends look expected and similar across the three velocity limits: as we increase $\gamma$, the DTW and DFD monotonically increase, velocity constraint satisfaction (CS) is maintained perfect by design, and task completion score (TC) drops monotonically. This suggests that (i) the projection can guarantee velocity constraint regardless of coupling strength; (ii) as $\gamma$ increases, the correlations between the variables get stronger, with the cost gradients gradually overwhelming the learned score and thus deviating from the original data distribution. A tradeoff exists between data adherence and correlation strength.

A similar ablation is also conducted on coupling-only method (CD) by removing the projection; results are in Figure 22. The same tradeoff between correlations and data adherence also exists. Unsurprisingly the velocity constraint satisfaction rates drop as $\gamma$ increases, and the LB cost function is more sensitive.

| METHOD \ Metric | SU(%)↑ | RS↑ | CS(%)↑ | DA↑ |
|---|---|---|---|---|
| **Task `Highways`, 2 Robots** | | | | |
| **Max Vel. = 0.878** | | | | |
| vanilla DIFFUSER | $93_{\pm26}$ | $0.81_{\pm0.39}$ | $(72_{\pm45}, 76_{\pm43})$ | $(0.99_{\pm0.11}, 0.98_{\pm0.13})$ |
| MMD-CBS | $100_{\pm0}$ | $1.0_{\pm0.0}$ | $(73_{\pm44}, 79_{\pm41})$ | $(0.99_{\pm0.099}, 0.97_{\pm0.17})$ |
| DIFFUSER + projection | $92_{\pm27}$ | $0.81_{\pm0.39}$ | $(100_{\pm0}, 100_{\pm0})$ | $(0.99_{\pm0.12}, 0.98_{\pm0.14})$ |
| CD-LB (w/o proj.) | $100_{\pm0}$ | $1.0_{\pm0.033}$ | $(22_{\pm42}, 18_{\pm38})$ | $(0.99_{\pm0.11}, 0.98_{\pm0.15})$ |
| CD-SHD (w/o proj.) | $100_{\pm0}$ | $1.0_{\pm0.029}$ | $(65_{\pm48}, 66_{\pm47})$ | $(0.99_{\pm0.12}, 0.98_{\pm0.13})$ |
| PCD-LB | $100_{\pm0}$ | $0.99_{\pm0.1}$ | $(100_{\pm0}, 100_{\pm0})$ | $(0.98_{\pm0.14}, 0.97_{\pm0.18})$ |
| PCD-SHD | $100_{\pm0}$ | $1.0_{\pm0.012}$ | $(100_{\pm0}, 100_{\pm0})$ | $(0.98_{\pm0.14}, 0.97_{\pm0.16})$ |
| **Max Vel. = 0.781** | | | | |
| vanilla DIFFUSER | $93_{\pm26}$ | $0.81_{\pm0.39}$ | $(63_{\pm48}, 65_{\pm48})$ | $(0.99_{\pm0.11}, 0.98_{\pm0.13})$ |
| MMD-CBS | $100_{\pm0}$ | $1.0_{\pm0.0}$ | $(67_{\pm47}, 67_{\pm47})$ | $(0.99_{\pm0.099}, 0.97_{\pm0.17})$ |
| DIFFUSER + projection | $95_{\pm22}$ | $0.81_{\pm0.39}$ | $(100_{\pm0}, 100_{\pm0})$ | $(0.98_{\pm0.12}, 0.98_{\pm0.14})$ |
| CD-LB (w/o proj.) | $100_{\pm0}$ | $1.0_{\pm0.033}$ | $(14_{\pm35}, 12_{\pm32})$ | $(0.99_{\pm0.11}, 0.98_{\pm0.15})$ |
| CD-SHD (w/o proj.) | $100_{\pm0}$ | $1.0_{\pm0.0}$ | $(55_{\pm50}, 53_{\pm50})$ | $(0.99_{\pm0.11}, 0.98_{\pm0.13})$ |
| PCD-LB | $100_{\pm0}$ | $0.99_{\pm0.11}$ | $(100_{\pm0}, 100_{\pm0})$ | $(0.98_{\pm0.14}, 0.96_{\pm0.19})$ |
| PCD-SHD | $100_{\pm0}$ | $1.0_{\pm0.041}$ | $(100_{\pm0}, 100_{\pm0})$ | $(0.98_{\pm0.14}, 0.97_{\pm0.16})$ |
| **Max Vel. = 0.647** | | | | |
| vanilla DIFFUSER | $93_{\pm26}$ | $0.81_{\pm0.39}$ | $(52_{\pm50}, 50_{\pm50})$ | $(0.99_{\pm0.11}, 0.98_{\pm0.13})$ |
| MMD-CBS | $100_{\pm0}$ | $1.0_{\pm0.0}$ | $(62_{\pm49}, 54_{\pm50})$ | $(0.99_{\pm0.099}, 0.97_{\pm0.17})$ |
| DIFFUSER + projection | $91_{\pm29}$ | $0.84_{\pm0.36}$ | $(100_{\pm0}, 100_{\pm0})$ | $(0.98_{\pm0.13}, 0.98_{\pm0.14})$ |
| CD-LB (w/o proj.) | $100_{\pm0}$ | $0.99_{\pm0.12}$ | $(29_{\pm45}, 22_{\pm41})$ | $(0.99_{\pm0.11}, 0.98_{\pm0.16})$ |
| CD-SHD (w/o proj.) | $100_{\pm0}$ | $1.0_{\pm0.029}$ | $(49_{\pm50}, 46_{\pm50})$ | $(0.99_{\pm0.12}, 0.98_{\pm0.13})$ |
| PCD-LB | $99_{\pm9.9}$ | $0.98_{\pm0.14}$ | $(100_{\pm0}, 100_{\pm0})$ | $(0.98_{\pm0.15}, 0.96_{\pm0.19})$ |
| PCD-SHD | $100_{\pm0}$ | $1.0_{\pm0.022}$ | $(100_{\pm0}, 100_{\pm0})$ | $(0.98_{\pm0.15}, 0.97_{\pm0.17})$ |

Table 7: Task `Highways`, 2 robots, 100 random tests, sample size 128 except MMD-CBS.

## C.5 ABLATION STUDY ON SAMPLING STEPS

We perform ablation study to explore the effect of the number of sample steps on the `PushT` task.

**Setup** We use a pretrained DDPM model that was originally trained with $T = 100$ denoising diffusion steps[14], and vary the number of denoising steps *during sampling* with $T \in \{15, 20, 30, 40, 50, 75, 100\}$ while keeping the same noise schedule. We set the velocity limit as $v_{\max} = 8.4$, and we use the same coupling strengths as those for obtaining the results reported in Table 2. For each method, we run it on 10 uniformly random initial conditions and generate 50 *pairs* of full trajectories. We report the same four metrics as in Section 5.2: DTW, DFD, CS, and TC.

**Results** The performance against number of sampling steps curves are shown in Figure 23. For the baseline Diffusion Policy, diversity metrics (DTW and DFD) appear to improve as $T$ decreases. However, this trend is mainly an artifact: when $T < 40$, the trajectory sample quality deteriorates significantly (*e.g.*, with jerky and teleporting motions), which is also reflected in degraded task completion score (TC) and rapidly dropping velocity constraint satisfaction rates (CS). Such low-quality samples often fail the manipulation task, making comparisons for $T < 40$ unreliable. Therefore, we shall focus on meaningful comparisons only for $T \geq 40$. Surprisingly, for all methods the TC scores do not peak at the highest number of sampling steps, even for the baseline. When the number of sampling steps $T \geq 40$, the *improvements* in diversity performance *over the baseline* grow steadily as $T$ increases. This pattern can be partly explained by the coupling effect benefiting from more

---

[14]This is also a pretrained model from Feng et al. (2024), but is *not* the one we used in our main experiment. We use this model only for this ablation.

| METHOD \ Metric | SU(%)↑ | RS↑ | CS(%)↑ | DA↑ |
|---|---|---|---|---|
| **Task `Conveyor`, 2 Robots** | | | | |
| **Max Vel. = 1.21** | | | | |
| vanilla DIFFUSER | $14_{\pm35}$ | $0.64_{\pm0.48}$ | $(0.19_{\pm4.3}, 0.45_{\pm6.7})$ | $(0.95_{\pm0.21}, 0.97_{\pm0.18})$ |
| MMD-CBS | $86_{\pm35}$ | $1.0_{\pm0.0}$ | $(0.0_{\pm0.0}, 1.0_{\pm9.9})$ | $(1.0_{\pm0.0}, 0.98_{\pm0.14})$ |
| DIFFUSER + projection | $10_{\pm30}$ | $0.68_{\pm0.47}$ | $(100_{\pm0}, 100_{\pm0})$ | $(0.71_{\pm0.46}, 0.70_{\pm0.46})$ |
| CD-LB (w/o proj.) | $100_{\pm0}$ | $0.96_{\pm0.2}$ | $(0.039_{\pm2.0}, 0.42_{\pm6.5})$ | $(0.99_{\pm0.11}, 0.98_{\pm0.13})$ |
| CD-SHD (w/o proj.) | $100_{\pm0}$ | $1.0_{\pm0.0}$ | $(0.0078_{\pm0.88}, 0.2_{\pm4.4})$ | $(0.99_{\pm0.093}, 0.99_{\pm0.098})$ |
| PCD-LB | $100_{\pm0}$ | $0.94_{\pm0.24}$ | $(100_{\pm0}, 100_{\pm0})$ | $(0.85_{\pm0.35}, 0.85_{\pm0.36})$ |
| PCD-SHD | $100_{\pm0}$ | $0.99_{\pm0.11}$ | $(100_{\pm0}, 100_{\pm0})$ | $(0.90_{\pm0.30}, 0.91_{\pm0.28})$ |
| **Max Vel. = 1.46** | | | | |
| vanilla DIFFUSER | $14_{\pm35}$ | $0.64_{\pm0.48}$ | $(2.4_{\pm15}, 1.7_{\pm13})$ | $(0.95_{\pm0.21}, 0.97_{\pm0.18})$ |
| MMD-CBS | $86_{\pm35}$ | $1.0_{\pm0.0}$ | $(4.0_{\pm20}, 5.0_{\pm22})$ | $(1.0_{\pm0.0}, 0.98_{\pm0.14})$ |
| DIFFUSER + projection | $14_{\pm35}$ | $0.67_{\pm0.47}$ | $(100_{\pm0}, 100_{\pm0})$ | $(0.87_{\pm0.33}, 0.89_{\pm0.31})$ |
| CD-LB (w/o proj.) | $100_{\pm0}$ | $0.96_{\pm0.20}$ | $(0.14_{\pm3.7}, 0.75_{\pm8.6})$ | $(0.99_{\pm0.11}, 0.98_{\pm0.13})$ |
| CD-SHD (w/o proj.) | $100_{\pm0}$ | $1.0_{\pm0.0}$ | $(0.80_{\pm8.9}, 0.82_{\pm9})$ | $(0.99_{\pm0.093}, 0.99_{\pm0.098})$ |
| PCD-LB | $100_{\pm0}$ | $0.95_{\pm0.23}$ | $(100_{\pm0}, 100_{\pm0})$ | $(0.95_{\pm0.23}, 0.94_{\pm0.24})$ |
| PCD-SHD | $100_{\pm0}$ | $0.99_{\pm0.073}$ | $(100_{\pm0}, 100_{\pm0})$ | $(0.98_{\pm0.15}, 0.98_{\pm0.14})$ |
| **Max Vel. = 1.76** | | | | |
| vanilla DIFFUSER | $14_{\pm35}$ | $0.64_{\pm0.48}$ | $(36_{\pm48}, 33_{\pm47})$ | $(0.95_{\pm0.21}, 0.97_{\pm0.18})$ |
| MMD-CBS | $86_{\pm35}$ | $1.0_{\pm0.0}$ | $(48_{\pm50}, 50_{\pm50})$ | $(1.0_{\pm0.0}, 0.98_{\pm0.14})$ |
| DIFFUSER + projection | $17_{\pm38}$ | $0.65_{\pm0.48}$ | $(100_{\pm0}, 100_{\pm0})$ | $(0.94_{\pm0.23}, 0.96_{\pm0.20})$ |
| CD-LB (w/o proj.) | $100_{\pm0}$ | $0.96_{\pm0.2}$ | $(5.4_{\pm23}, 4.4_{\pm21})$ | $(0.99_{\pm0.11}, 0.98_{\pm0.13})$ |
| CD-SHD (w/o proj.) | $100_{\pm0}$ | $1.0_{\pm0.0}$ | $(22_{\pm41}, 20_{\pm40})$ | $(0.99_{\pm0.093}, 0.99_{\pm0.098})$ |
| PCD-LB | $100_{\pm0}$ | $0.96_{\pm0.21}$ | $(100_{\pm0}, 100_{\pm0})$ | $(0.98_{\pm0.14}, 0.97_{\pm0.18})$ |
| PCD-SHD | $100_{\pm0}$ | $1.0_{\pm0.037}$ | $(100_{\pm0}, 100_{\pm0})$ | $(0.99_{\pm0.093}, 0.99_{\pm0.10})$ |

Table 8: Task `Conveyor`, 2 robots, 100 random tests, sample size 128 except MMD-CBS.

accurate Tweedie's estimates used in the PS cost variants at later denoising steps, as a higher total number of sampling steps leads to a greater number of steps with more accurate Tweedie's estimates.

## C.6 CONSTRAINED COUPLED IMAGE PAIR GENERATION

### C.6.1 ADDITIONAL QUALITATIVE RESULTS

Figure 24 presents additional pairs generated by PCD when *two* exemplar images (Figure 4h) define the convex hull. We repeat the experiment with *six* exemplars per model (Figure 25); the corresponding samples are shown in Figure 26.

### C.6.2 ADDITIONAL QUANTITATIVE RESULTS

Table 15 reports the full set of metrics including the standard deviations for the samples generated using *two* exemplars per model, while Table 16 for the samples generated with *six* exemplars per model. Quantitatively, both setups show the same trend throughout, that is: Projection-based methods (SD+P, PCD) achieve 100% gender satisfaction (M/F) and the strongest alignment to exemplars (higher SE-CLIP, lower SE-LPIPS and SE-FID) compared to vanilla SD, coupling-only variants and other baselines. Projection reduces diversity (low IS-LPIPS); adding coupling partially recovers diversity relative to projection alone. Coupling-based methods improve the age-group XOR satisfaction, with PCD attaining the highest rate (96% for the setup using *two* exemplars and 76% for the setup using *six* exemplars). In both experiments, SyncDiffusion matches SD+C on XOR (48%) but shows slightly reduced sample variation (lower IS-LPIPS). We also observe that even with the use of a generic text prompt as outlined in Appendix **B.4**, ControlNet-XS exhibits a male-bias, consistently

| METHOD \ Metric | SU(%)↑ | RS↑ | CS(%)↑ | DA↑ |
|---|---|---|---|---|
| **Task `DropRegion`, 2 Robots** | | | | |
| **Max Vel. = 0.928** | | | | |
| vanilla DIFFUSER | $60_{\pm 49}$ | $0.61_{\pm 0.49}$ | $(11_{\pm 32}, 9.5_{\pm 29})$ | $(0.89_{\pm 0.32}, 0.88_{\pm 0.32})$ |
| MMD-CBS | $100_{\pm 0.0}$ | $1.0_{\pm 0.0}$ | $(22_{\pm 41}, 21_{\pm 41})$ | $(1.0_{\pm 0.0}, 1.0_{\pm 0.0})$ |
| DIFFUSER + projection | $63_{\pm 48}$ | $0.67_{\pm 0.47}$ | $(100_{\pm 0}, 100_{\pm 0})$ | $(0.92_{\pm 0.27}, 0.92_{\pm 0.27})$ |
| CD-LB (w/o proj.) | $100_{\pm 0}$ | $0.97_{\pm 0.18}$ | $(9.6_{\pm 30}, 9_{\pm 29})$ | $(0.59_{\pm 0.49}, 0.60_{\pm 0.49})$ |
| CD-SHD (w/o proj.) | $100_{\pm 0}$ | $1.0_{\pm 0.07}$ | $(7.2_{\pm 26}, 5.7_{\pm 23})$ | $(0.96_{\pm 0.19}, 0.96_{\pm 0.19})$ |
| PCD-LB | $100_{\pm 0}$ | $0.91_{\pm 0.29}$ | $(100_{\pm 0}, 100_{\pm 0})$ | $(0.86_{\pm 0.35}, 0.87_{\pm 0.34})$ |
| PCD-SHD | $100_{\pm 0}$ | $1.0_{\pm 0.044}$ | $(100_{\pm 0}, 100_{\pm 0})$ | $(0.97_{\pm 0.18}, 0.96_{\pm 0.18})$ |
| **Max Vel. = 1.13** | | | | |
| vanilla DIFFUSER | $60_{\pm 49}$ | $0.61_{\pm 0.49}$ | $(37_{\pm 48}, 35_{\pm 48})$ | $(0.89_{\pm 0.32}, 0.88_{\pm 0.32})$ |
| MMD-CBS | $100_{\pm 0.0}$ | $1.0_{\pm 0.0}$ | $(51_{\pm 50}, 58_{\pm 49})$ | $(1.0_{\pm 0.0}, 1.0_{\pm 0.0})$ |
| DIFFUSER + projection | $61_{\pm 49}$ | $0.65_{\pm 0.48}$ | $(100_{\pm 0}, 100_{\pm 0})$ | $(0.92_{\pm 0.28}, 0.91_{\pm 0.28})$ |
| CD-LB (w/o proj.) CD-LB | $100_{\pm 0}$ | $0.97_{\pm 0.18}$ | $(18_{\pm 39}, 19_{\pm 39})$ | $(0.59_{\pm 0.49}, 0.60_{\pm 0.49})$ |
| CD-SHD (w/o proj.) | $100_{\pm 0.0}$ | $1.0_{\pm 0.07}$ | $(24_{\pm 43}, 22_{\pm 42})$ | $(0.96_{\pm 0.19}, 0.96_{\pm 0.19})$ |
| PCD-LB | $100_{\pm 0.0}$ | $0.90_{\pm 0.30}$ | $(100_{\pm 0}, 100_{\pm 0})$ | $(0.85_{\pm 0.35}, 0.86_{\pm 0.35})$ |
| PCD-SHD | $100_{\pm 0.0}$ | $1.0_{\pm 0.042}$ | $(100_{\pm 0}, 100_{\pm 0})$ | $(0.96_{\pm 0.19}, 0.96_{\pm 0.18})$ |
| **Max Vel. = 1.34** | | | | |
| vanilla DIFFUSER | $60_{\pm 49}$ | $0.61_{\pm 0.49}$ | $(78_{\pm 42}, 73_{\pm 44})$ | $(0.89_{\pm 0.32}, 0.88_{\pm 0.32})$ |
| MMD-CBS | $100_{\pm 0.0}$ | $1.0_{\pm 0.0}$ | $(71_{\pm 45}, 81_{\pm 39})$ | $(1.0_{\pm 0.0}, 1.0_{\pm 0.0})$ |
| DIFFUSER + projection | $59_{\pm 49}$ | $0.63_{\pm 0.48}$ | $(100_{\pm 0}, 100_{\pm 0})$ | $(0.91_{\pm 0.29}, 0.90_{\pm 0.30})$ |
| CD-LB (w/o proj.) CD-LB | $100_{\pm 0}$ | $0.97_{\pm 0.18}$ | $(31_{\pm 46}, 33_{\pm 47})$ | $(0.59_{\pm 0.49}, 0.60_{\pm 0.49})$ |
| CD-SHD (w/o proj.) | $100_{\pm 0.0}$ | $1.0_{\pm 0.07}$ | $(50_{\pm 50}, 48_{\pm 50})$ | $(0.96_{\pm 0.19}, 0.96_{\pm 0.19})$ |
| PCD-LB | $100_{\pm 0.0}$ | $0.90_{\pm 0.30}$ | $(100_{\pm 0}, 100_{\pm 0})$ | $(0.85_{\pm 0.36}, 0.85_{\pm 0.35})$ |
| PCD-SHD | $100_{\pm 0.0}$ | $1.0_{\pm 0.033}$ | $(100_{\pm 0}, 100_{\pm 0})$ | $(0.96_{\pm 0.19}, 0.96_{\pm 0.19})$ |

Table 9: Task `DropRegion`, 2 robots, 100 random tests, sample size 128 except MMD-CBS.

producing more male samples, thus yielding 100% male gender satisfaction (M) and generally lower female gender satisfaction (F) across both runs.

### C.6.3 ABLATION STUDY

We ablate both the coupling strength $\gamma$ and the noise-scaling factor $k$. For **PCD**, we vary $\gamma \in \{50, 150, 250, 350, 450, 500\}$ with $M \in \{2, 6\}$ exemplar images per model. Separately, for **SD+P** we vary $k \in \{1, 2, 5, 10, 20\}$ (again with $M \in \{2, 6\}$) to examine how amplifying the noise standard deviation impacts diversity and other metrics. Figures 27 and 28 summarize the trends for XOR%, M/F%, SE-CLIP, SE-LPIPS, SE-FID and IS-LPIPS as $\gamma$ and $k$ increase, respectively.

As $\gamma$ increases, we observe that XOR% increases monotonically, M/F% (gender constraint satisfaction) is maintained perfect by design, SE-CLIP maintains in place, SE-LPIPS and SE-FID drops almost monotonically, while IS-LPIPS increases. This suggests that (i) a larger $\gamma$ strengthens the coupling signal, yielding a clearer young/old contrast (higher XOR%), (ii) generated sample pairs resemble more of the exemplars provided (denoted by gradually decreasing SE-LPIPS and SE-FID) due to gradients from the coupling loss, (iii) stronger coupling also injects modest additional diversity (higher IS-LPIPS), and (iv) projection ensures the gender constraint is satisfied regardless of $\gamma$.

In the independent study of $k$, we observe that as $k$ increases, the trend across all metrics are very much similar to that of $\gamma$, with one notable difference: IS-LPIPS rises sharply and monotonically as $k$ grows. This suggests that increasing the noise scaling factor boosts sample variation, and yet does not violate the attribute constraints provided by the exemplars (perfect M/F%, high SE-CLIP, low SE-LPIPS and SE-FID).

| | | | Task **Empty**, 4 Robots | |
|---|---|---|---|---|
| METHOD \ Metric | SU(%)↑ | RS↑ | CS(%)↑ | DA↑ |
| | | | **Max Vel. = 0.703** | |
| vanilla DIFFUSER | $65_{\pm48}$ | $.62_{\pm.49}$ | $(69_{\pm46}, 65_{\pm48}, 64_{\pm48}, 62_{\pm48})$ | $(.99_{\pm.073}, 1._{\pm.00055}, 1._{\pm.0014}, 1._{\pm.0016})$ |
| MMD-CBS | $100_{\pm0}$ | $1._{\pm0.}$ | $(16_{\pm36}, 17_{\pm38}, 11_{\pm31}, 17_{\pm38})$ | $(.99_{\pm.044}, 1._{\pm.0035}, 1._{\pm.019}, 1._{\pm0.})$ |
| DIFFUSER + proj. | $65_{\pm48}$ | $.61_{\pm.49}$ | $(100_{\pm0}, 100_{\pm0}, 100_{\pm0}, 100_{\pm0})$ | $(.99_{\pm.073}, 1._{\pm.00055}, 1._{\pm.0014}, 1._{\pm.0016})$ |
| CD-LB (w/o proj.) | $100_{\pm0}$ | $.99_{\pm.085}$ | $(.031_{\pm1.8}, 1_{\pm9.9}, 3_{\pm17}, 3.1_{\pm17})$ | $(.81_{\pm.33}, .88_{\pm.26}, .86_{\pm.26}, .85_{\pm.28})$ |
| CD-SHD (w/o proj.) | $100_{\pm0}$ | $1_{\pm0}$ | $(38_{\pm49}, 38_{\pm48}, 35_{\pm48}, 37_{\pm48})$ | $(.99_{\pm.076}, 1._{\pm.001}, 1._{\pm.0085}, 1._{\pm.003})$ |
| PCD-LB | $96_{\pm20}$ | $.92_{\pm.28}$ | $(100_{\pm0}, 100_{\pm0}, 100_{\pm0}, 100_{\pm0})$ | $(.57_{\pm.38}, .51_{\pm.37}, .52_{\pm.37}, .49_{\pm.37})$ |
| PCD-SHD | $100_{\pm0}$ | $.99_{\pm.084}$ | $(100_{\pm0}, 100_{\pm0}, 100_{\pm0}, 100_{\pm0})$ | $(.96_{\pm.1}, .98_{\pm.046}, .97_{\pm.069}, .96_{\pm.078})$ |
| | | | **Max Vel. = 0.692** | |
| vanilla DIFFUSER | $65_{\pm48}$ | $.62_{\pm.49}$ | $(42_{\pm49}, 42_{\pm49}, 43_{\pm49}, 41_{\pm49})$ | $(.99_{\pm.073}, 1._{\pm.00055}, 1._{\pm.0014}, 1._{\pm.0016})$ |
| MMD-CBS | $100_{\pm0}$ | $1._{\pm0.}$ | $(15_{\pm36}, 17_{\pm38}, 11_{\pm31}, 16_{\pm37})$ | $(.99_{\pm.044}, 1._{\pm.0035}, 1._{\pm.019}, 1._{\pm0})$ |
| DIFFUSER + proj. | $64_{\pm48}$ | $.61_{\pm.49}$ | $(100_{\pm0}, 100_{\pm0}, 100_{\pm0}, 100_{\pm0})$ | $(.99_{\pm.073}, 1._{\pm.00053}, 1._{\pm.0014}, 1._{\pm.0016})$ |
| CD-LB (w/o proj.) | $100_{\pm0}$ | $.99_{\pm.085}$ | $(0_{\pm0}, 1_{\pm9.9}, 2.5_{\pm16}, 3.1_{\pm17})$ | $(.81_{\pm.33}, .88_{\pm.26}, .86_{\pm.26}, .85_{\pm.28})$ |
| CD-SHD (w/o proj.) | $100_{\pm0}$ | $1._{\pm0.}$ | $(28_{\pm45}, 29_{\pm45}, 27_{\pm44}, 27_{\pm45})$ | $(.99_{\pm.076}, 1._{\pm.001}, 1._{\pm.0085}, 1._{\pm.003})$ |
| PCD-LB | $94_{\pm24}$ | $.92_{\pm.28}$ | $(100_{\pm0}, 100_{\pm0}, 100_{\pm0}, 100_{\pm0})$ | $(.6_{\pm.39}, .56_{\pm.37}, .58_{\pm.38}, .55_{\pm.38})$ |
| PCD-SHD | $100_{\pm0}$ | $.99_{\pm.095}$ | $(100_{\pm0}, 100_{\pm0}, 100_{\pm0}, 100_{\pm0})$ | $(.96_{\pm.1}, .98_{\pm.045}, .97_{\pm.07}, .96_{\pm.078})$ |
| | | | **Max Vel. = 0.675** | |
| vanilla DIFFUSER | $65_{\pm48}$ | $.62_{\pm.49}$ | $(24_{\pm43}, 22_{\pm42}, 24_{\pm42}, 26_{\pm44})$ | $(.99_{\pm.073}, 1._{\pm.00055}, 1._{\pm.0014}, 1._{\pm.0016})$ |
| MMD-CBS | $100_{\pm0}$ | $1._{\pm0.}$ | $(15_{\pm36}, 17_{\pm38}, 11_{\pm31}, 14_{\pm35})$ | $(.99_{\pm.044}, 1._{\pm.0035}, 1._{\pm.019}, 1._{\pm0.})$ |
| DIFFUSER + proj. | $64_{\pm48}$ | $.63_{\pm.48}$ | $(100_{\pm0}, 100_{\pm0}, 100_{\pm0}, 100_{\pm0})$ | $(.99_{\pm.073}, 1._{\pm.00055}, 1._{\pm.0014}, 1._{\pm.0016})$ |
| CD-LB (w/o proj.) | $100_{\pm0.}$ | $.99_{\pm.085}$ | $(0._{\pm0.}, 1._{\pm9.9}, 1.7_{\pm13}, 3._{\pm17})$ | $(.81_{\pm.33}, .88_{\pm.26}, .86_{\pm.26}, .85_{\pm.28})$ |
| CD-SHD (w/o proj.) | $100_{\pm0}$ | $1._{\pm0.}$ | $(21_{\pm41}, 21_{\pm41}, 19_{\pm39}, 21_{\pm40})$ | $(.99_{\pm.076}, 1._{\pm.001}, 1._{\pm.0085}, 1._{\pm.003})$ |
| PCD-LB | $92_{\pm27}$ | $.88_{\pm.33}$ | $(100_{\pm0}, 100_{\pm0}, 100_{\pm0}, 100_{\pm0})$ | $(.73_{\pm.38}, .79_{\pm.33}, .77_{\pm.34}, .75_{\pm.34})$ |
| PCD-SHD | $100_{\pm0}$ | $.99_{\pm.099}$ | $(100_{\pm0}, 100_{\pm0}, 100_{\pm0}, 100_{\pm0})$ | $(.96_{\pm.1}, .98_{\pm.045}, .97_{\pm.069}, .96_{\pm.077})$ |

Table 10: Task `Empty`, 4 robots, 100 random tests, sample size 128 except MMD-CBS.

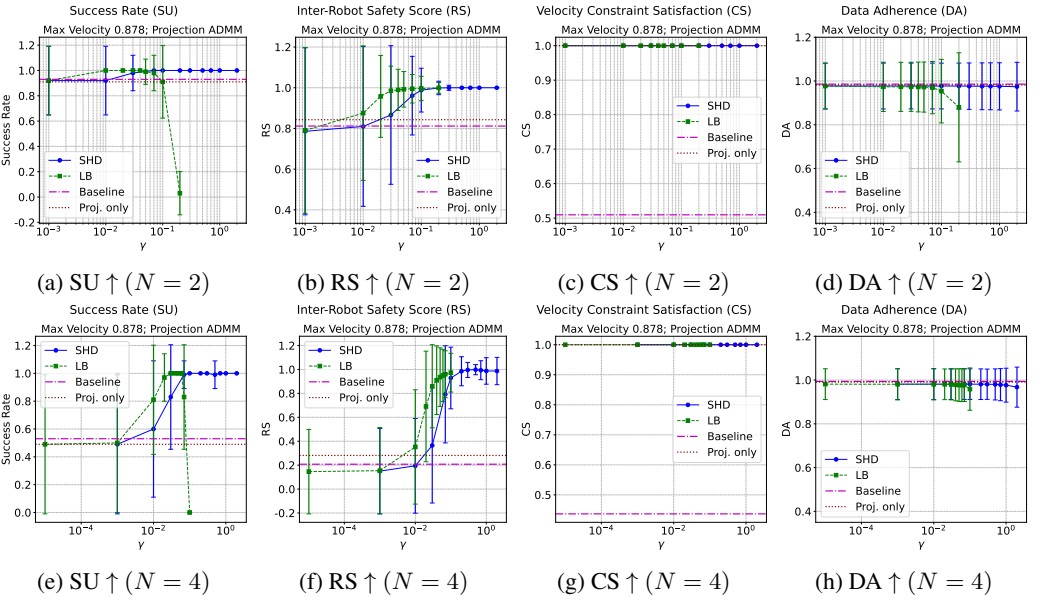

(a) SU ↑ ($N = 2$)  (b) RS ↑ ($N = 2$)  (c) CS ↑ ($N = 2$)  (d) DA ↑ ($N = 2$)

(e) SU ↑ ($N = 4$)  (f) RS ↑ ($N = 4$)  (g) CS ↑ ($N = 4$)  (h) DA ↑ ($N = 4$)

Figure 18: Coupling Strength Ablation of **PCD** on task **Highways** with velocity limit $v_{\max} = 0.878$. (a,b,c,d) $N = 2$ robots; (e,f,g,h) $N = 4$ robots.

| METHOD \ Metric | SU(%)↑ | RS↑ | CS(%)↑ | DA↑ |
|---|---|---|---|---|
| **Task `Highways`, 4 Robots** | | | | |
| | | | **Max Vel. = 0.878** | |
| vanilla DIFFUSER | $53_{\pm50}$ | $.21_{\pm.41}$ | $(73_{\pm44}, 68_{\pm47}, 68_{\pm47}, 67_{\pm47})$ | $(1._{\pm.04}, .98_{\pm.14}, 1._{\pm.04}, 1._{\pm.028})$ |
| MMD-CBS | $100_{\pm0}$ | $1._{\pm0.}$ | $(68_{\pm47}, 64_{\pm48}, 63_{\pm48}, 65_{\pm48})$ | $(.98_{\pm.14}, .96_{\pm.20}, .97_{\pm.17}, .99_{\pm.10})$ |
| DIFFUSER + proj. | $54_{\pm50}$ | $.21_{\pm.41}$ | $(100_{\pm0}, 100_{\pm0}, 100_{\pm0}, 100_{\pm0})$ | $(1._{\pm.052}, .98_{\pm.15}, 1._{\pm.07}, 1._{\pm.047})$ |
| CD-LB (w/o proj.) | $100_{\pm0}$ | $1._{\pm.032}$ | $(0_{\pm0}, .11_{\pm3.3}, 0._{\pm0.}, 0._{\pm0.})$ | $(.99_{\pm.12}, .99_{\pm.11}, 1._{\pm.067}, 1._{\pm.066})$ |
| CD-SHD (w/o proj.) | $100_{\pm0}$ | $1._{\pm.012}$ | $(35_{\pm48}, 35_{\pm48}, 36_{\pm48}, 35_{\pm48})$ | $(1._{\pm.07}, .98_{\pm.15}, 1._{\pm.07}, 1._{\pm.044})$ |
| PCD-LB | $100_{\pm0}$ | $.95_{\pm.22}$ | $(100_{\pm0}, 100_{\pm0}, 100_{\pm0}, 100_{\pm0})$ | $(.97_{\pm.16}, .96_{\pm.2}, .99_{\pm.12}, .99_{\pm.098})$ |
| PCD-SHD | $100_{\pm0}$ | $1._{\pm.063}$ | $(100_{\pm0}, 100_{\pm0}, 100_{\pm0}, 100_{\pm0})$ | $(.98_{\pm.13}, .96_{\pm.19}, .99_{\pm.12}, .99_{\pm.076})$ |
| | | | **Max Vel. = 0.781** | |
| vanilla DIFFUSER | $53_{\pm50}$ | $.21_{\pm.41}$ | $(62_{\pm49}, 57_{\pm49}, 58_{\pm49}, 57_{\pm50})$ | $(1._{\pm.04}, .98_{\pm.14}, 1._{\pm.04}, 1._{\pm.028})$ |
| MMD-CBS | $100_{\pm0}$ | $1._{\pm0.}$ | $(58_{\pm49}, 51_{\pm50}, 58_{\pm49}, 59_{\pm49})$ | $(.98_{\pm.14}, .96_{\pm.20}, .97_{\pm.17}, .99_{\pm.10})$ |
| DIFFUSER + proj. | $53_{\pm50}$ | $.22_{\pm.42}$ | $(100_{\pm0}, 100_{\pm0}, 100_{\pm0}, 100_{\pm0})$ | $(1._{\pm.054}, .98_{\pm.15}, .99_{\pm.075}, 1._{\pm.048})$ |
| CD-LB (w/o proj.) | $100_{\pm0}$ | $1._{\pm.069}$ | $(0._{\pm0.}, .031_{\pm1.8}, .07_{\pm2.7}, .0078_{\pm.88})$ | $(.99_{\pm.11}, .98_{\pm.13}, 1._{\pm.067}, 1._{\pm.062})$ |
| CD-SHD (w/o proj.) | $100_{\pm0}$ | $1._{\pm.051}$ | $(49_{\pm50}, 42_{\pm49}, 45_{\pm50}, 43_{\pm50})$ | $(.99_{\pm.072}, .98_{\pm.15}, 1._{\pm.071}, 1._{\pm.048})$ |
| PCD-LB | $100_{\pm0}$ | $.91_{\pm.28}$ | $(100_{\pm0}, 100_{\pm0}, 100_{\pm0}, 100_{\pm0})$ | $(.98_{\pm.15}, .95_{\pm.21}, .99_{\pm.12}, .99_{\pm.093})$ |
| PCD-SHD | $100_{\pm0}$ | $1._{\pm.064}$ | $(100_{\pm0}, 100_{\pm0}, 100_{\pm0}, 100_{\pm0})$ | $(.98_{\pm.14}, .96_{\pm.19}, .98_{\pm.12}, .99_{\pm.082})$ |
| | | | **Max Vel. = 0.647** | |
| vanilla DIFFUSER | $53_{\pm50}$ | $.21_{\pm.41}$ | $(46_{\pm50}, 43.5_{\pm50}, 39.2_{\pm49}, 45.9_{\pm50})$ | $(1._{\pm.04}, .98_{\pm.14}, 1._{\pm.04}, 1._{\pm.028})$ |
| MMD-CBS | $100_{\pm0}$ | $1._{\pm0.}$ | $(48_{\pm50}, 41_{\pm49}, 46_{\pm50}, 51_{\pm50})$ | $(.98_{\pm.14}, .96_{\pm.20}, .97_{\pm.17}, .99_{\pm.10})$ |
| DIFFUSER + proj. | $49_{\pm50}$ | $.28_{\pm.45}$ | $(100_{\pm0}, 100_{\pm0}, 100_{\pm0}, 100_{\pm0})$ | $(1._{\pm.059}, .98_{\pm.15}, .99_{\pm.084}, 1._{\pm.052})$ |
| CD-LB (w/o proj.) | $100_{\pm0}$ | $.97_{\pm.17}$ | $(.07_{\pm2.7}, .21_{\pm4.6}, .27_{\pm5.1}, .055_{\pm2.3})$ | $(.99_{\pm.11}, .97_{\pm.16}, 1._{\pm.068}, 1._{\pm.061})$ |
| CD-SHD (w/o proj.) | $100_{\pm0}$ | $1._{\pm.012}$ | $(18_{\pm39}, 16_{\pm37}, 22_{\pm41}, 21_{\pm41})$ | $(1._{\pm.07}, .98_{\pm.15}, 1._{\pm.07}, 1._{\pm.044})$ |
| PCD-LB | $87_{\pm34}$ | $.92_{\pm.27}$ | $(100_{\pm0}, 100_{\pm0}, 100_{\pm0}, 100_{\pm0})$ | $(.98_{\pm.15}, .95_{\pm.22}, .98_{\pm.13}, .99_{\pm.099})$ |
| PCD-SHD | $100_{\pm0}$ | $.99_{\pm.1}$ | $(100_{\pm0}, 100_{\pm0}, 100_{\pm0}, 100_{\pm0})$ | $(.96_{\pm.2}, .93_{\pm.25}, .97_{\pm.18}, .97_{\pm.17})$ |

Table 11: Task `Highways`, 4 robots, 100 random tests, sample size 128 except MMD-CBS.

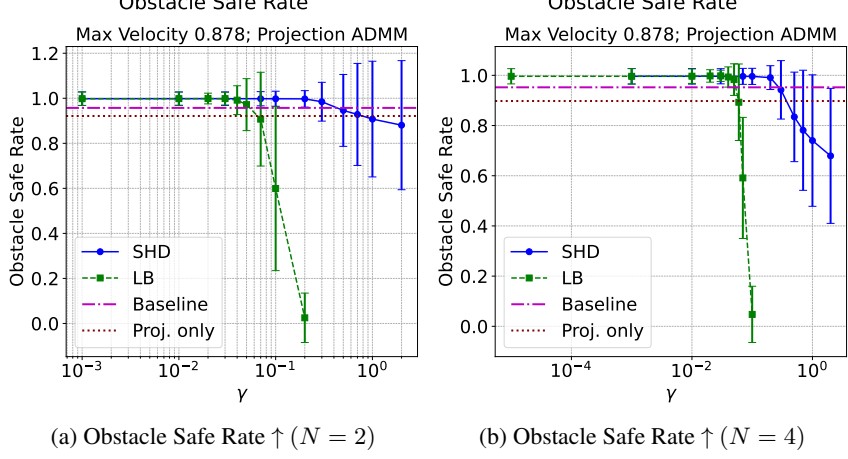

(a) Obstacle Safe Rate ↑ $(N = 2)$      (b) Obstacle Safe Rate ↑ $(N = 4)$

Figure 19: Obstacle safe rates for the coupling strength ablation of **PCD** on task **`Highways`** with velocity limit $v_{\max} = 0.878$. The obstacle safe rate is defined as the average of an indicator of whether all trajectories are *not* colliding into static obstacles.

| Task **Conveyor**, **4** Robots | | | | |
|---|---|---|---|---|
| METHOD \ Metric | SU(%)↑ | RS↑ | CS(%)↑ | DA↑ |
| **Max Vel. = 1.21** | | | | |
| vanilla DIFFUSER | $1._{\pm9.9}$ | $.074_{\pm.26}$ | $(.53_{\pm7.3}, .6_{\pm7.7}, .34_{\pm5.9}, .23_{\pm4.8})$ | $(.96_{\pm.21}, .96_{\pm.2}, .95_{\pm.21}, .97_{\pm.16})$ |
| MMD-CBS$^*$ | $79_{\pm41}$ | $.99_{\pm.10}$ | $(0._{\pm0.}, 0._{\pm0.}, 0._{\pm0.}, 0._{\pm0.})$ | $(.99_{\pm.10}, .96_{\pm.20}, .99_{\pm.10}, .99_{\pm.10})$ |
| DIFFUSER + projection | $0._{\pm0.}$ | $.11_{\pm.32}$ | $(100_{\pm0}, 100_{\pm0}, 100_{\pm0}, 100_{\pm0})$ | $(.73_{\pm.44}, .72_{\pm.45}, .70_{\pm.46}, .75_{\pm.43})$ |
| CD-LB (w/o proj.) | $100_{\pm0}$ | $.99_{\pm.10}$ | $(0._{\pm0.}, 0._{\pm0.}, 0._{\pm0.}, .023_{\pm1.5})$ | $(.89_{\pm.31}, .91_{\pm.28}, .90_{\pm.30}, .89_{\pm.32})$ |
| CD-SHD (w/o proj.) | $100_{\pm0.}$ | $1._{\pm0.}$ | $(.078_{\pm2.8}, .047_{\pm2.2}, .039_{\pm2.}, .023_{\pm1.5})$ | $(.98_{\pm.14}, .98_{\pm.14}, .98_{\pm.13}, .98_{\pm.13})$ |
| PCD-LB | $95_{\pm22}$ | $.84_{\pm.36}$ | $(100_{\pm0}, 100_{\pm0}, 100_{\pm0}, 100_{\pm0})$ | $(.80_{\pm.40}, .81_{\pm.40}, .80_{\pm.40}, .79_{\pm.41})$ |
| PCD-SHD | $100_{\pm0}$ | $.93_{\pm.26}$ | $(100_{\pm0}, 100_{\pm0}, 100_{\pm0}, 100_{\pm0})$ | $(.91_{\pm.28}, .90_{\pm.30}, .90_{\pm.30}, .89_{\pm.31})$ |
| **Max Vel. = 1.46** | | | | |
| vanilla DIFFUSER | $1._{\pm9.9}$ | $.074_{\pm.26}$ | $(3.2_{\pm18}, 4.1_{\pm20}, 1.9_{\pm14}, 1.6_{\pm13})$ | $(.96_{\pm.21}, .96_{\pm.2}, .95_{\pm.21}, .97_{\pm.16})$ |
| MMD-CBS$^*$ | $79_{\pm41}$ | $.99_{\pm.10}$ | $(2._{\pm14}, 2._{\pm14}, 4._{\pm20}, 4._{\pm20})$ | $(.99_{\pm.10}, .96_{\pm.20}, .99_{\pm.10}, .99_{\pm.10})$ |
| DIFFUSER + projection | $1._{\pm9.9}$ | $.092_{\pm.29}$ | $(100_{\pm0}, 100_{\pm0}, 100_{\pm0}, 100_{\pm0})$ | $(.90_{\pm.30}, .89_{\pm.32}, .89_{\pm.31}, .91_{\pm.28})$ |
| CD-LB (w/o proj.) | $100_{\pm0}$ | $.99_{\pm.10}$ | $(.14_{\pm3.7}, .016_{\pm1.2}, .0078_{\pm.88}, .13_{\pm3.6})$ | $(.89_{\pm.31}, .91_{\pm.28}, .90_{\pm.30}, .89_{\pm.32})$ |
| CD-SHD (w/o proj.) | $100_{\pm0}$ | $1._{\pm0.}$ | $(.68_{\pm8.2}, .60_{\pm7.7}, .33_{\pm5.7}, .31_{\pm5.6})$ | $(.98_{\pm.14}, .98_{\pm.14}, .98_{\pm.13}, .98_{\pm.13})$ |
| PCD-LB | $100_{\pm0}$ | $.89_{\pm.32}$ | $(100_{\pm0}, 100_{\pm0}, 100_{\pm0}, 100_{\pm0})$ | $(.90_{\pm.29}, .90_{\pm.29}, .91_{\pm.28}, .90_{\pm.30})$ |
| PCD-SHD | $100_{\pm0}$ | $.97_{\pm.18}$ | $(100_{\pm0}, 100_{\pm0}, 100_{\pm0}, 100_{\pm0})$ | $(.97_{\pm.17}, .96_{\pm.19}, .96_{\pm.19}, .96_{\pm.19})$ |
| **Max Vel. = 1.76** | | | | |
| vanilla DIFFUSER | $1._{\pm9.9}$ | $.074_{\pm.26}$ | $(34_{\pm47}, 32_{\pm47}, 30_{\pm46}, 30_{\pm46})$ | $(.96_{\pm.21}, .96_{\pm.2}, .95_{\pm.21}, .97_{\pm.16})$ |
| MMD-CBS$^*$ | $79_{\pm41}$ | $.99_{\pm.10}$ | $(48_{\pm50}, 53_{\pm50}, 40_{\pm49}, 43_{\pm50})$ | $(.99_{\pm.10}, .96_{\pm.20}, .99_{\pm.10}, .99_{\pm.10})$ |
| DIFFUSER + projection | $1._{\pm9.9}$ | $.08_{\pm.27}$ | $(100_{\pm0}, 100_{\pm0}, 100_{\pm0}, 100_{\pm0})$ | $(.95_{\pm.22}, .94_{\pm.23}, .94_{\pm.23}, .97_{\pm.18})$ |
| CD-LB (w/o proj.) | $100_{\pm0}$ | $.99_{\pm.1}$ | $(.35_{\pm5.9}, .34_{\pm5.9}, .20_{\pm4.4}, .42_{\pm6.5})$ | $(.89_{\pm.31}, .91_{\pm.28}, .90_{\pm.3}, .89_{\pm.32})$ |
| CD-SHD (w/o proj.) | $100_{\pm0}$ | $1._{\pm0.}$ | $(13_{\pm33}, 14_{\pm35}, 12_{\pm32}, 10_{\pm30})$ | $(.98_{\pm.14}, .98_{\pm.14}, .98_{\pm.13}, .98_{\pm.13})$ |
| PCD-LB | $100_{\pm0}$ | $.92_{\pm.27}$ | $(100_{\pm0}, 100_{\pm0}, 100_{\pm0}, 100_{\pm0})$ | $(.95_{\pm.21}, .96_{\pm.21}, .96_{\pm.19}, .95_{\pm.22})$ |
| PCD-SHD | $100_{\pm0}$ | $.99_{\pm.097}$ | $(100_{\pm0}, 100_{\pm0}, 100_{\pm0}, 100_{\pm0})$ | $(.98_{\pm.13}, .98_{\pm.14}, .98_{\pm.16}, .98_{\pm.14})$ |

Table 12: Task Conveyor, 4 robots, 100 random tests, sample size 128 except MMD-CBS. $^*$MMD-CBS yielded *no* trajectories when it failed to find an inter-robot-collision-free solution within time limit; therefore, the CS metric for MMD-CBS is calculated only with inter-robot-collision-free trajectories.

| **Task `DropRegion`, 4 Robots** | | | | |
|---|---|---|---|---|
| METHOD \ Metric | SU(%)↑ | RS↑ | CS(%)↑ | DA↑ |
| **Max Vel. = 0.928** | | | | |
| vanilla DIFFUSER | $3.0_{\pm17}$ | $.053_{\pm.22}$ | $(11_{\pm32}, 9.7_{\pm30}, 13_{\pm33}, 11_{\pm32})$ | $(.90_{\pm.30}, .89_{\pm.32}, .89_{\pm.32}, .89_{\pm.31})$ |
| MMD-CBS | $100_{\pm0.}$ | $1._{\pm0.}$ | $(18_{\pm38}, 16_{\pm37}, 17_{\pm38}, 19_{\pm39})$ | $(.98_{\pm.14}, .98_{\pm.14}, 1._{\pm0.}, .99_{\pm.10})$ |
| DIFFUSER + projection | $4.0_{\pm20}$ | $.084_{\pm.28}$ | $(100_{\pm0}, 100_{\pm0}, 100_{\pm0}, 100_{\pm0})$ | $(.93_{\pm.26}, .92_{\pm.27}, .93_{\pm.26}, .92_{\pm.27})$ |
| CD-LB (w/o proj.) | $100_{\pm0.}$ | $.87_{\pm.34}$ | $(4.1_{\pm20}, 4.0_{\pm20}, 3.7_{\pm19}, 3.0_{\pm17})$ | $(.24_{\pm.43}, .24_{\pm.42}, .26_{\pm.44}, .24_{\pm.43})$ |
| CD-SHD (w/o proj.) | $100_{\pm0}$ | $.93_{\pm.25}$ | $(5.3_{\pm22}, 4.3_{\pm20}, 6.1_{\pm24}, 4.6_{\pm21})$ | $(.93_{\pm.26}, .92_{\pm.27}, .93_{\pm.26}, .93_{\pm.26})$ |
| PCD-LB | $98_{\pm14}$ | $.72_{\pm.45}$ | $(100_{\pm0}, 100_{\pm0}, 100_{\pm0}, 100_{\pm0})$ | $(.23_{\pm.42}, .21_{\pm.41}, .26_{\pm.44}, .22_{\pm.42})$ |
| PCD-SHD | $100_{\pm0}$ | $.98_{\pm.15}$ | $(100_{\pm0}, 100_{\pm0}, 100_{\pm0}, 100_{\pm0})$ | $(.92_{\pm.27}, .91_{\pm.29}, .92_{\pm.27}, .92_{\pm.27})$ |
| **Max Vel. = 1.13** | | | | |
| vanilla DIFFUSER | $3.0_{\pm17}$ | $.053_{\pm.22}$ | $(40_{\pm49}, 37_{\pm48}, 37_{\pm48}, 40_{\pm49})$ | $(.90_{\pm.30}, .89_{\pm.32}, .89_{\pm.32}, .89_{\pm.31})$ |
| MMD-CBS | $100_{\pm0.}$ | $1._{\pm0.}$ | $(43_{\pm30}, 39_{\pm49}, 51_{\pm50}, 50_{\pm50})$ | $(.98_{\pm.14}, .98_{\pm.14}, 1._{\pm0.}, .99_{\pm.10})$ |
| DIFFUSER + projection | $3.0_{\pm17}$ | $.074_{\pm.26}$ | $(100_{\pm0}, 100_{\pm0}, 100_{\pm0}, 100_{\pm0})$ | $(.93_{\pm.26}, .91_{\pm.28}, .92_{\pm.27}, .91_{\pm.28})$ |
| CD-LB (w/o proj.) | $100_{\pm0}$ | $.87_{\pm.34}$ | $(9.6_{\pm29}, 9.9_{\pm30}, 8.7_{\pm28}, 9.0_{\pm29})$ | $(.24_{\pm.43}, .24_{\pm.42}, .26_{\pm.44}, .24_{\pm.43})$ |
| CD-SHD (w/o proj.) | $100_{\pm0}$ | $.93_{\pm.25}$ | $(18_{\pm38}, 15_{\pm36}, 18_{\pm39}, 18_{\pm38})$ | $(.93_{\pm.26}, .92_{\pm.27}, .93_{\pm.26}, .93_{\pm.26})$ |
| PCD-LB | $99_{\pm9.9}$ | $.72_{\pm.45}$ | $(100_{\pm0}, 100_{\pm0}, 100_{\pm0}, 100_{\pm0})$ | $(.22_{\pm.42}, .2_{\pm.4}, .25_{\pm.43}, .22_{\pm.41})$ |
| PCD-SHD | $100_{\pm0}$ | $.98_{\pm.13}$ | $(100_{\pm0}, 100_{\pm0}, 100_{\pm0}, 100_{\pm0})$ | $(.92_{\pm.28}, .91_{\pm.29}, .92_{\pm.28}, .92_{\pm.28})$ |
| **Max Vel. = 1.34** | | | | |
| vanilla DIFFUSER | $3.0_{\pm17}$ | $.053_{\pm.22}$ | $(78_{\pm41}, 75_{\pm43}, 76_{\pm42}, 77_{\pm42})$ | $(.90_{\pm.30}, .89_{\pm.32}, .89_{\pm.32}, .89_{\pm.31})$ |
| MMD-CBS | $100_{\pm0.}$ | $1._{\pm0.}$ | $(65_{\pm48}, 69_{\pm46}, 75_{\pm43}, 75_{\pm43})$ | $(.98_{\pm.14}, .98_{\pm.14}, 1._{\pm0.}, .99_{\pm.10})$ |
| DIFFUSER + projection | $4.0_{\pm20}$ | $.068_{\pm.25}$ | $(100_{\pm0}, 100_{\pm0}, 100_{\pm0}, 100_{\pm0})$ | $(.92_{\pm.27}, .91_{\pm.29}, .91_{\pm.28}, .91_{\pm.29})$ |
| CD-LB (w/o proj.) | $100_{\pm0}$ | $.87_{\pm.34}$ | $(15_{\pm36}, 16_{\pm37}, 16_{\pm37}, 15_{\pm36})$ | $(.24_{\pm.43}, .24_{\pm.42}, .26_{\pm.44}, .24_{\pm.43})$ |
| CD-SHD (w/o proj.) | $100_{\pm0}$ | $.93_{\pm.25}$ | $(39_{\pm49}, 34_{\pm47}, 39_{\pm49}, 36_{\pm48})$ | $(.93_{\pm.26}, .92_{\pm.27}, .93_{\pm.26}, .93_{\pm.26})$ |
| PCD-LB | $99_{\pm9.9}$ | $.74_{\pm.44}$ | $(100_{\pm0}, 100_{\pm0}, 100_{\pm0}, 100_{\pm0})$ | $(.21_{\pm.41}, .2_{\pm.4}, .24_{\pm.43}, .21_{\pm.41})$ |
| PCD-SHD | $100_{\pm0}$ | $.99_{\pm.10}$ | $(100_{\pm0}, 100_{\pm0}, 100_{\pm0}, 100_{\pm0})$ | $(.91_{\pm.28}, .91_{\pm.29}, .91_{\pm.28}, .91_{\pm.28})$ |

Table 13: Task `DropRegion`, 4 robots, 100 random tests, sample size 128 except MMD-CBS.

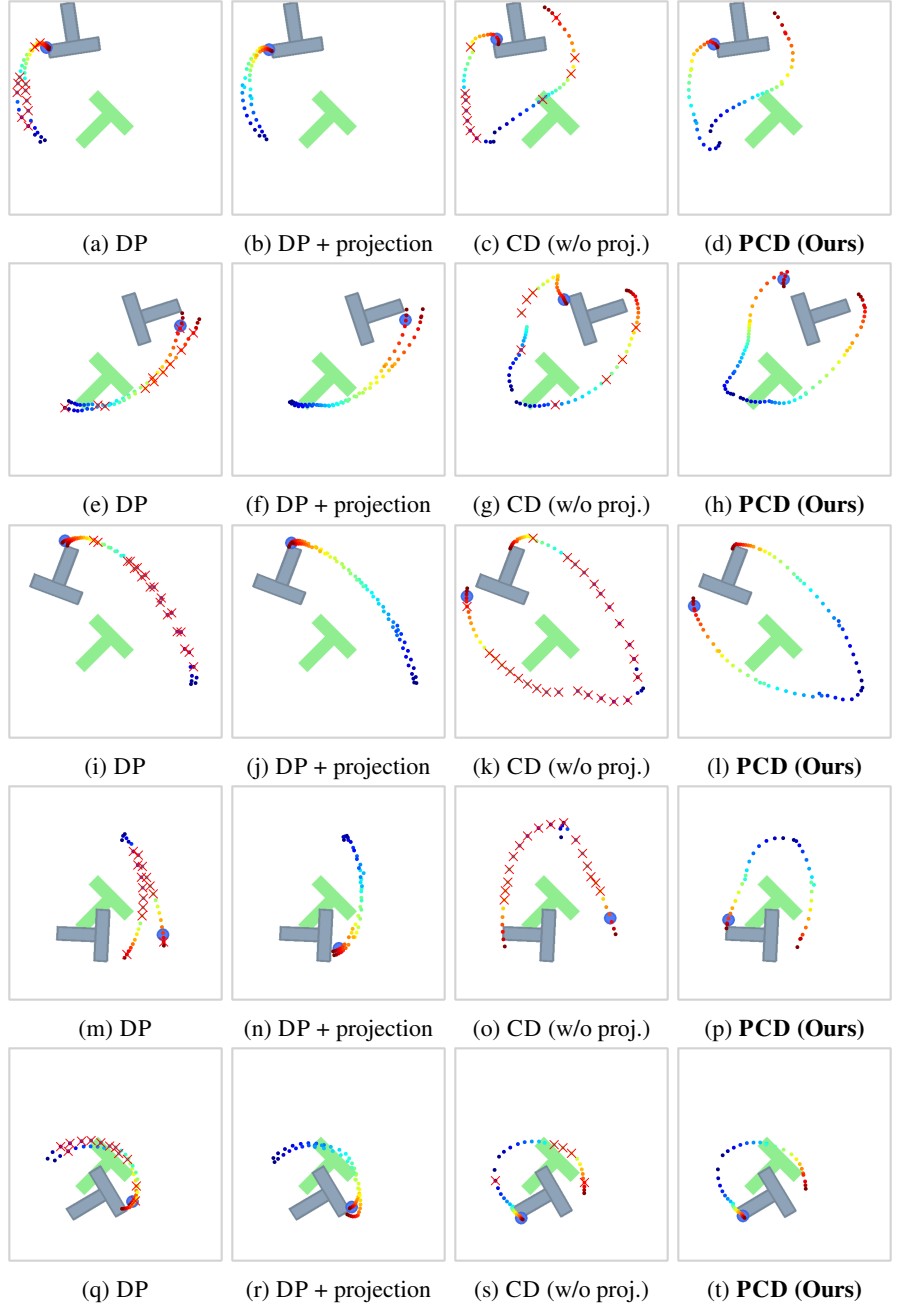

Figure 20: Additional qualitative results of the `PushT` experiment. Rows correspond to initial observations; columns correspond to methods. Robot trajectories use a colormap (warmer colors indicate later time steps); red crosses mark velocity violations. Only the first few dozen steps are shown for clarity.

**`PushT` Experiment Results**

| Method\Metric | DTW↑ | DFD↑ | CS(%)↑ | TC↑ |
|---|---|---|---|---|
| **Max Vel. = 6.2** | | | | |
| DP | $3.16_{\pm1.13}$ | $0.465_{\pm0.149}$ | $(47.6_{\pm17.1}, 48_{\pm17.0})$ | $(0.927_{\pm0.195}, 0.931_{\pm0.188})$ |
| DP + proj. | $2.95_{\pm1.26}$ | $0.422_{\pm0.166}$ | $(100_{\pm0}, 100_{\pm0})$ | $(0.862_{\pm0.253}, 0.860_{\pm0.255})$ |
| CD-DPP | $3.74_{\pm1.04}$ | $0.544_{\pm0.135}$ | $(44.4_{\pm16.6}, 44.5_{\pm16.3})$ | $(0.923_{\pm0.206}, 0.922_{\pm0.204})$ |
| CD-DPP-PS | $4.48_{\pm0.911}$ | $0.648_{\pm0.114}$ | $(39.6_{\pm15.5}, 39.7_{\pm15.5})$ | $(0.912_{\pm0.208}, 0.917_{\pm0.199})$ |
| CD-LB | $4.13_{\pm1.17}$ | $0.596_{\pm0.151}$ | $(41.6_{\pm15.4}, 41.3_{\pm15.5})$ | $(0.910_{\pm0.208}, 0.912_{\pm0.201})$ |
| CD-LB-PS | $4.50_{\pm1.01}$ | $0.646_{\pm0.129}$ | $(41.4_{\pm16.4}, 41.4_{\pm16.3})$ | $(0.921_{\pm0.199}, 0.925_{\pm0.189})$ |
| PCD-DPP | $4.63_{\pm1.07}$ | $0.643_{\pm0.135}$ | $(100_{\pm0}, 100_{\pm0})$ | $(0.776_{\pm0.301}, 0.777_{\pm0.298})$ |
| PCD-DPP-PS | $4.34_{\pm1.06}$ | $0.610_{\pm0.134}$ | $(100_{\pm0}, 100_{\pm0})$ | $(0.856_{\pm0.258}, 0.855_{\pm0.253})$ |
| PCD-LB | $5.07_{\pm1.05}$ | $0.696_{\pm0.132}$ | $(100_{\pm0}, 100_{\pm0})$ | $(0.747_{\pm0.3}, 0.750_{\pm0.3})$ |
| PCD-LB-PS | $4.32_{\pm1.09}$ | $0.605_{\pm0.138}$ | $(100_{\pm0}, 100_{\pm0})$ | $(0.865_{\pm0.257}, 0.861_{\pm0.259})$ |
| **Max Vel. = 8.4** | | | | |
| DP | $3.16_{\pm1.13}$ | $0.465_{\pm0.149}$ | $(64.8_{\pm13.8}, 65_{\pm13.7})$ | $(0.927_{\pm0.195}, 0.931_{\pm0.188})$ |
| DP + proj. | $2.96_{\pm1.21}$ | $0.428_{\pm0.159}$ | $(100_{\pm0}, 100_{\pm0})$ | $(0.896_{\pm0.227}, 0.888_{\pm0.237})$ |
| CD-DPP | $3.74_{\pm1.04}$ | $0.544_{\pm0.135}$ | $(62.5_{\pm13.9}, 62.4_{\pm13.6})$ | $(0.923_{\pm0.206}, 0.922_{\pm0.204})$ |
| CD-DPP-PS | $4.48_{\pm0.911}$ | $0.648_{\pm0.114}$ | $(57.4_{\pm14.0}, 57.3_{\pm13.9})$ | $(0.912_{\pm0.208}, 0.917_{\pm0.199})$ |
| CD-LB | $4.13_{\pm1.17}$ | $0.596_{\pm0.151}$ | $(58.9_{\pm13.8}, 58.6_{\pm14.1})$ | $(0.910_{\pm0.208}, 0.912_{\pm0.201})$ |
| CD-LB-PS | $4.50_{\pm1.01}$ | $0.646_{\pm0.129}$ | $(58.8_{\pm14.4}, 58.7_{\pm14.3})$ | $(0.921_{\pm0.199}, 0.925_{\pm0.189})$ |
| PCD-DPP | $4.55_{\pm1.00}$ | $0.638_{\pm0.126}$ | $(100_{\pm0}, 100_{\pm0})$ | $(0.829_{\pm0.272}, 0.834_{\pm0.271})$ |
| PCD-DPP-PS | $4.39_{\pm1.05}$ | $0.622_{\pm0.133}$ | $(100_{\pm0}, 100_{\pm0})$ | $(0.885_{\pm0.236}, 0.885_{\pm0.233})$ |
| PCD-LB | $5.12_{\pm1.08}$ | $0.708_{\pm0.135}$ | $(100_{\pm0}, 100_{\pm0})$ | $(0.778_{\pm0.288}, 0.791_{\pm0.275})$ |
| PCD-LB-PS | $4.38_{\pm1.02}$ | $0.618_{\pm0.129}$ | $(100_{\pm0}, 100_{\pm0})$ | $(0.890_{\pm0.234}, 0.882_{\pm0.240})$ |
| **Max Vel. = 10.7** | | | | |
| DP | $3.16_{\pm1.13}$ | $0.465_{\pm0.149}$ | $(77.6_{\pm9.97}, 77.6_{\pm9.81})$ | $(0.927_{\pm0.195}, 0.931_{\pm0.188})$ |
| DP + proj. | $3.00_{\pm1.18}$ | $0.435_{\pm0.155}$ | $(100_{\pm0}, 100_{\pm0})$ | $(0.905_{\pm0.222}, 0.906_{\pm0.216})$ |
| CD-DPP | $3.74_{\pm1.04}$ | $0.544_{\pm0.135}$ | $(76.2_{\pm10.3}, 76.2_{\pm10.1})$ | $(0.923_{\pm0.206}, 0.922_{\pm0.204})$ |
| CD-DPP-PS | $4.48_{\pm0.911}$ | $0.648_{\pm0.114}$ | $(71.8_{\pm10.8}, 71.7_{\pm10.8})$ | $(0.912_{\pm0.208}, 0.917_{\pm0.199})$ |
| CD-LB | $4.13_{\pm1.17}$ | $0.596_{\pm0.151}$ | $(72.7_{\pm10.9}, 72.3_{\pm11.0})$ | $(0.910_{\pm0.208}, 0.912_{\pm0.201})$ |
| CD-LB-PS | $4.50_{\pm1.01}$ | $0.646_{\pm0.129}$ | $(72.6_{\pm11.1}, 72.6_{\pm11.1})$ | $(0.921_{\pm0.199}, 0.925_{\pm0.189})$ |
| PCD-DPP | $4.52_{\pm0.996}$ | $0.637_{\pm0.125}$ | $(100_{\pm0}, 100_{\pm0})$ | $(0.852_{\pm0.26}, 0.855_{\pm0.257})$ |
| PCD-DPP-PS | $4.40_{\pm1.02}$ | $0.626_{\pm0.127}$ | $(100_{\pm0}, 100_{\pm0})$ | $(0.892_{\pm0.228}, 0.900_{\pm0.220})$ |
| PCD-LB | $5.16_{\pm1.09}$ | $0.716_{\pm0.136}$ | $(100_{\pm0}, 100_{\pm0})$ | $(0.804_{\pm0.271}, 0.803_{\pm0.273})$ |
| PCD-LB-PS | $4.39_{\pm1.01}$ | $0.622_{\pm0.129}$ | $(100_{\pm0}, 100_{\pm0})$ | $(0.896_{\pm0.227}, 0.900_{\pm0.222})$ |

Table 14: Results of `PushT` task by all compared methods with all three velocity limits. DP refers to DIFFUSION POLICY; CD denotes DP+coupling only; PS denotes posterior sampling variants of cost functions.

| Method | XOR%↑ | M/F%↑ | SE-CLIP↑ | SE-LPIPS↓ | SE-FID↓ | IS-LPIPS↑ |
|---|---|---|---|---|---|---|
| SD | 20 | 71/14 | $.40_{\pm0.046}/.41_{\pm0.035}$ | $.77_{\pm0.043}/.76_{\pm0.026}$ | $.42/.43$ | $\mathbf{.75}_{\pm0.059}/\mathbf{.75}_{\pm0.036}$ |
| CNet[†] | 8 | $\mathbf{100}/56$ | $.68_{\pm0.036}/.75_{\pm0.044}$ | $.48_{\pm0.039}/.50_{\pm0.056}$ | $.29/.44$ | $.45_{\pm0.043}/.50_{\pm0.046}$ |
| SD+P | 44 | $\mathbf{100}/\mathbf{100}$ | $\mathbf{.88}_{\pm0.001}/.91_{\pm0.003}$ | $.15_{\pm0.022}/.16_{\pm0.038}$ | $.11/.17$ | $.06_{\pm0.043}/.09_{\pm0.064}$ |
| Sync[†] | 48 | $95/17$ | $.62_{\pm0.052}/.61_{\pm0.066}$ | $.67_{\pm0.051}/.68_{\pm0.049}$ | $.39/.81$ | $.70_{\pm0.051}/.69_{\pm0.060}$ |
| SD+C | 48 | $51/47$ | $.45_{\pm0.057}/.46_{\pm0.067}$ | $.76_{\pm0.042}/.74_{\pm0.033}$ | $.47/.54$ | $\mathbf{.75}_{\pm0.052}/\mathbf{.75}_{\pm0.051}$ |
| SD+C[†] | 64 | $47/37$ | $.55_{\pm0.064}/.60_{\pm0.097}$ | $.70_{\pm0.054}/.68_{\pm0.044}$ | $.38/.42$ | $.69_{\pm0.066}/.69_{\pm0.064}$ |
| PCD | $\mathbf{96}$ | $\mathbf{100}/\mathbf{100}$ | $\mathbf{.88}_{\pm0.003}/\mathbf{.92}_{\pm0.004}$ | $\mathbf{.11}_{\pm0.019}/\mathbf{.14}_{\pm0.032}$ | $\mathbf{.05}/\mathbf{.09}$ | $.11_{\pm0.088}/.13_{\pm0.089}$ |

Table 15: Paired face-generation results with projection and coupling applied. $M=2$ exemplars used are as per Figure 4a. Boldface indicates the best score(s) for each metric.

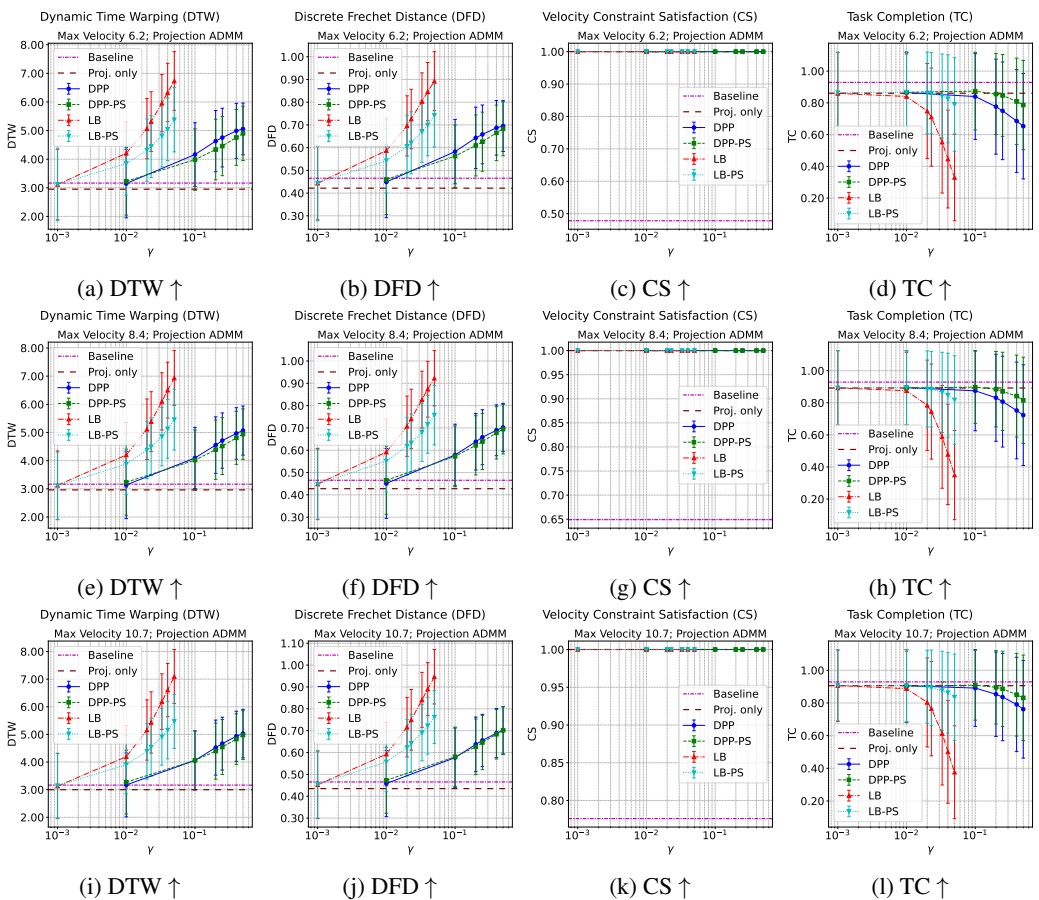

Figure 21: Coupling Strength Ablation of **PCD**: Evaluation metrics results at different $v_{max}$s as coupling stength $\gamma$ and coupling function vary. (a,b,c,d) $v_{max} = 6.2$; (e,f,g,h) $v_{max} = 8.4$; (i,j,k,l) $v_{max} = 10.7$.

| METHOD | XOR%↑ | M/F%↑ | SE-CLIP↑ | SE-LPIPS↓ | SE-FID↓ | IS-LPIPS↑ |
|---|---|---|---|---|---|---|
| SD | 20 | 71/14 | $.40_{\pm 0.046}/.41_{\pm 0.035}$ | $.77_{\pm 0.043}/.76_{\pm 0.026}$ | .39/.40 | $\mathbf{.75}_{\pm 0.060}/\mathbf{.75}_{\pm 0.036}$ |
| CNet[†] | 12 | **100**/29 | $.64_{\pm 0.051}/.73_{\pm 0.055}$ | $.47_{\pm 0.054}/.51_{\pm 0.050}$ | .24/.30 | $.44_{\pm 0.043}/.49_{\pm 0.045}$ |
| SD+P | 44 | **100**/**100** | $.84_{\pm 0.014}/\mathbf{.91}_{\pm 0.006}$ | $.22_{\pm 0.036}/.27_{\pm 0.023}$ | .11/.20 | $.14_{\pm 0.051}/.11_{\pm 0.039}$ |
| Sync[†] | 48 | 98/17 | $.64_{\pm 0.049}/0.59_{\pm 0.067}$ | $.67_{\pm 0.045}/.67_{\pm 0.052}$ | .41/.74 | $.71_{\pm 0.062}/.69_{\pm 0.060}$ |
| SD+C | 48 | 51/46 | $.44_{\pm 0.055}/.44_{\pm 0.065}$ | $.76_{\pm 0.042}/.73_{\pm 0.029}$ | .43/.49 | $\mathbf{.75}_{\pm 0.052}/\mathbf{.75}_{\pm 0.051}$ |
| SD+C[†] | 64 | 47/38 | $.54_{\pm 0.064}/.59_{\pm 0.104}$ | $.70_{\pm 0.0535}/.68_{\pm 0.044}$ | .35/.37 | $.69_{\pm 0.066}/.69_{\pm 0.064}$ |
| PCD | **76** | **100**/**100** | $\mathbf{.85}_{\pm 0.013}/.90_{\pm 0.008}$ | $\mathbf{.19}_{\pm 0.064}/\mathbf{.24}_{\pm 0.041}$ | **.09**/**.14** | $.18_{\pm 0.082}/.16_{\pm 0.065}$ |

Table 16: Paired face-generation results with projection and coupling applied. $M=6$ exemplars used are as per Figure 25. Boldface indicates the best score(s) for each metric.

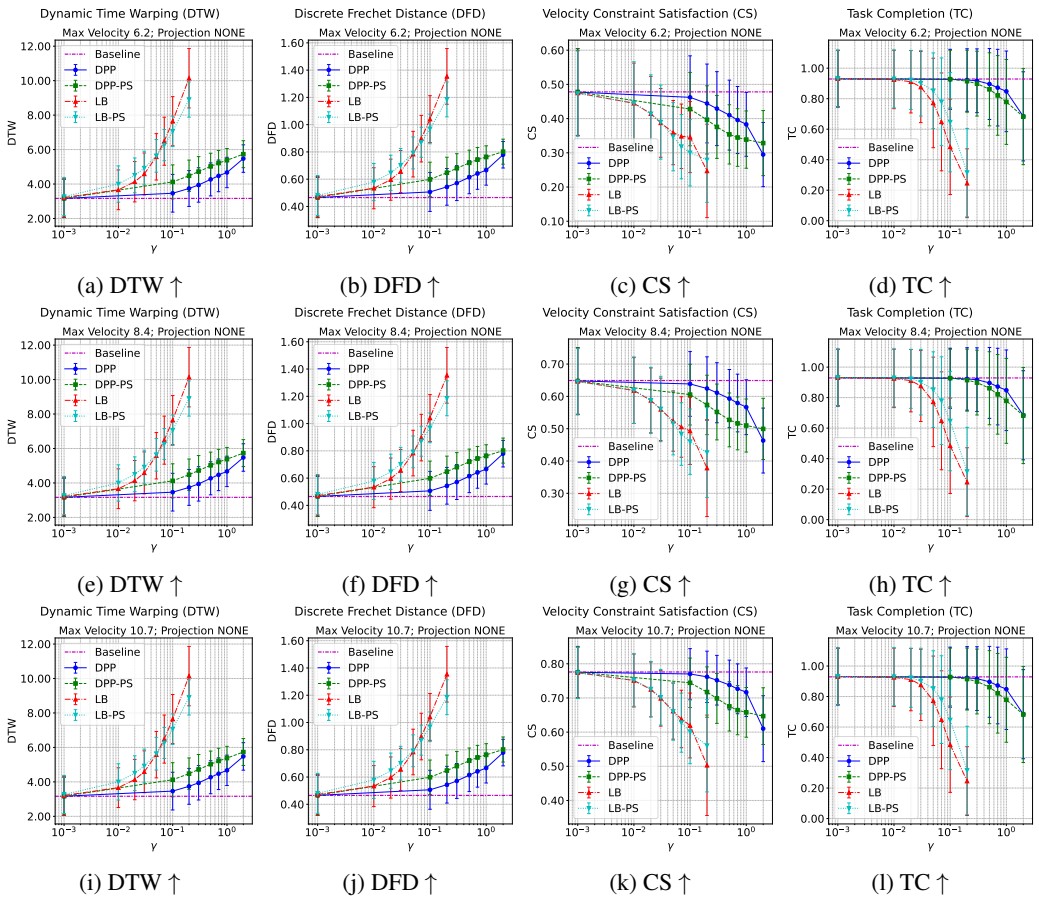

Figure 22: Coupling Strength Ablation of **CD** (coupling-only, without projection): Evaluation metrics results at different $v_{\max}$s as coupling stength $\gamma$ and coupling function vary. (a,b,c,d) $v_{\max} = 6.2$; (e,f,g,h) $v_{\max} = 8.4$; (i,j,k,l) $v_{\max} = 10.7$.

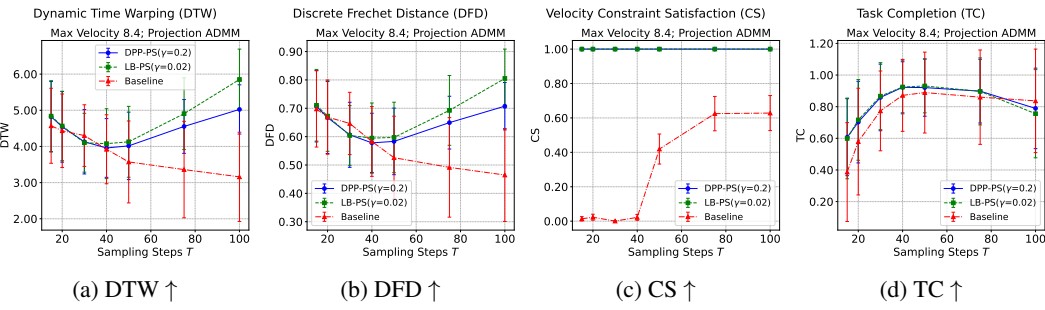

Figure 23: Sampling steps ablation of PCD: Evaluation metrics results with different number of sampling steps $T$ given the same coupling strength $\gamma$ for each type of cost.

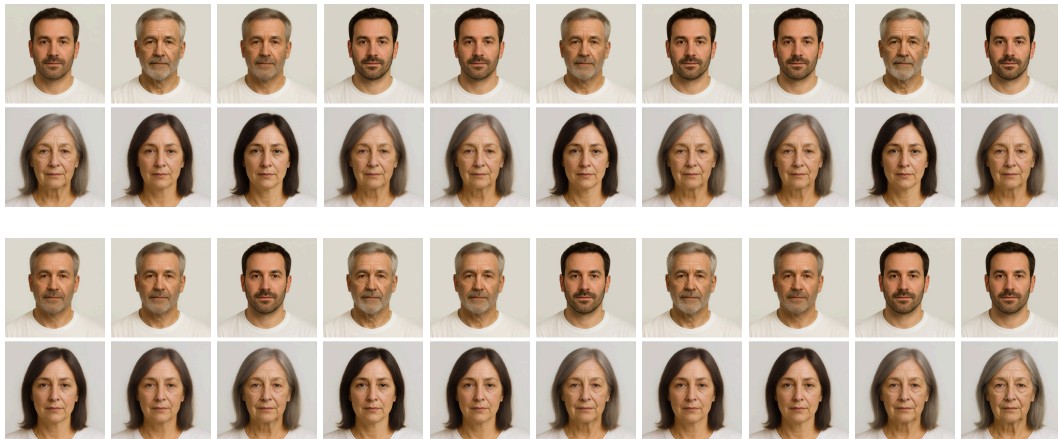

Figure 24: Sample pairs generated with both projection and coupling applied. The two exemplars used for each model (each row) are as per Figure 4a.

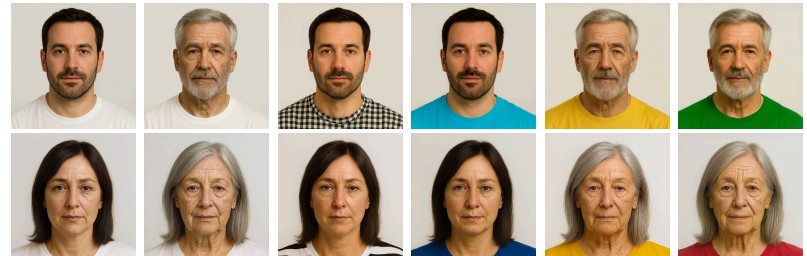

Figure 25: *Six* exemplar images for each model (one row per model), generated via ChatGPT OpenAI (2023). The exemplars differ only by minor visual details, ensuring close structural alignment.

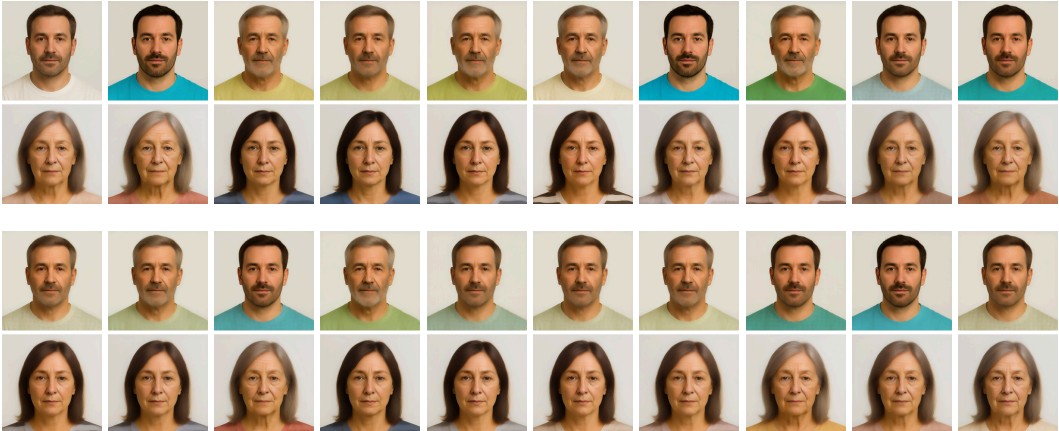

Figure 26: Sample pairs generated with both projection and coupling applied. The *six* exemplars used for each model (each row) are as per Figure 25.

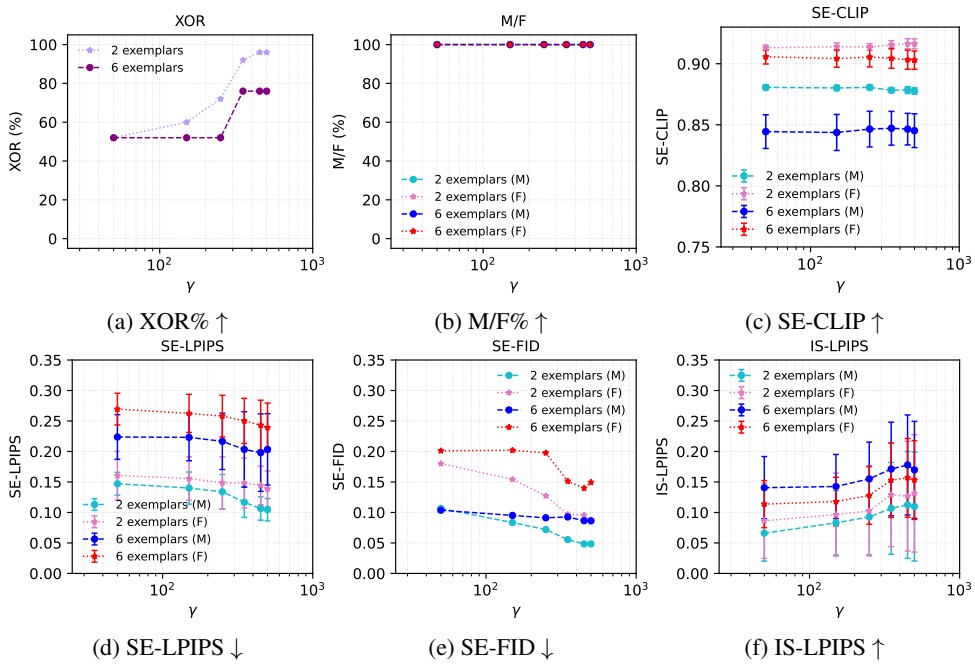

Figure 27: Ablation of coupling strength $\gamma$ for **PCD**, using $M \in \{2, 6\}$ exemplars per model. We evaluate $\gamma \in \{50, 150, 250, 350, 450, 500\}$.

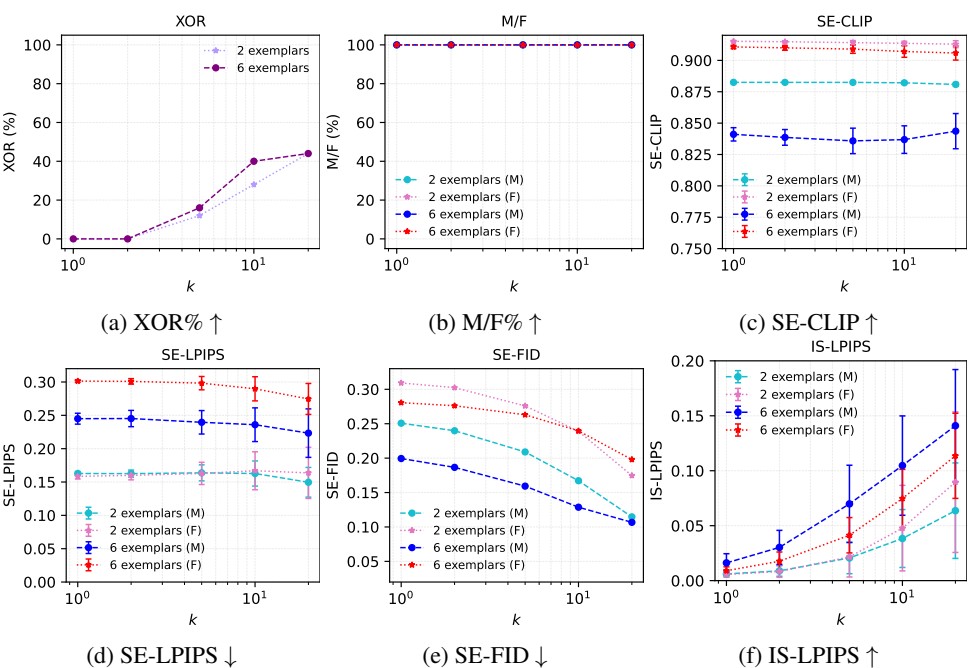

Figure 28: Ablation of noise scaling factor $k$ for **SD+P**, using $M \in \{2, 6\}$ exemplars per model. We evaluate $k \in \{1, 2, 5, 10, 20\}$.

