# OpenReview forum: "Projected Coupled Diffusion for Test-Time Constrained Joint Generation"
_ICLR.cc/2026/Conference — ICLR 2026 Poster_

### Official Review · Reviewer_DJcn · 2025-10-26

**Soundness:** 3
**Presentation:** 4
**Contribution:** 3
**Rating:** 6
**Confidence:** 4

**Summary:**

The paper proposes a method to jointly sample two independent variables coupled using a joint cost function. Practically, given the independent denoising score functions of each variable, the goal is to couple their denoising process directly at test time to sample them jointly. The paper introduces some variants: (1) standard LMC/DDPM based: perform one step-denoising, use classifier guidance using the gradient of the cost function and then perform projection to map the noisy latents to the safe-set. (2) use tweedie clean estimates to calculate the gradient of the cost function. Overall the paper claims that the proposed method generalizes the formulation of classifier-guidance, projected diffusion, compositional diffusion and joint diffusion.

I am familiar with a concurrent work in this line of research: https://arxiv.org/abs/2509.08775
While my concerns are based on the concurrent work, my judgment of this paper will be independent of it.

**Strengths:**

1. The paper's unified formulation of classifier-guidance, projected diffusion, compositional diffusion and joint diffusion is very impactful and timely. The authors have provided exhaustive results to analyze the performance of their algorithm on toy-domain, a planar robotics task and an image generation task as well.

2. PCD can impose hard constraints in addition to joint diffusion while being computationally efficient.

**Weaknesses:**

1. Data fidelity vs projection: Since the cost function and projection operation hold for the clean distribution, data fidelity is of primary importance here. The more realistic the clean estimates are, the better evaluation and guidance can be done. However, as the authors acknowledge as well, the projection operation hurts data fidelity, as also observed by the concurrent work, when applied to low-quality clean estimates (particularly at higher noise levels).

2. Differentiability of cost function: This is a considerably strong assumption (which is also used by many prior works like MPD, DPCC). For example, the signed distance based indicator function in SHD might not be differentiable everywhere. This is a case, especially for collision checking objectives for real robot executions. I agree that effective engineering design can mitigate this, but this limits the scalability of the approach.

3. convexity of constraints: Since two experiments in the paper deal with navigation and manipulation, it is worth noting that non-convex safe sets are pretty common in these two settings, most commonly arising from obstacle avoidance constraints. For example, in the highway task if the trajectories are trained without the rectangle in between and forced to avoid it at test time, the resulting safe-set becomes non-convex. This again limits the scalability.

**Questions:**

1. How feasible is designing a projection operator for every task? How sensitive is the overall quality of samples to projection hyperparameters?

2. How is the cost function in general defined for noisy latents? It seems from the algorithms that the method always uses the clean estimates for projection and using the same for cost function also empirically results in the best performance.

---

> ### Author Response · Authors · 2025-11-20
> **Response to Questions 1 & 2**
>
> We thank the reviewer for taking time to review our work, recognizing the strengths of our method, and providing constructive feedback and suggestions. In the following, we provide clarifications to your questions and concerns. Please note that all references to sections, line numbers, etc. below refer to the *initial version* of the manuscript unless stated otherwise.
>
>
> ## Q1 Practicality of projection design and sensitivity
>
> > How feasible is designing a projection operator for every task? How sensitive is the overall quality of samples to projection hyperparameters?
>
> - Feasibility of projector design: Indeed, the projection operator is domain- and task-specific. For instance, in PushT and the multi-robot experiments, the convex safe set describes velocity constraints and projection is efficiently done via alternating direction method of multipliers (ADMM). For face pair generation, the constrained set is the defined by the convex hull of a set of exemplars in the *latent space* and the projection algorithm is based on online mirror descent (MD). For the toy example, the projection is as simple as clipping. Empirically speaking, if the constraints posed by the tasks are convex, various tools from convex optimization can provide *efficient* means for designing and implementing the projection operator, most with theoretical guarantees in convergence. These are standard and accessible methods taught in graduate-level optimization classes, so designing projection operators can be quite feasible under such circumstances. When the constraints are non-convex, one might just formulate the projection as a general optimization problem and leverage off-the-shelf solvers, but neither optimality, convergence, nor efficiency can be guaranteed in general. Note that many non-convex problems are not even computationally tractable.
> - Projection hyperparameters sensitivity: In our experiments, both operators (ADMM and MD) are designed for solving convex optimizations. From our empirical experience, both methods are decently robust given reasonable hyperparameters at roughly the same order of magnitude (including a reasonable stopping tolerance *for the specific use case*); the adopted hyperparameters are also shared in Appendix B. We refer readers to [1,2] for details in further analyses of ADMM and MD if they are interested in those.
>
>
> ## Q2 Costs for noisy latents
>
> > How is the cost function in general defined for noisy latents? It seems from the algorithms that the method always uses the clean estimates for projection and using the same for cost function also empirically results in the best performance.
>
>
> - In Section 4, the coupled dynamics in Eq (3) use a generic cost that can technically take in noisy latents at each diffusion step. As we state there, the cost can be any differentiable function (or function admitting subgradients) or neural network defined on these variables (e.g., a classifier or regressor).
>     - In the case of using neural networks, we demonstrate the use of a latent classifier as part of the cost function in the face image pair experiment, i.e., the cost therein (line 426-430) is
>     $c(x,y) := \sum_{a} [ p^\psi(a|x)(1 − p^\psi(a|y)) + p^\psi(a|y)(1 − p^\psi(a|x))],$
>     where $p^\psi(a|\cdot)$ is the softmaxed logits from a trained latent classifier parameterized by $\psi$ predicting the age group ($a$) given the noisy latents of an image.
>     - In the case of analytical functions, noisy variables (assuming same dimension as the output sample) can be directly used as the input of a cost function designed for clean samples. We empirically find this simple setup working decently when compared to using Tweedie's estimates. In the PushT experiment (Section 5.2 and Appendix C.3), we explicitly compare both instantiations: methods without the "–PS" suffix (CD-DPP, CD-LB, PCD-DPP, PCD-LB) directly use noisy samples, while the PS variants (CD-DPP-PS, CD-LB-PS, PCD-DPP-PS, PCD-LB-PS) use Tweedie's estimates. Table 2 and Table 13 show that both families are effective: non-PS variants tend to achieve slightly higher coupling metrics (DTW/DFD), whereas PS variants often yield better task completion scores (TC), indicating better data fidelity.
>
>
>
> **References**
> [1] Beck, A., & Teboulle, M. (2003). Mirror descent and nonlinear projected subgradient methods for convex optimization. *Operations Research Letters*, 31(3), 167-175.
> [2] Boyd, S., Parikh, N., Chu, E., Peleato, B., & Eckstein, J. (2011). Distributed optimization and statistical learning via the alternating direction method of multipliers. *Foundations and Trends® in Machine learning*, 3(1), 1-122.

---

> ### Author Response · Authors · 2025-11-20
> **Response to Weaknesses 1 & 2**
>
> ## W1: Data fidelity vs projection
>
> > Data fidelity vs projection: Since the cost function and projection operation hold for the clean distribution, data fidelity is of primary importance here. The more realistic the clean estimates are, the better evaluation and guidance can be done. However, as the authors acknowledge as well, the projection operation hurts data fidelity, as also observed by the concurrent work, when applied to low-quality clean estimates (particularly at higher noise levels).
>
> - Thank you for your comment and sharing the similar findings in the concurrent work. As we checked, the concurrent work has made contributions to address this issue, and is valuable for our reference. We have added discussions on this point into the limitation section and mentioned this work and other related references in our *updated* manuscript (page 10).
>
>
> ## W2: Differentiability of cost function
>
> > Differentiability of cost function: This is a considerably strong assumption (which is also used by many prior works like MPD, DPCC). For example, the signed distance based indicator function in SHD might not be differentiable everywhere. This is a case, especially for collision checking objectives for real robot executions. I agree that effective engineering design can mitigate this, but this limits the scalability of the approach.
>
> - Thank you for pointing this out. Indeed, as the reviewer mentions, the cost differentiability assumption is not restrictive to our method and many existing approaches (e.g., MPD[1], DPCC[2]) make similar assumptions. For the signed distance example the reviewer mentioned, note that in our method using *subgradients* is fully acceptable and naturally accommodates ReLU-like costs or our SHD cost; this covers most reasonable *continuous* cost functions. We will clarify this point in our updated manuscript.
> - If the specific application permits, differentiable approximations for non-differentiable costs are available, e.g., for the rule-based costs specified with temporal logic [3,4], and their applications in diffusion-based sampling were also proposed recently [5,6].
> - Further, if gradient (or subgradient) information is indeed unavailable, alternatives such as Monte Carlo based importance sampling could be of use, and the concurrent work JM2D [7] the reviewer mentions has made concrete contributions in this direction and is good for reference. We will incorporate this point in the limitation discussion section of the revised manuscript and identify these recent efforts.
>
>
>
> **References**
> [1] Carvalho, J., Le, A. T., Baierl, M., Koert, D., & Peters, J. (2023). Motion planning diffusion: Learning and planning of robot motions with diffusion models. *2023 IEEE/RSJ International Conference on Intelligent Robots and Systems (IROS)* (pp. 1916-1923).
> [2] Römer, R., Rohr, A.v.; Schoellig, A.. (2025). Diffusion Predictive Control with Constraints. *Proceedings of the 7th Annual Learning for Dynamics \& Control Conference*, in *Proceedings of Machine Learning Research* 283:791-803.
> [3] Leung, K., Aréchiga, N., & Pavone, M. (2023). Backpropagation through signal temporal logic specifications: Infusing logical structure into gradient-based methods. *The International Journal of Robotics Research*, 42(6), 356-370.
> [4] Xu, Z., Rawat, Y., Wong, Y., Kankanhalli, M. S., & Shah, M. (2022). Don't pour cereal into coffee: Differentiable temporal logic for temporal action segmentation. *Advances in Neural Information Processing Systems*, 35, 14890-14903.
> [5] Feng, Z., Luan, H., Goyal, P., & Soh, H. (2024). LTLDoG: Satisfying temporally-extended symbolic constraints for safe diffusion-based planning. *IEEE Robotics and Automation Letters*.
> [6] Meng, Y., & Fan, C. (2024). Diverse controllable diffusion policy with signal temporal logic. *IEEE Robotics and Automation Letters*.
> [7] Jung, W., Mishra, U.A., Arachchige, N.R., Chen, Y., Xu, D., Kousik, S.. (2025). Joint Model-based Model-free Diffusion for Planning with Constraints. *Proceedings of The 9th Conference on Robot Learning*, 305:4328-4350.

---

> ### Author Response · Authors · 2025-11-20
> **Response to Weakness 3**
>
> ## W3: Convexity of constraints
>
> > convexity of constraints: Since two experiments in the paper deal with navigation and manipulation, it is worth noting that non-convex safe sets are pretty common in these two settings, most commonly arising from obstacle avoidance constraints. For example, in the highway task if the trajectories are trained without the rectangle in between and forced to avoid it at test time, the resulting safe-set becomes non-convex. This again limits the scalability.
>
> - Thank you for pointing out this important aspect. Indeed, tackling non-convex constraints is an open research topic in optimization as well as in areas like robotics. In the submission, we defer the accommodation of projection onto non-convex sets as for future work. However, we would like to briefly bring up some potential approaches:
>     - (i) It is possible to project onto certain types of non-convex sets by using convex projections. For example, for projecting onto a *union of a finite number of convex sets*, one can project onto each convex set and then choose the closest point among those projections. In the case of task Highways, we could *roughly* divide the feasible regions (the safe set) into two vertical and two horizontal rectangles (rows/columns). In practice, many reasonable non-convex sets can be regarded as unions of convex sets (e.g., in [1], obstacles are over-approximated with a union of convex polytopes; the same route can be applied to non-convex safe sets). The problem with this approach is that the number of convex sets might be large, and that will introduce a tradeoff between computational efficiency and accurate modeling.
>     - (ii) It is also sometimes possible to directly perform projections onto non-convex sets. For example, another concurrent work [2] deals with a restricted type of non-convex constraints and proposes an Augmented Lagrangian solver. However, the method does not give guarantees on the original non-convex problem and induces noticeable computational overheads.
>
>
> **References**
> [1] Thirugnanam, A., Zeng, J., & Sreenath, K. (2022). Safety-critical control and planning for obstacle avoidance between polytopes with control barrier functions. In *2022 International Conference on Robotics and Automation (ICRA)* (pp. 286-292).
> [2] Liang, J., Christopher, J. K., Koenig, S., & Fioretto, F.. (2025). Simultaneous Multi-Robot Motion Planning with Projected Diffusion Models. In *42nd International Conference on Machine Learning*.

---

> > ### Comment · Reviewer_DJcn · 2025-11-25
> > **Response to rebuttal**
> >
> > Thanks for all the clarifications and for acknowledging the weaknesses. This is an interesting line of research. I will maintain my score.

---

### Official Review · Reviewer_XYUt · 2025-10-31

**Soundness:** 3
**Presentation:** 3
**Contribution:** 3
**Rating:** 6
**Confidence:** 3

**Summary:**

This paper introduces Projected Coupled Diffusion (PCD), a novel test-time framework for generating jointly correlated samples from multiple pre-trained diffusion models while simultaneously enforcing hard, task-specific constraints. The core problem is that generating from a joint distribution $p(x, y)$ is difficult, especially when $x$ and $y$ must be correlated in a specific way and satisfy some given hard constraints. PCD addresses this by modifying the reverse diffusion sampling process. At each step, the update for each variable is guided by three components:

1. The score from its own pre-trained model (e.g., $s_X^\theta(x_t, t)$).

2. A gradient from a coupling cost function $c(x, y)$ that encourages the desired correlation between variables.

3. A projection operator $\Pi_{\mathcal{K}}$ that forces the updated sample back into the feasible set of hard constraints.

This approach requires no retraining and unifies compositional generation with hard-constraint enforcement at test time. The authors demonstrate PCD's effectiveness across three distinct domains: multi-robot motion planning, diverse robot manipulation, and constrained image-pair generation.

**Strengths:**

1. This paper unifies two important aspects in diffusion sampling: coupled generation and constrained generation through a clear and effective way. The proposed PCD provides a general, test-time-only framework to address this. PCD operates over multiple pre-trained models and costs (analytic or learned), requiring no retraining of the base diffusions, which enables the method applicable to a wide variety of settings, for example, where paired data and costs are scarce or proprietary.

2. This paper is well written, easy-to-follow, and conducts extensive studies across various domains, including multi-robot planning, Push-T trajectory pairs, and paired face generation.

**Weaknesses:**

1. PCD relies on projection and gradient-based updates to enforce constraints. How does it handle test-time constraints that are non-differentiable, for instance, a logic-based rule where a sample is accepted only if it passes some non-differentiable verification? A concrete example would help.

2. The performance of the proposed method might require the estimated Tweedie to be of high-quality. Otherwise, the further guidance term might likely be inaccurate or even compromise the overall sampling process.

3. This paper introduces gradient-based guidance and per-step projections, which can increase wall-clock latency compared with non-gradient based baselines. In appendix C.1, the authors also mention "PCD is approximately 4 ∼ 7× slower than vanilla diffusion mainly due to the per-step projection operation." Are there any potential ways to enable faster sampling with PCD?

4. Could the authors provide a curve of performance v.s. the number of sampling steps to illustrate how will the quality of the estimated Tweedie term affect performance? An intuition is that with more sampling steps, the Tweedie estimate will be more accurate, which could facilitate better guidance.


5. I would encourage the authors to include discussion on the limitations of PCD along with failure analysis with qualitative examples.

**Questions:**

See the Weakness above.

---

> ### Author Response · Authors · 2025-11-20
> **Response to Weaknesses 1 & 2**
>
> We thank the reviewer for taking time to review our work, recognizing the strengths of our method, and providing constructive feedback. In the following, we provide clarifications to your questions and concerns. Please note that all references to sections, line numbers, etc. below refer to the *initial version* of the manuscript unless stated otherwise.
>
> ## W1 Differentiability
>
> > PCD relies on projection and gradient-based updates to enforce constraints. How does it handle test-time constraints that are non-differentiable, for instance, a logic-based rule where a sample is accepted only if it passes some non-differentiable verification? A concrete example would help.
>
> - First, we need to clarify that our approach treats *constraints* and *costs* as separate. Constraints are hard in nature, meaning that we do not want them to be *ever* violated, while costs simply seek to reduce undesirable behavior. In general, constraints are best handled by projection, while costs by coupling.
> - The projection operation in PCD does not require constraints to be differentiable. Crucially, both projection operators used in our experiments (lines 260, 422) do *not* use gradients of the constraints.
> - Indeed, the velocity constraints (line 260) used in our experiments are a specific type of "logic based" rule where samples are projected onto the feasible set $\mathcal K_X$ or $\mathcal K_Y$ each diffusion step. (Note that we do not "reject" or "accept" samples, but instead utilize projection to guarantee constraints are satisfied.)
> - If the reviewer is referring to costs that are determined by "logic-based" rules, this is an interesting point. Our understanding is that the main issue is not so much about differentiability (which we can mostly sidestepped by utilizing subgradients instead of gradients) but *continuity*. We identify two cases here:
>     - (i) For such "logic-based" rules where costs rise to infinity in certain regions, then we reiterate that it is more appropriate to cast them as hard constraints to be handled by projections. Indeed, the velocity constraints (line 262-263) can be seen as a cost that goes to infinity if velocity is outside a particular region and 0 otherwise.
>     - (ii) For costs that are discontinuous but cannot be cast as constraints, e.g., 0-1 losses, there are differentiable approximations such as the softmax or hinge loss; for more complex logic-based costs such as temporal logic specifications (e.g., LTL, STL), differentiable approximations are available [1,2] and have been recently applied to diffusion models [3,4]. Note that in practice, we often do not want discontinuous losses/costs since they can result in instability and brittleness.
>
>
> ## W2 Reliance on clean Tweedie's estimate
>
> > The performance of the proposed method might require the estimated Tweedie to be of high-quality. Otherwise, the further guidance term might likely be inaccurate or even compromise the overall sampling process.
>
> - This perspective is intuitively correct and we partially agree with this. In the experiments, however, we empirically find that our method works well *with or without* the Tweedie estimate. Note that in our proposed framework, we consider the Tweedie's estimate as part of the coupled cost design and it is optional (line 141~161 in the main text).
>     - In the PushT experiment (Table 2 in main paper and Table 13 in the appendix), we have shown results of cost functions of either using the estimated Tweedie (named with a "-PS" suffix) or not, and they all show diversity performance *improvement over the baseline*. Nevertheless, costs with Tweedie's estimates show better preservation of data fidelity as evidenced by higher TC scores comparing to costs without Tweedie, as we already point out in the paper (line 413~415).
>     - In the face pair generation experiment, no Tweedie's estimate was involved since we used a latent classifier to predict the age group of the person directly from noisy image latents (see line 426~427).
>
>
> **References**
> [1] Leung, K., Aréchiga, N., & Pavone, M. (2023). Backpropagation through signal temporal logic specifications: Infusing logical structure into gradient-based methods. *The International Journal of Robotics Research*, 42(6), 356-370.
> [2] Xu, Z., Rawat, Y., Wong, Y., Kankanhalli, M. S., & Shah, M. (2022). Don't pour cereal into coffee: Differentiable temporal logic for temporal action segmentation. *Advances in Neural Information Processing Systems*, 35, 14890-14903.
> [3] Feng, Z., Luan, H., Goyal, P., & Soh, H. (2024). LTLDoG: Satisfying temporally-extended symbolic constraints for safe diffusion-based planning. *IEEE Robotics and Automation Letters*.
> [4] Meng, Y., & Fan, C. (2024). Diverse controllable diffusion policy with signal temporal logic. *IEEE Robotics and Automation Letters*.

---

> ### Author Response · Authors · 2025-11-20
> **Response to Weaknesses 3 & 4**
>
> ## W3 Projection efficiency
>
> > This paper introduces gradient-based guidance and per-step projections, which can increase wall-clock latency compared with non-gradient based baselines. In appendix C.1, the authors also mention "PCD is approximately 4 ∼ 7× slower than vanilla diffusion mainly due to the per-step projection operation." Are there any potential ways to enable faster sampling with PCD?
>
> - This is indeed a crucial consideration for the practical deployment of PCD, as the projection operations are the run time bottleneck. In practice, faster sampling may be achieved through the following simple strategies:
>     - Reduced projection frequency: Perform projection operations every $K$ diffusion steps (including the final step to enforce constraints), rather than at every step.
>     - Approximate projections: Limit the number of optimization iterations during each projection to achieve near-convergence, and only run to full convergence every $K$ diffusion steps (including the final step).
>     - Warm-starting: Warm-start ADMM dual/primal variables from the previous diffusion step.
>
>
> ## W4 Sampling steps ablation
>
> > Could the authors provide a curve of performance v.s. the number of sampling steps to illustrate how will the quality of the estimated Tweedie term affect performance? An intuition is that with more sampling steps, the Tweedie estimate will be more accurate, which could facilitate better guidance.
>
> - Thank you for the helpful suggestion on this ablation. We have added the requested performance vs. sampling steps curves for the PushT benchmark. We have included these curves in the *updated* manuscript at page 19 (and will move them to appropriate sections).
> - In this ablation, we vary the number of sampling steps $T$ while keeping the base diffusion model fixed (a DDPM trained with 100 diffusion steps). The results generally support the reviewer’s intuition that more sampling steps yield more accurate Tweedie's estimates and therefore facilitate guidance.
> - Some key observations from the curves are as below:
>     - For the baseline Diffusion Policy, diversity metrics (DTW and DFD) appear to improve as $T$ decreases. However, this trend is mainly an artifact: when $T<40$, the trajectory sample quality deteriorates significantly (e.g., jerky or teleporting motions), which is also reflected in degraded task completion score (TC) and lower satisfaction rates of velocity constraints (CS). Such low-quality samples often fail the manipulation task, making comparisons for $T<40$ unreliable. Therefore, *we shall focus on meaningful comparisons only for $T\ge 40$*.
>     - When the number of sampling steps $T\ge 40$, the *improvements* in diversity performance *over the baseline* grow steadily as $T$ increases. This pattern is consistent with the hypothesis that guidance (coupling) benefits from more accurate Tweedie's estimates at later denoising steps.
> - Please note that these observations *do not diminish our main contributions in this submission*.

---

> ### Author Response · Authors · 2025-11-20
> **Response to Weakness 5**
>
> ## W5 Include discussions of limitations
>
> > I would encourage the authors to include discussion on the limitations of PCD along with failure analysis with qualitative examples.
>
> - Thank you for highlighting the importance of discussing limitations. These were discussed in different sections of our original manuscript, we are now consolidating them in a standalone section of the *updated* manuscript on page 10.
> - Below we summarize the main limitations and provide clarification:
> 	- The projection step in PCD introduces computational overhead *compared to vanilla or standard guided diffusion*. Please refer to our response to a previous question for practical strategies to mitigate this issue.
>     - PCD introduces a coupling strength hyperparameter, and practitioners may need to tune it based on their desired outcome. We observe degraded original task performance if the coupling strength is set too high, introducing a trade-off between data fidelity and desired correlations between variables. This type of trade-off is common in most training-free, post hoc diffusion guidance methods and is therefore not unique to our method.
>     - PCD assumes that the feasible region of the test-time constraints has an overlap with the support of the distribution learned by the pretrained model. When *extreme* constraints impose a large distribution shift relative to the training data, original task performance can degrade. For example, restricting end-effector velocity to only 1% of its original average in the PushT task would cause PCD to generate trajectories failing the manipulation task at most times. However, this is expected for most test-time only training-free techniques.
>     - PCD assumes differentiability or at least existence of subgradients for the cost functions. However, this is not restricted to our method and is also adopted by other works (e.g., [1-5]). Further, differentiable approximations for discontinuous and non-differentiable costs are also available [6-9].
>
>
> ### References
> [1] Dhariwal, P., & Nichol, A. (2021). Diffusion models beat GANs on image synthesis. *Advances in Neural Information Processing Systems*, 34, 8780-8794.
> [2] Chung, H., Sim, B., Ryu, D., & Ye, J. C. (2022). Improving diffusion models for inverse problems using manifold constraints. *Advances in Neural Information Processing Systems*, 35, 25683-25696.
> [3] Chung, H., Kim, J., Mccann, M. T., Klasky, M. L., & Ye, J. C. (2023). Diffusion posterior sampling for general noisy inverse problems. *The Eleventh International Conference on Learning Representations*.
> [4] Carvalho, J., Le, A. T., Baierl, M., Koert, D., & Peters, J. (2023). Motion planning diffusion: Learning and planning of robot motions with diffusion models. *2023 IEEE/RSJ International Conference on Intelligent Robots and Systems (IROS)*, 1916-1923.
> [5] Römer, R., Rohr, A.v., Schoellig, A.. (2025). Diffusion Predictive Control with Constraints. *Proceedings of the 7th Annual Learning for Dynamics \& Control Conference*, in *Proceedings of Machine Learning Research* 283:791-803.
> [6] Leung, K., Aréchiga, N., & Pavone, M. (2023). Backpropagation through signal temporal logic specifications: Infusing logical structure into gradient-based methods. *The International Journal of Robotics Research*, 42(6), 356-370.
> [7] Xu, Z., Rawat, Y., Wong, Y., Kankanhalli, M. S., & Shah, M. (2022). Don't pour cereal into coffee: Differentiable temporal logic for temporal action segmentation. *Advances in Neural Information Processing Systems*, 35, 14890-14903.
> [8] Feng, Z., Luan, H., Goyal, P., & Soh, H. (2024). LTLDoG: Satisfying temporally-extended symbolic constraints for safe diffusion-based planning. *IEEE Robotics and Automation Letters*.
> [9] Meng, Y., & Fan, C. (2024). Diverse controllable diffusion policy with signal temporal logic. *IEEE Robotics and Automation Letters*.

---

> ### Author Response · Authors · 2025-11-27
> **Gentle Reminder**
>
> As we are approaching the end of the discussion period, we would like to thank the reviewer again for their comments and present a gentle reminder that we have posted a response. May we kindly ask whether our replies address your concerns and improve your evaluation of our paper?
>
> Sincerely,
> The authors

---

### Official Review · Reviewer_fLqz · 2025-10-31

**Soundness:** 3
**Presentation:** 2
**Contribution:** 3
**Rating:** 6
**Confidence:** 2

**Summary:**

The submission proposes a test-time framework to sample from multiple pre-trained diffusion models while (1) promoting some notion of joint "correlation" across variables and (2) enforcing constraints on the individual outputs of each model. PCD augments the usual Langevin / DDPM updates with (1) a user-specified  coupling cost between variables and (2) projections at every diffusion step to guarantee individual constraint satisfaction. Their framework  several existing methods (e.g., classifier guidance, projected diffusion, and some forms of compositional diffusion) as special cases. Empirical demonstrations cover three domains: multi-robot navigation (collision avoidance coupling cost, velocity constraints), robot manipulation on PushT (diverse, non-intersecting trajectories  as a coupling cost with velocity constraints), and ``paired'' face generation (age-contrast coupling with gender/attribute constraints).

**Strengths:**

The paper is well structure and clearly written. The problem of composing several trained diffusion models at test time under constraints is indeed relevant, and prior work has aimed to tackle similar problems when sampling a single variable (e.g. a single image). The main novelty, in my understanding, is the ability to compose diffusion models potentially defined over different variables. The approach is widely applicable and requires minimal modifications of standard sampling algorithms. The use case in muti-robot systems is well motivated, since constraints arise naturally in physical systems and training joint distributions can be computationally costly as the number of agents grows. The toy-example in images -- although artificial -- showcases the applicability of the framework in a totally different setting.

**Weaknesses:**

I think that the notion of "correlated" variables, which is emphasized throughout the paper, could be better defined/explained/motivated. Since the framework is flexible to accommodate arbitrary coupling costs, correlations are in my view an understatement with respect to the practical utility and applicability of the proposed method and underlying problem it is tackling.

In PushT trajectory dissimilarity appears to me as a contrived objective/task, there might be manipulation examples (e.g. bimanual over multiple objects) that lend themselves more naturally/directly to the framework.

**Questions:**

Can you scale the approach beyond two variables/agents?

Can you numerically verify that the two coupling limits match known methods (classifier guidance, projection), at least in a toy experiment?

---

> ### Author Response · Authors · 2025-11-20
> **Response to Questions 1 & 2**
>
> We thank the reviewer for taking time to review our work, recognizing the strengths of our method, and providing constructive feedback. In the following, we provide clarifications to your questions and concerns. Please note that all references to sections, line numbers, etc. refer to the *initial version* of the manuscript unless stated otherwise.
>
> ## Q1 Scaling to multiple variables
>
> > Can you scale the approach beyond two variables/agents?
>
> - Yes, our formulation naturally supports scaling to more than two variables/agents. In fact, we have demonstrated this point in the multi-robot experiment in Section 5.1, where not only 2 but 4 robots are involved in several navigation tasks; please refer to the cost construction in line 313-319 in the main paper, and results in Figure 2 and Table 1. Detailed descriptions of the tasks and settings are in Appendix B.2, with additional results reported in Appendix C.2. Moreover, we have also provided run time profiles for the multi-robot experiment with up to 8 agents in Appendix C.1 (line 1425-1438, Figure 11).
>
>
> ## Q2 Numerical verification of limit cases
>
> > Can you numerically verify that the two coupling limits match known methods (classifier guidance, projection), at least in a toy experiment?
>
> - Thank you for the suggestion. We numerically verify that PCD recovers classifier guidance (CG) and projected diffusion (PD) in the appropriate limits using the 1D toy example from Section 4 (Figure 1). We include these results in the *updated* manuscript at page 18 (and will move them to appropriate sections).
>
> - **Setup**:
>     - We evaluate degenerated versions of PCD that correspond to each limit and compare their empirical sample distributions over the $X$ variable. Specifically, we discretize the range of $X$ variable into 200 bins and compare the resulting histograms with a sample size of $10^{6}$.
>     - PCD vs. CG: We fix the $Y$ variable in PCD to the center of the corridor, matching the conditioning used in CG, and adopt the same coupling/guidance strength for PCD/CG.
>     - PCD vs. PD: We set the coupling strength in PCD as $\gamma=0$, yielding standard PD.
>     - "Baseline vs. Baseline": We draw two groups of samples from the standard normal distribution, each with a different random seed and the same sample size. This is to provide a numerical reference in the reported metrics for samples from the same distribution with this sample size.
> - **Metrics**:
>     - To quantify similarity between distributions, we report three common metrics: Jensen–Shannon divergence (JS), total variation distance (TV), and $L_2$ distance (L2).
> - Results:
>
>     | Method Pair \ Metric           | JS                   | TV                   | L2                   |
>     | ---------------------- | -------------------- | -------------------- | -------------------- |
>     | PCD vs. CG              | $3.91\times 10^{-5}$ | $5.40\times 10^{-3}$ | $1.38\times 10^{-3}$ |
>     | PCD vs. PD              | $5.23\times 10^{-5}$ | $7.97\times 10^{-3}$ | $1.44\times 10^{-3}$ |
>     | Baseline vs. Baseline   | $4.48\times 10^{-5}$ | $5.58\times 10^{-3}$ | $1.32\times 10^{-3}$ |
>
>   - These divergences show that the discrepancies in sample distributions between the degenerated PCD and the corresponding special cases are *at the same order of magnitude* as the *Baseline vs. Baseline* case, wherein two groups of samples are indeed drawn *from the same distribution*.
>   - *Remark*: While *zero discrepancy* is possible with fixed random seeds, we preserve the stochasticity to reflect real diffusion processes.

---

> ### Author Response · Authors · 2025-11-20
> **Response to Weaknesses**
>
> ## W1
>
> > I think that the notion of "correlated" variables, which is emphasized throughout the paper, could be better defined/explained/motivated. Since the framework is flexible to accommodate arbitrary coupling costs, correlations are in my view an understatement with respect to the practical utility and applicability of the proposed method and underlying problem it is tackling.
>
> - Thank you for the comment. We will further clarify this point in the final version of the paper.
>
>
> ## W2 Experiment "naturalness"
>
> > In PushT trajectory dissimilarity appears to me as a contrived objective/task, there might be manipulation examples (e.g. bimanual over multiple objects) that lend themselves more naturally/directly to the framework.
>
> - Thank you for the suggestion in more realistic applications. Even though "trajectory dissimilarity" might seem contrived *on its own*, generating *diverse yet kinematically feasible* low-level trajectory plans is practical and valuable for high-level planning or other applications, such as robust policy selection, online exploration, data augmentation, etc. For instance, the method in [1] (in which the DPP guidance was proposed) relies on diversified diffusion policy as options in a hierarchical reinforcement learning pipeline to follow temporal logic specifications, but without taking kinematic constraints such as velocity limits into account. PCD is a clean yet effective way to produce such alternatives while guaranteeing constraint satisfaction at test time. At the same time, we agree that bimanual/multi-object tasks are natural fits and view them as promising future applications.
>
>
> **Reference**
>
> [1] Feng, Z., Luan, H., Ma, K. Y., & Soh, H. (2025). Diffusion meets options: Hierarchical generative skill composition for temporally-extended tasks. *2025 IEEE International Conference on Robotics and Automation (ICRA)* (pp. 10854-10860).

---

> ### Author Response · Authors · 2025-11-27
> **Gentle Reminder**
>
> As we are approaching the end of the discussion period, we would like to thank the reviewer again for their comments and present a gentle reminder that we have posted a response. May we kindly ask whether our replies address your concerns and improve your evaluation of our paper?
>
> Sincerely,
> The authors

---

### Official Review · Reviewer_Emdc · 2025-11-04

**Soundness:** 2
**Presentation:** 3
**Contribution:** 2
**Rating:** 2
**Confidence:** 2

**Summary:**

The paper studies the problem of combining pretrained diffusion models while enforcing task-specific constraints. The authors present a projected coupled diffusion framework for constrained joint generation. This framework introduces two generative dynamics with coupled  guidance terms, and use projection to impose hard constraints. Several experiments are used to show the effectiveness.

**Strengths:**

- This paper studies the problem of generating samples from pretrained diffusion models while satisfying task-specific constraints. This problem is important because pretrained diffusion models are often available, but sample constraints are typically not enforced during training.

- The authors formulate a new problem of generating correlated samples under hard constraints.

- They propose a generation method based on coupled dynamics, combining a coupled cost with projection onto hard constraints. This approach generalizes projected diffusion to coupled dynamics.

- The authors further show that several existing methods can be viewed as special cases of the proposed coupled dynamics framework.

- Several experiments show benefits from both projected diffusion and cost guidance.

**Weaknesses:**

- The projected coupled dynamics are intuitively designed by combining cost and projection. However, the conditions under which this method converges have not been studied, and to where.

- The effect of coupled costs has not been analyzed, and costs may not always be differentiable.

- The projection step is not discussed in detail. It can be infeasible when the constraints are non-convex. It is also not clear how to do projection for latent diffusion models.

- All special cases correspond to degenerate forms of the projected coupled dynamics. New application scenarios for the projected coupled dynamics have not been explored.

- Experiments demonstrate the effectiveness of combining cost and projection, which is expected since it benefits from both projected diffusion and reward guidance. However, it remains unclear to what extent the experiments reveal the advantages of the coupled dynamics.

**Questions:**

See comments in Weaknesses.

---

> ### Author Response · Authors · 2025-11-20
> **Response to Weaknesses 1 & 2**
>
> We thank the reviewer for taking time to review our work, recognizing the strengths of our method, and providing feedback. In the following, we provide point-wise clarifications to your questions and concerns. Please note that all references to sections, line numbers, etc. refer to the *initial version* of the manuscript unless stated otherwise.
>
> ## W1 "Convergence"
>
> > The projected coupled dynamics are intuitively designed by combining cost and projection. However, the conditions under which this method converges have not been studied, and to where.
>
> - It is unclear what the reviewer means by "convergence" of our method. If it refers to convergence of our projection subroutine (which are mostly iterative in nature), then note that convergence is guaranteed with enough iterations. If, however, the reviewer is referring to "convergence" of the entire diffusion process, then this same problem exists even in vanilla diffusion models. **We hope that the reviewer is able to clarify if something else was meant**.
>
> ## W2 Coupled cost effect and differentiability
>
> > The effect of coupled costs has not been analyzed, and costs may not always be differentiable.
>
> - Coupled costs are task-/domain-specific, *intentionally designed* and *constructed* to induce the desired joint behaviors (e.g., inter-robot collision avoidance in multi-robot navigation, trajectory diversity in PushT, and age-group contrast in image-pair generation).
> - While the coupled costs differ across applications, we have *already quantified and analyzed their effects* (type and strength) on task-related metrics in each experiment with substantial empirical results provided in the manuscript:
>     - Toy example: Figure 1 and lines 190-200 demonstrate the importance of simultaneous coupling.
>     - Multi-robot navigation: Figure 2 and Table 1 (line 270-295) show qualitative and quantitative efficacy in inter-robot collision avoidance for coupling-only variants over the vanilla diffusion; additional tasks, results, and ablations are in Appendix C.2.
>     - PushT manipulation: Table 2 (line 383-386) and Figure 3(b),(d) show that the adopted coupled costs increase trajectory diversity relative to vanilla Diffusion Policy.
>     - Face pairs: Table 3 and Figure 4 in main text, and Figures 22-24 and Tables 14-15 in Appendix C.4, all demonstrate the effectiveness of the XOR coupled cost.
>     - Ablations: Appendix C.2 (line 1550-1565; Figures 16-17) and Appendix C.3 (line 1610-1619; Figures 20-21) study the effect of coupling strength.
> - Differentiability: While we may have assumed in our approach that gradients exist, it can be trivially relaxed to cost functions that possess *subgradients*. We will clarify this in the updated version of the manuscript. This extension naturally handles both our SHD and other ReLU-like costs, as well as most other reasonable continuous cost functions. Furthermore, the subgradient operator is readily supported by modern deep learning frameworks. Moreover, the differentiability assumption is not restricted to our method but common in related work (e.g., [1-5]). Finally, for many practical non-differentiable costs, such as temporal-logic specifications, differentiable approximations are available [6,7] and have recently been applied in diffusion-based generation [8,9].
>
> ### References
>
> [1] Dhariwal, P., & Nichol, A. (2021). Diffusion models beat GANs on image synthesis. *NeurIPS*, 34, 8780-8794.
> [2] Chung, H., Sim, B., Ryu, D., & Ye, J. C. (2022). Improving diffusion models for inverse problems using manifold constraints. *NeurIPS*, 35, 25683-25696.
> [3] Nichol, A.Q., Dhariwal, P., Ramesh, A., Shyam, P., Mishkin, P., Mcgrew, B., Sutskever, I., Chen, M. (2022). GLIDE: Towards Photorealistic Image Generation and Editing with Text-Guided Diffusion Models. *ICML*.
> [4] Chung, H., Kim, J., Mccann, M. T., Klasky, M. L., & Ye, J. C. (2023). Diffusion posterior sampling for general noisy inverse problems. *ICLR*.
> [5] Carvalho, J., Le, A. T., Kicki, P., Koert, D., & Peters, J. (2025). Motion Planning Diffusion: Learning and Adapting Robot Motion Planning With Diffusion Models. *IEEE Transactions on Robotics*, 41, 4881–4901.
> [6] Leung, K., Aréchiga, N., & Pavone, M. (2023). Backpropagation through signal temporal logic specifications: Infusing logical structure into gradient-based methods. *The International Journal of Robotics Research*, 42(6), 356-370.
> [7] Xu, Z., Rawat, Y., Wong, Y., Kankanhalli, M. S., & Shah, M. (2022). Don't pour cereal into coffee: Differentiable temporal logic for temporal action segmentation. *NeurIPS*, 35, 14890-14903.
> [8] Feng, Z., Luan, H., Goyal, P., & Soh, H. (2024). LTLDoG: Satisfying temporally-extended symbolic constraints for safe diffusion-based planning. *IEEE Robotics and Automation Letters*.
> [9] Meng, Y., & Fan, C. (2024). Diverse controllable diffusion policy with signal temporal logic. *IEEE Robotics and Automation Letters*.

---

> ### Author Response · Authors · 2025-11-20
> **Response to Weaknesses 3–5**
>
> We thank the reviewer for taking time to review our work and providing feedback. Please note that all references to sections, line numbers, etc. below refer to the *initial version* of the manuscript unless stated otherwise.
>
> ## W3 Projection details
>
> > The projection step is not discussed in detail. It can be infeasible when the constraints are non-convex. It is also not clear how to do projection for latent diffusion models.
>
> - The projection step is domain- or task-specific, therefore the implementation *details are deferred to Appendix B.2 and B.4*. For instance, in PushT and the multi-robot experiments, the convex set describes velocity constraints and projection is efficiently done via alternating direction method of multipliers (ADMM). For face pair generation, the convex set is the defined by the convex hull of a set of exemplars in the ***latent space***; here, the projection algorithm is based on online mirror descent. For the toy example, the projection is as simple as clipping.
> - Regarding feasibility, please note that the projection is always feasible as long as the set being projected onto is *non-empty*, which is a reasonable assumption to make (see line 100), and this is regardless of convexity. Our assumption that the set $\mathcal K_X$ or $\mathcal K_Y$ is convex is made in order for projection to remain computationally *tractable*. Note that projections onto non-convex sets continue to be feasible (assuming compactness, though not necessarily unique); the issue is computational costs –– in general, efficient algorithms do not exist.
> - We reiterate that the face pair generation experiment is conducted with *a latent diffusion model* (see line 417-418). The projection therein is performed in the latent space (line 421-426), and detailed descriptions of the method with relevant derivations are in Appendix B.4 (line 1234-1293), which is also pointed out in the main paper.
>
>
> ## W4 Applications of projected coupled dynamics
>
> > All special cases correspond to degenerate forms of the projected coupled dynamics. New application scenarios for the projected coupled dynamics have not been explored.
>
> - Because PCD generalizes methods such as projected diffusion and classifier guidance, these methods arise naturally as special or degenerated cases of PCD.
> - Regarding applications, please note that we have provided substantial empirical results for settings where *both* projection and coupling are required for satisfactory performance; see Table 1-3 and Figure 1-4 in the main paper for comparisons when either only coupling or projection is available, with additional results offered in Appendix C.2-C.4.
>
> ## W5 Advantages of coupled dynamics:
>
> > Experiments demonstrate the effectiveness of combining cost and projection, which is expected since it benefits from both projected diffusion and reward guidance. However, it remains unclear to what extent the experiments reveal the advantages of the coupled dynamics.
>
> - We first demonstrated in the toy example (Figure 1 and line 190-200) the advantage of coupled dynamics compared to naive sequential conditional sampling.
> - Then, through multiple experiments across different applications, we have shown the effectiveness of coupled dynamics compared to *vanilla diffusion or projection-only methods*. Please refer to Table 1-3 and Figure 1-4 in the main text for comparisons across baselines, projection-only, coupling-only, and PCD methods, also Appendix C.2-C.4 for additional results and ablations.

---

> ### Comment · Reviewer_Emdc · 2025-11-25
>
> Thank you for the response. I’ve read it and appreciate the clarification.

---

> > ### Author Response · Authors · 2025-11-28
> >
> > Thank you for reading our rebuttal. Since our response clarified and addressed all points you raised regarding convergence, coupled costs, projection details, application scope, and the advantages of coupled dynamics, we wanted to check whether these clarifications resolved the concerns you highlighted.
> >
> > Please let us know if any concerns remain. We would be happy to provide further clarifications.
> >
> > Sincerely,
> > The authors

---

### Author Response · Authors · 2025-11-20
**General Message to All Reviewers**

We thank all reviewers for the time and effort invested in reviewing our submission. We value the suggestions offered across the reviews. Please find our comments to *each individual review* and let us know if there are any further questions.

Sincerely,
The authors

---

### Author Response · Authors · 2025-12-02
**Rebuttal & Updates Summary**

# Summary of Rebuttal and Updates

We thank all reviewers again for their constructive feedback. We summarize the discussion and our key updates below.

## Key Contributions

Our work introduces Projected Coupled Diffusion (PCD), a general test-time framework to jointly sample from multiple pre-trained diffusion models under hard constraints *without retraining*. It unifies existing methods such as projected diffusion and classifier guidance as special cases. We demonstrate PCD across different domains including multi-robot navigation, diverse robot manipulation, and constrained image-pair generation with extensive experiments.

## Strengths Recognized by Reviewers

The reviewers recognize:
1. the impact and timeliness of our unified formulation (Reviewer DJcn),
2. the novelty of composing models over multiple variables with hard constraints (Reviewers Emdc, fLqz),
3. the method's wide applicability and effectiveness across diverse domains supported by extensive empirical results (Reviewers Emdc, fLqz, XYUt), and
4. the paper’s clear presentation (Reviewers Emdc, fLqz, XYUt, DJcn).


## Addressing Reviewers' Concerns

We have *addressed all main concerns* raised by the reviewers and updated the manuscript accordingly:

1. **Added ablations and numerical verification**:
	- We added ablations on sampling steps to analyze the impact of Tweedie’s estimate in one variant of our method (answering Reviewer XYUt; updated on *Page 19 in the revised PDF*).
    - We also added experiments that numerically verify PCD recovers classifier guidance and projected diffusion in special cases (answering Reviewer fLqz; updated on *Page 18*).

2. **Cost differentiability and constraint discussion**:
	- We clarified that our projection operators handle hard constraints without requiring gradients, while costs (even non-differentiable ones) can be handled via subgradients or differentiable approximations (answering Reviewers Emdc, XYUt, DJcn; updated on *Page 3 footnote*).
    - We also discussed projection feasibility and non-convex constraints (answering Reviewers Emdc, XYUt, DJcn).

3. **Scalability**:
    - We *confirmed* scalability beyond two variables (referring to the 4-robot experiments already in the *original* manuscript) and discussed practical strategies to mitigate computational overhead (answering Reviewers fLqz, XYUt).

5. **Theoretical convergence**:
	- We clarified that our projection operators have convergence guarantees as they are standard algorithms used in convex optimization, and noted that global convergence of the entire diffusion process in general is still an open topic in the area and thus *not specific to our method* (answering Reviewer Emdc).

6. **Limitations discussion**:
	- We consolidated a standalone section to discuss limitations such as trade-offs and computational overhead, and reference concurrent work (answering Reviewers XYUt, DJcn; updated on *Page 10*).


We believe that our responses have *answered all questions* from the reviewers and substantially *clarified all main concerns*, and the corresponding revisions have strengthened the paper’s clarity and technical soundness. Two reviewers have *acknowledged* our clarifications. We thank the reviewers and the AC for their time.


Sincerely,
The authors

---

### Meta-Review · Area_Chair_Dj34 · 2026-01-11

**Summary:**

While most reviewers appreciate the interesting idea of composing models over multiple variables with hard constraints and its wide applicability and effectiveness across diverse domains, they also raise several concerns about weak motivation, missing analyses, strong assumptions about cost differentiability and constraint convexity, and limited experiments. The authors’ rebuttal addressed each concern substantively, providing detailed explanations and additional experiments. The final scores were 2(Emdc), 6(fLqz), 6(XYUt), and 6(DJcn). AC feels that the authors address most of the critical issues; the assumptions of cost differentiability and constraint convexity may not hold, but the issue is common to other methods, and the authors suggest potential solutions. AC finds the concerns of the only negative reviewer, Emdc, are well addressed point by point, but the reviewer with low confidence did not update the score. Considering all of these, AC decided to recommend acceptance of this submission, encouraging the authors to carefully incorporate the rebuttal content into their final manuscript.

**Reviewer Concerns:**

The main concerns were about weak motivation, missing analyses, strong assumptions about cost differentiability and constraint convexity, and limited experiments. The authors’ rebuttal addressed most of the critical concerns, providing detailed explanations and additional experiments.  The assumptions of cost differentiability and constraint convexity may not hold, but the issue is common to other methods, and the authors suggest potential solutions: e.g., projection operators handle hard constraints without requiring gradients, while costs can be handled via subgradients or differentiable approximations. AC finds no outstanding critical concerns remaining.

**Reviewer Scores:**

Reviewer Emdc would have changed the score from 2 to 6.
Reviewer fLqz would have retained the original rating of 6.
Reviewer XYUt would have changed the score from 6 to 8.
Reviewer DJcn retained the original rating of 6 after discussion.

---

### Decision · Program_Chairs · 2026-01-26

Accept (Poster)